# C-terminal amides mark proteins for degradation via SCF–FBXO31

Matthias F. Muhar[1,7], Jakob Farnung[2,3,7], Martina Cernakova[1], Raphael Hofmann[2], Lukas T. Henneberg[3], Moritz M. Pfleiderer[4], Annina Denoth-Lippuner[5], Filip Kalčic[2], Ajse S. Nievergelt[4], Marwa Peters Al-Bayati[1], Nikolaos D. Sidiropoulos[1], Viola Beier[3], Matthias Mann[6], Sebastian Jessberger[5], Martin Jinek[4], Brenda A. Schulman[3], Jeffrey W. Bode[2✉] & Jacob E. Corn[1✉]

During normal cellular homeostasis, unfolded and mislocalized proteins are recognized and removed, preventing the build-up of toxic byproducts[1]. When protein homeostasis is perturbed during ageing, neurodegeneration or cellular stress, proteins can accumulate several forms of chemical damage through reactive metabolites[2,3]. Such modifications have been proposed to trigger the selective removal of chemically marked proteins[3–6]; however, identifying modifications that are sufficient to induce protein degradation has remained challenging. Here, using a semi-synthetic chemical biology approach coupled to cellular assays, we found that C-terminal amide-bearing proteins (CTAPs) are rapidly cleared from human cells. A CRISPR screen identified FBXO31 as a reader of C-terminal amides. FBXO31 is a substrate receptor for the SKP1–CUL1–F-box protein (SCF) ubiquitin ligase SCF–FBXO31, which ubiquitylates CTAPs for subsequent proteasomal degradation. A conserved binding pocket enables FBXO31 to bind to almost any C-terminal peptide bearing an amide while retaining exquisite selectivity over non-modified clients. This mechanism facilitates binding and turnover of endogenous CTAPs that are formed after oxidative stress. A dominant human mutation found in neurodevelopmental disorders reverses CTAP recognition, such that non-amidated neosubstrates are now degraded and FBXO31 becomes markedly toxic. We propose that CTAPs may represent the vanguard of a largely unexplored class of modified amino acid degrons that could provide a general strategy for selective yet broad surveillance of chemically damaged proteins.

Cellular protein homeostasis is the essential process of regulating the biogenesis, localization and turnover of proteins. Selective degradation of proteins by the ubiquitin-proteasome system is a central effector mechanism in protein homeostasis[7]. At the molecular level, degradation is typically initiated by post-translational modifications (PTMs), such as Lys48-linked poly-ubiquitylation, that mark proteins for proteolysis by the proteasome. The specificity of this system is established by over 600 human ubiquitin ligases that can bind to specific interaction motifs, termed degrons, on their respective client proteins[8].

While many well-studied PTMs are deposited or erased by dedicated enzymes, amino acid side chains and the protein backbone itself can also experience a plethora of non-enzymatic modifications, including oxidative damage or alkylation[2,9]. In total, over 700 protein modifications have been described to date[10]. For the majority of these, their role in protein homeostasis remains unexplored, in part due to a lack of scalable experimental methods. Non-canonical PTMs accumulate after proteasome inhibition[5], and treatment with alkylating or oxidating

agents can stimulate protein turnover[11–13]. Particular chemical modifications have therefore long been postulated as marks for protein damage to trigger protein clearance[3]. But specific modifications sufficient to induce degradation and the mechanisms by which they are recognized remain largely unclear.

## C-terminal amides function as degrons

We sought a reductionist approach to elucidate the effect of individual chemical modifications on protein turnover. To this end, we prepared a set of semi-synthetic fluorescent reporters and introduced them into human cells to measure their stability (Fig. 1a and Extended Data Fig. 1a). Using solid-phase peptide synthesis, we generated a set of peptides carrying defined modifications representing different types of protein damage: tyrosine modification through oxidation or misincorporation (L-3,4-dihydroxyphenylalanine, L-DOPA)[14], carbonylation (N(6)-hexanoyllysine)[15], advanced glycation end products (N(6)-carboxymethyllysine)[16], carbamylation (homocitrulline)[4] and

[1]Institute of Molecular Health Sciences, Department of Biology, Swiss Federal Institute of Technology (ETH) Zurich, Zurich, Switzerland. [2]Laboratory for Organic Chemistry, Department of Chemistry and Applied Biosciences, Swiss Federal Institute of Technology (ETH) Zurich, Zurich, Switzerland. [3]Department of Molecular Machines and Signaling, Max Planck Institute of Biochemistry, Martinsried, Germany. [4]Department of Biochemistry, University of Zurich, Zurich, Switzerland. [5]Laboratory of Neural Plasticity, Faculties of Medicine and Science, Brain Research Institute, University of Zurich, Zurich, Switzerland. [6]Department of Proteomics and Signal Transduction, Max Planck Institute of Biochemistry, Martinsried, Germany. [7]These authors contributed equally: Matthias F. Muhar, Jakob Farnung. ✉e-mail: bode@org.chem.ethz.ch; jacob.corn@biol.ethz.ch

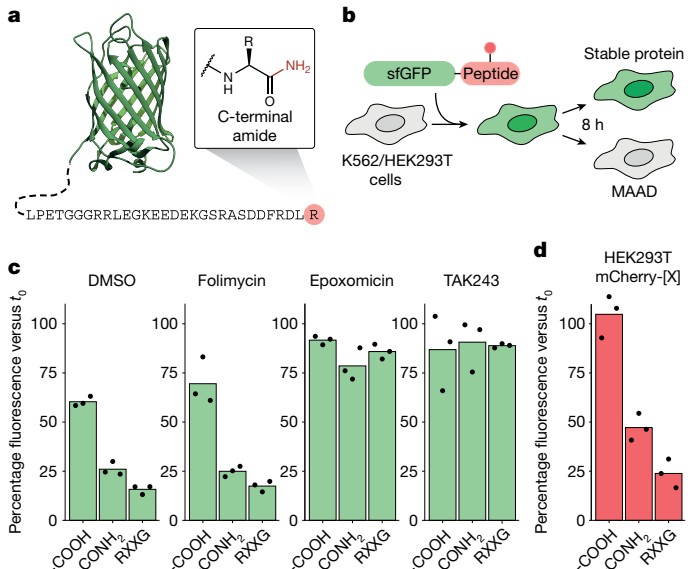

**Fig. 1 | Proteins with C-terminal amides are selectively degraded by the ubiquitin proteasome system. a**, Schematic of an sfGFP-based semi-synthetic reporter protein carrying a C-terminal amide. **b**, Schematic of a fluorescent in-cell reporter assay to distinguish modified amino acid degrons (MAADs) from neutral modifications by electroporation of reporter proteins into human cell lines. **c**, The results of an in-cell reporter assay for the sfGFP conjugate shown in **a** with (·CONH$_2$) or without (·COOH) a terminal amide. RXXG denotes an sfGFP variant containing a positive control degron motif (LPETGGGRRLEGKEEDEKGSRASDRFRGLR). K562 cells received mock treatment (DMSO), lysosomal inhibitor folimycin (100 nM), proteasome inhibitor epoxomicin (500 nM) or E1 ubiquitin ligase inhibitor TAK243 (1 μM) after sfGFP delivery. **d**, Reporter assay as in **c** in HEK293T cells using the same peptides conjugated to mCherry. For **c** and **d**, data are the mean of three independent experiments (*n* = 3), each represented by black dots.

primary-amide-forming backbone cleavage (C-terminal amide)[17,18]. Sortase-A-mediated conjugation of these peptides onto the C terminus of recombinant fluorescent proteins yielded a set of chemically defined modified reporters (Extended Data Fig. 1b,c).

We measured the effect of each modification on protein degradation in the human erythroleukaemia cell line K562 using a fluorescence in-cell reporter assay. We introduced the individual semi-synthetic proteins by electroporation into human cells and followed their clearance over time by flow cytometry (Fig. 1b). In this setting, unconjugated superfolder GFP (sfGFP−SRT−His$_6$) alone was highly stable ($t_{1/2}$ > 16 h), while installation of a strong degron sequence derived from the human ASCC3 C terminus (RXXG)[19] induced rapid degradation (Extended Data Fig. 1d). None of the internal modifications tested affected protein stability in this cell line and amino acid context, suggesting that they are not universally sufficient to induce protein degradation (Extended Data Fig. 1e).

By contrast, sfGFP carrying a C-terminal amide was rapidly degraded in two independent sequence contexts (Fig. 1c and Extended Data Fig. 1e). Its degradation was prevented by inhibitors of ubiquitylation (TAK243) or the proteasome (epoxomicin), but not by inhibition of lysosomal acidification (folimycin). This implies that CTAPs are actively cleared from cells by the ubiquitin proteasome system. CTAP forms of both mCherry and mTagBFP2, but not their unmodified counterparts, were also efficiently degraded when delivered to human embryonic kidney (HEK)-derived HEK293T cells (Fig. 1d and Extended Data Fig. 1f). The presence of a C-terminal amide is therefore sufficient to induce protein degradation in different peptide, protein and cell contexts.

## SCF−FBXO31 mediates CTAP clearance

To identify the cellular machinery underlying CTAP clearance, we devised a genome-wide CRISPR screen for genes that are responsible for specific degradation of C-terminally amidated sfGFP (sfGFP-CONH$_2$; Fig. 2a). As knockout of central protein quality-control and turnover genes may impede cell survival, we generated a clonal K562 cell line with a tightly controllable Cas9 allele (iCas9) (Extended Data Fig. 2a,b). After transduction with a genome-wide sgRNA library[20], Cas9 expression was induced for 5 days to allow for efficient knockout while minimizing the loss of cells with defects in essential pathways[21]. We then delivered sfGFP-CONH$_2$ by electroporation together with mTagBFP2−RXXG, to control for effects on general protein turnover. After short-term culture allowing for protein degradation (14 h), we isolated cells that were proficient in general protein turnover but deficient in CTAP clearance (sfGFP$^+$mTagBFP2$^-$), and a control population with no defect in protein clearance (sfGFP$^-$mTagBFP2$^-$). Next-generation sequencing analysis of sgRNAs from both populations revealed a marked enrichment of few targeted genes in the CTAP-clearance-deficient population.

The most prominent screen hit was *FBXO31* (Fig. 2b and Supplementary Table 1), a substrate adaptor for the SCF ubiquitin ligase assembly. The screen also identified SCF's central scaffolding protein CUL1 and several subunits of the COP9 signalosome complex, which regulates SCF function[22] (Extended Data Fig. 2c). Individually silencing *FBXO31* by CRISPR inhibition (CRISPRi)[23] using two independent guide RNAs in K562 cells completely stabilized the CTAP reporter protein, while the RXXG degron form remained unaffected (Fig. 2c). *FBXO31* knockout in HEK293 cells also abolished CTAP clearance (Extended Data Fig. 2d). Overexpression of wild-type FBXO31 rescued CTAP degradation in the knockout background, but a mutant of FBXO31 lacking the F-box domain required for SCF assembly did not (Extended Data Fig. 2d). These findings establish SCF−FBXO31 as a central effector of CTAP clearance in the context of multiple model substrates and multiple cellular backgrounds.

We mechanistically investigated the role of SCF−FBXO31 in CTAP clearance using multiple orthogonal assays. We first tested whether SCF−FBXO31 directly binds to and ubiquitylates amidated clients. We purified recombinant FBXO31 in a complex with its binding partner SKP1 and measured its affinity for various peptides by fluorescence polarization (FP). In vitro, FBXO31 bound to the peptide amide used for screening with high affinity ($K_D$ = 16 ± 2 nM) while no binding could be detected for the carboxylic acid form (Fig. 2d). This interaction was also confirmed in cells by co-immunoprecipitation (co-IP) of affinity-tagged FBXO31, which exhibited binding exclusively to the model substrate mCherry-CONH$_2$, but not the unmodified counterpart (Extended Data Fig. 2e).

To test whether FBXO31 binding leads to productive ubiquitylation of substrates, we reconstituted the full SCF−FBXO31 ubiquitin ligase assembly from recombinant components. The SCF−FBXO31 complex ubiquitylated sfGFP-CONH$_2$ in vitro with high efficiency but showed no detectable activity towards sfGFP-COOH (Extended Data Fig. 2f). In reactions simultaneously containing both unmodified and amide-bearing substrates, SCF−FBXO31 exclusively ubiquitylated the CTAP form. SCF−FBXO31 is therefore capable of precisely distinguishing amidated from non-amidated substrates without crosstalk (Fig. 2e). Together these results establish that SCF−FBXO31 is a reader of C-terminal protein amides, and that this amide is required for ubiquitylation.

## FBXO31 broadly binds to CTAPs

If FBXO31 is a general surveillance factor for C-terminal amides, it would need to sample a broad range of C-terminal sequences while maintaining high selectivity for amides over carboxylic acids. Moreover, it could not simply bind to any primary amide group, as these are

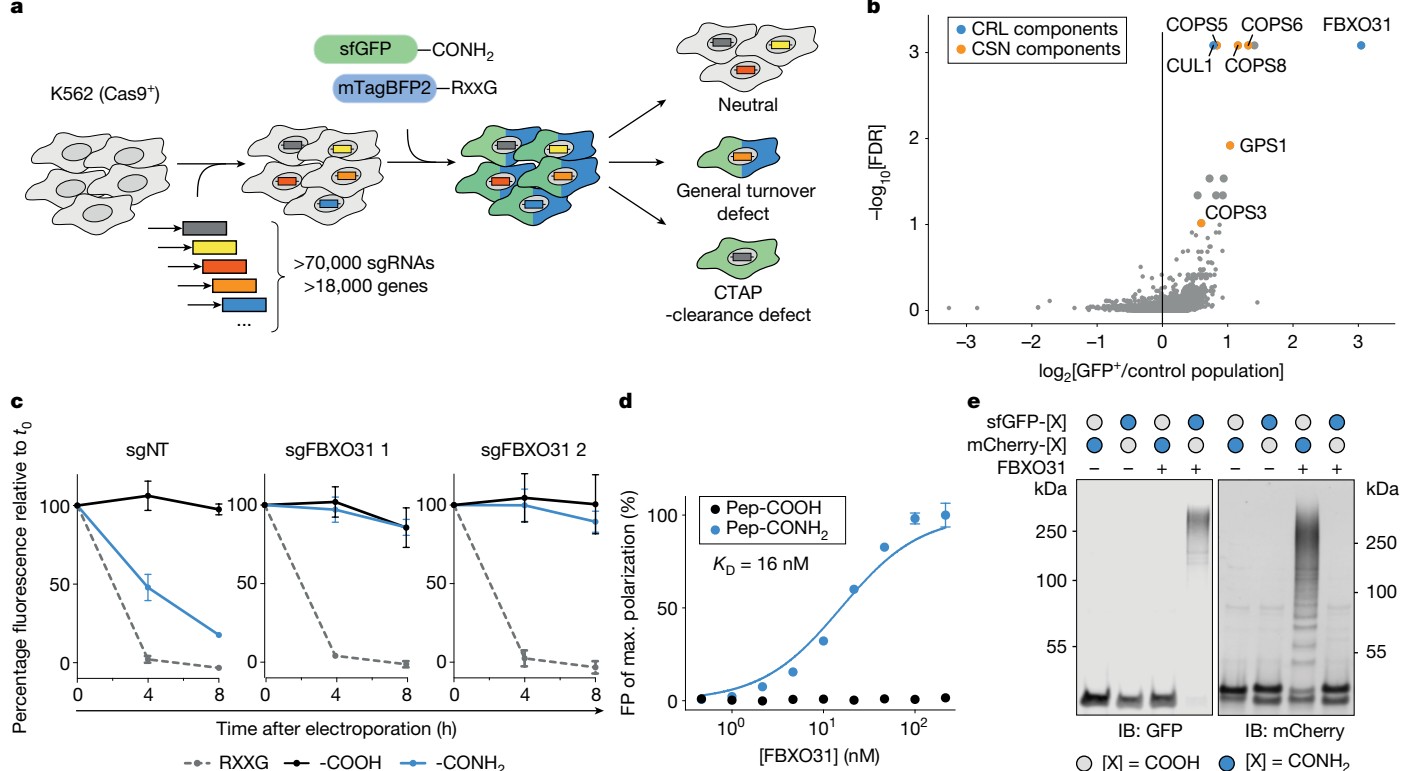

**Fig. 2 | CRISPR screen identifies SCF–FBXO31 as a CTAP-clearance factor.**
**a**, Schematic of a genome-wide CRISPR screen to identify genes required for CTAP clearance. Model substrates were sfGFP–GGGKDLEGKGGSAGSGSAGG SKYPYDVPDYAKS-CONH$_2$ (sfGFP–pep1-CONH$_2$) and mTagBFP2–GGGRRLEGKE EDEKGSRASDRFRGLR-COOH (pep2–RXXG). **b**, Screening results showing the mean enrichment of sgRNAs targeting each gene in CTAP-clearance-deficient cells (sfGFP$^+$mTagBFP2$^-$) compared with the control population (sfGFP$^-$mTagBFP2$^-$) on the abscissa. Ordinate values show the false-discovery-rate (FDR)-adjusted significance of enrichment across sgRNAs and duplicate screens. **c**, The time course of sfGFP degradation in CRISPRi-competent K562 cells expressing sgRNAs targeting *FBXO31* and a non-targeting control (NT). sfGFP was conjugated to pep1 or the positive control degron RXXG as in **b**. Data are mean ± s.d. of *n* = 3 independent experiments. **d**, FP assay of fluorescently labelled peptide (pep2-short, KEEDEKGSRASDDFRDLR) with the indicated C termini. Data are mean ± s.d. of *n* = 3 parallel experiments. **e**, In vitro ubiquitylation assay using sfGFP–pep1 and mCherry–pep2 with the indicated C termini, recombinant FBXO31, remaining E3 complex members (SKP1, CUL1–NEDD8, RBX1), E1 (UBA1) and E2 (UBE2R1, UBE2D3) enzymes. Gel source data are provided in Supplementary Fig. 1. The blot is representative of two independent experiments.

found in asparagine and glutamine side chains of almost all proteins. We therefore set out to determine the substrate scope of FBXO31 using in vitro binding studies.

We first tested whether FBXO31 specifically binds C-terminal amides rather than side-chain amides in asparagine or glutamine. FP assays using recombinant FBXO31–SKP1 and fluorescently labelled peptides showed no affinity for peptides with unmodified Asn or Gln at the C terminus (Fig. 3a), but high affinity for peptides carrying a C-terminal primary amide (X-Asn-CONH$_2$, $K_D$ = 24 ± 3 nM; X-Gln-CONH$_2$, $K_D$ = 55 ± 4 nM). Extending this assay to peptides bearing the primary amide derivatives of each of the 20 natural amino acids revealed that FBXO31 binds to almost any C-terminal amide with nanomolar affinity (Fig. 3b and Extended Data Fig. 3). The weakest binders were peptides bearing glycine and acidic residues, with X-Asp-CONH$_2$ still exhibiting a $K_D$ of 304 ± 22 nM. Hydrophobic residues were most strongly bound, with X-Phe-CONH$_2$ binding even more strongly ($K_D \approx 6$ nM), followed by uncharged and charged hydrophilic side chains. These findings demonstrate that FBXO31 binds to diverse C-terminal amides with high affinity and selectivity over unmodified C termini and side-chain amides.

We extended the screen of amidated C termini by devising a massively parallel protein–peptide interaction screen to identify sequence preferences for FBXO31 binding (Extended Data Fig. 4a). Using isokinetic mixtures of 19 natural amino acids (all except cysteine) in the first three coupling steps, we synthesized peptide libraries containing more than 2,000 distinct C termini detectable by mass spectrometry (MS) (Supplementary Table 2). We performed in vitro co-IP of the library and

quantified the abundance of each peptide alongside the input pool using isobaric labelling and MS.

Overall, FBXO31 exhibited broad binding to 841 distinct C-terminal-amide-bearing sequences, and just 73 unmodified C termini. The C-terminal amides showed 7.6-fold greater enrichment compared with the few unmodified peptides detected in this assay (Extended Data Fig. 4b). Consistent with FP measurements, FBXO31 preferred hydrophobic side chains, while acidic residues were less efficiently bound, especially in the terminal position (Extended Data Fig. 4c). Despite these preferences, each tested amino acid was detected in all three ultimate positions among bound peptides (Extended Data Fig. 4d,e). To validate that these in vitro measurements also translate to CTAP-clearance in cells, we measured degradation for sfGFP conjugated to the two top-scoring motifs (VIN, YNR) in HEK293T cells. Both motifs were destabilized by the addition of a C-terminal amide in a cellular assay, which was rescued by *FBXO31* knockdown (Extended Data Fig. 4f,g).

Overall, we conclude that FBXO31 specifically recognizes C-terminal peptide amides with a slight preference for hydrophobic termini. However, in contrast to conventional sequence-based C-terminal degrons, FBXO31 is mostly agnostic to specific sequence motifs, potentially enabling it to broadly surveil C-terminal amides across the proteome.

## CTAPs bind to a conserved pocket in FBXO31

CTAPs are characterized by the isosteric substitution of an oxygen atom with nitrogen and the associated loss of a negative charge under

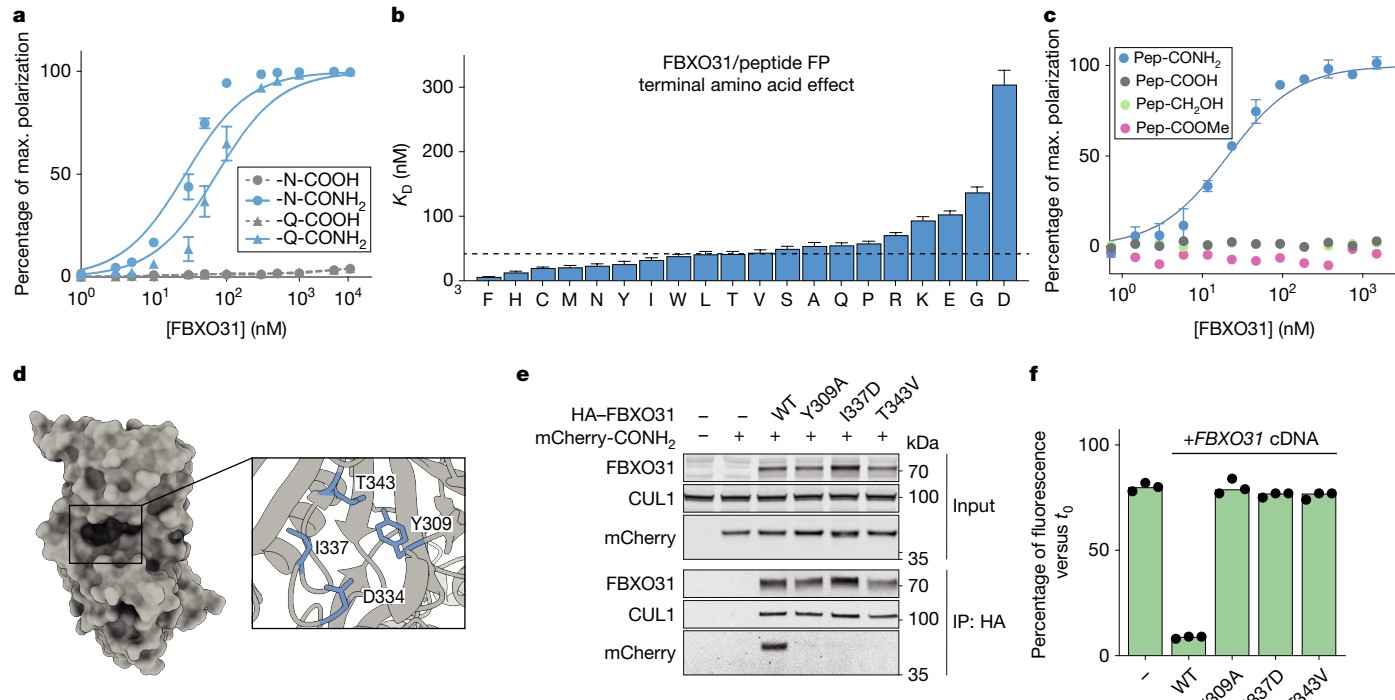

**Fig. 3 | FBXO31 broadly and selectively binds to CTAPs. a**, FP assay of pep3 with the indicated terminal amino acids and modifications with increasing FBXO31 concentrations. **b**, The $K_D$ of the interaction between FBXO31 and C-terminally amidated pep3 (KKYRYDVPDYSA[X]-CONH₂) as in **a** for different amino acids in the terminal position. The dashed line indicates the median across all 20 amino acids. The underlying binding curves and individual FP measurements ($n = 3$ parallel experiments) are shown in Extended Data Fig. 3. **c**, FP assay of fluorescein-labelled peptide (KEEDEKGSRASDDFRDLR) with increasing concentrations of FBXO31 for the indicated C-terminal modifications. **d**, The crystal structure of FBXO31 from Protein Data Bank (PDB) 5VZT. Inset:

magnification of FBXO31's substrate-binding pocket. **e**, Co-immunoprecipitation and immunoblotting of a model CTAP (mCherry–pep2-CONH₂) from *FBXO31*-knockout HEK293 cells expressing HA-tagged cDNAs of wild-type FBXO31 or the indicated mutants. The blot is representative of two independent experiments. **f**, Rescue of CTAP degradation in *FBXO31*-knockout HEK293 cells with the indicated *FBXO31* mutant cDNAs from **e** for an independent model CTAP (GFP–GGGKYSESATPESKGGSKGF-CONH₂). For **a–c**, data are mean ± s.d. of $n = 3$ parallel experiments. For **f**, data are the mean of three independent experiments ($n = 3$), each shown as black dots. Gel source data are provided in Supplementary Fig. 1.

physiological conditions. We examined how FBXO31 exhibits such strong and selective binding to amide-bearing C termini. Conventional unmodified C-terminal degrons are typically recognized through positive charges on their respective ubiquitin ligases[24–27]. By contrast, a previous crystal structure of FBXO31 in a complex with an unmodified C-terminal peptide from cyclin D1 revealed a substrate-binding pocket that exhibits a negative charge at Asp334[28] and might therefore disfavour unmodified C termini (Extended Data Fig. 5a,b).

To test whether the exclusion of a negatively charged C terminus alone underlies CTAP selectivity, we measured FBXO31 binding of various modified peptides using FP. Replacing the negatively charged C terminus (-COOH) with an alcohol (-CH₂OH) or blocking it with a methyl ester (-COOMe) did not induce FBXO31 binding (Fig. 3c). These data suggest that specific, additional interactions with the terminal primary amide are required for binding.

To assess the contribution of residues in the FBXO31-binding pocket to CTAP recognition, we overexpressed affinity-tagged *FBXO31* cDNAs in an *FBXO31*-knockout background and measured the effect of individual point mutations on CTAP binding using co-IP (Fig. 3d,e). All of the mutants tested retained the ability to form SCF assemblies, as shown by co-IP with CUL1. However, mutations in the binding pocket completely abolished binding to a model CTAP, either when mutating hydrophobic residues in the binding pocket (Y309A and I337D) or conservatively replacing a putative hydrogen-bonding residue (T343V). cDNA re-expression of each mutant in an *FBXO31*-knockout background revealed that they also failed to degrade CTAPs (Fig. 3f).

On the basis of these and the above findings, we propose that the selectivity of FBXO31 for C-terminal amides arises from the combined

charge exclusion of carboxylic acids and direct interaction with the primary amide of its clients.

## CTAPs form after oxidative cleavage

To the best of our knowledge, C-terminal amidation has so far not been reported for intracellular proteins. However, enzymatic and non-enzymatic mechanisms for terminal amide formation have been described in other contexts. In vivo, most secreted peptide hormones are processed by peptidyl glycine α-amidating monooxygenase (PAM) using copper-catalysed hydroxyl radical attack and subsequent cleavage of a terminal glycine to yield shortened C-terminally amidated peptides[18]. In vitro, hydroxyl radicals can trigger peptide bond cleavage by a similar mechanism, giving rise to C-terminal amide fragments[3,29] (Fig. 4a). We therefore hypothesized that CTAP degradation through FBXO31 could serve to eliminate protein fragments arising from mistrafficking of secretory proteins or oxidative damage.

We reanalysed public deep proteome profiles of healthy human donors for signatures of CTAP formation. If CTAPs arise from α-amidating cleavage, we would expect to detect peptides with C-terminal amidation (−0.984 Da) together with a neo-C-terminus (not the annotated protein C terminus or peptides formed by tryptic digest) (Extended Data Fig. 6a). Neo-C-termini were overall rare, but formed the majority for peptides with a CTAP-associated −0.984 Da shift, indicating that amidation did not occur at native or tryptic termini during sample processing (Extended Data Fig. 6d). In the inverse analysis, C-terminal amidation was nearly absent from annotated protein C termini and regular tryptic peptides, but strongly enriched among peptides with

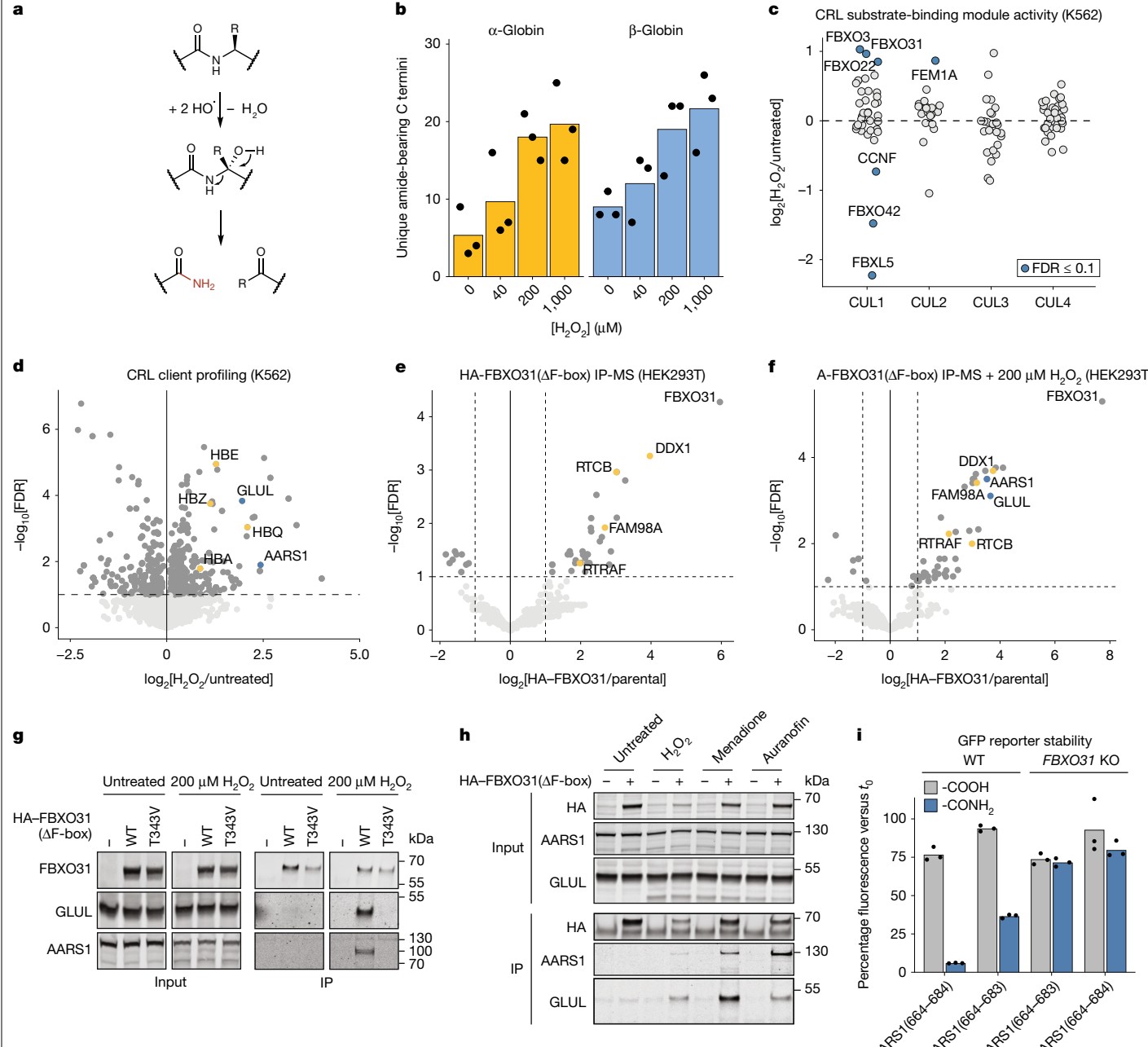

**Fig. 4 | FBXO31 recognizes endogenous CTAPs formed under oxidative stress. a**, The proposed model of C-terminal amidation by PAM and hydroxyl radicals. **b**, Reconstitution of in vitro CTAP formation on purified human haemoglobin by hydrogen peroxide for 1 h at 37 °C. Amidated neo-C-termini (−0.984 Da) were identified by tryptic digest and MS/MS. **c**, Activity-based profiling of CRLs after oxidative challenge (2 h, 200 μM $H_2O_2$) of K562 cells. The y axis shows differential enrichment of substrate-binding modules by isolation of neddylated CRLs after treatment. **d**, Profiling of the CRL-bound proteome after oxidative damage as in **c**. Haemoglobin subunits present in K562 cells are indicated in yellow. **e**, IP–MS analysis of HA-tagged FBXO31(ΔF-box) expressed from cDNA in *FBXO31*-knockout HEK293T cells. The components of the tRNA ligase complex are highlighted in orange. **f**, IP–MS analysis as in **e** for HEK293T

neo-C-termini (Extended Data Fig. 6c). We conclude that C-terminal amides primarily form after protein fragmentation.

Applying these criteria and stringent confidence thresholds, we identified up to 427 CTAP cleavage sites affecting up to 165 different proteins per tissue sample including haemoglobin subunits (Extended Data Fig. 6d,e and Supplementary Table 3). Notably,

cells treated for 20 min with 200 μM $H_2O_2$. The components of the tRNA ligase complex are highlighted in orange. Clients AARS1 and GLUL are highlighted in blue. **g**, Validation of FBXO31 clients by co-IP of cells treated as in **f**. **h**, Co-IP analysis of FBXO31 and endogenous clients as in **g** for cells treated with $H_2O_2$ (20 min, 200 μM), menadione (2 h, 10 μM) or auranofin (2 h, 10 μM). **i**, Validation of degron activity for AARS1-derived CTAP-fragments. The indicated AARS1 residues were conjugated to sfGFP by sortylation and delivered to HEK293T cells followed by quantification of protein levels using flow cytometry. For **b** and **i**, data are the mean of three independent experiments (n = 3), each shown as black dots. Gel source data are provided in Supplementary Fig. 1. Blots in **g** and **h** are representative of two independent experiments.

haemoglobin-bound Fe(II) can catalyse hydroxyl radical formation, which might promote CTAP formation. MS analysis of purified adult haemoglobin showed CTAP formation for both subunits, HBA and HBB, which was exacerbated by $H_2O_2$ exposure (Fig. 4b, Extended Data Fig. 7a,b and Supplementary Table 3). The majority of the in vivo cleavage sites identified in deep proteomes also occurred in vitro,

confirming that in vivo identified CTAPs can form by protein fragmentation (Extended Data Fig. 7c).

Hydroxyl-radical damage could, in principle, affect any position in any protein. Accordingly, high-coverage mapping of in vitro CTAP formation identified 77 cleavage sites throughout HBA and HBB at 14 different amino acids (Extended Data Fig. 7c). Sequence analysis indicates the presence of all 20 amino acids flanking α-amidating cleavage sites with weak enrichment of Cys and Lys (Extended Data Fig. 7d). This supports a model of CTAP formation by diffuse backbone breakage and is consistent with FBXO31's ability to broadly recognize CTAPs independent of the terminal amino acid identity.

## FBXO31 responds to cellular oxidation

Having established peptide fragmentation as a source of CTAPs, we examined whether FBXO31 forms part of the cellular response to oxidative stress. Cullin ring ligases (CRLs) form the largest family of ubiquitin ligases with more than 300 different assemblies[30], including SCF–FBXO31. We mapped the composition and clients of ubiquitin CRL complexes after oxidative stress in K562 cells using a recently developed activity-based profiling method (CRL-ABP)[31]. Affinity purification of active CRL complexes followed by MS identified SCF–FBXO31 as the second most strongly stress-activated CRL (Fig. 4c and Supplementary Table 4). Analysis of co-purified client proteins revealed extensive rewiring of the CRL-associated proteome, with 266 proteins showing increased CRL targeting (Fig. 4d). Among them were all four haemoglobin subunits expressed in K562 cells (HBA, HBE, HBZ and HBQ), consistent with our finding that these form CTAPs in vivo and in vitro (Fig. 4b, Extended Data Fig. 7a–c and Supplementary Table 3).

To test whether FBXO31 directly recognizes clients in cells, we screened for FBXO31-targets using co-IP followed by MS (IP–MS). We stably expressed catalytically inactive affinity-tagged FBXO31 (HA–FBXO31(ΔF-box)) in *FBXO31*-knockout HEK293T cells and performed IP–MS under basal conditions and under oxidative stress. Under basal conditions, FBXO31 co-purified with relatively few proteins but, notably, with all of the core members of the tRNA ligase complex (Fig. 4e and Supplementary Table 5), which is subject to oxidative hydroxyl radical damage, in particular in the presence of Cu(II)[32]. MS analysis of recombinantly expressed and purified tRNA ligase complex confirmed CTAP formation under oxidizing conditions with CuCl$_2$ (Extended Data Fig. 7e,f).

After acute oxidative stress (20 min, 200 μM H$_2$O$_2$), IP–MS analysis of FBXO31 detected several additional putative clients. Notably, these included the aminoacyl tRNA synthetase AARS1 and the glutamine synthetase GLUL. Returning to the unbiased CRL-ABP datasets, we found that both AARS1 and GLUL were also oxidation-induced CRL targets in K562 cells (Fig. 4d and Supplementary Tables 4 and 5). Individual co-IP western blotting confirmed that FBXO31(ΔF-box) binds to GLUL and AARS1 in cells only after H$_2$O$_2$ treatment (Fig. 4g). This binding is furthermore dependent on Thr343 in the FBXO31 substrate-binding pocket, which is also critical for CTAP recognition. Client recognition was also induced by shifting the cellular redox balance, either by stimulating redox cycling with menadione or by inhibiting the antioxidant thioredoxin reductase with auranofin (Fig. 4h). FBXO31 therefore preferentially recognizes multiple endogenous clients in response to multiple forms of oxidative stress.

We tested whether the identified FBXO31 clients undergo primary amide-forming fragmentation in cells. Affinity-tagged AARS1 in *FBXO31*-knockout cells was immunoprecipitated after oxidative challenge (1 h, 200 μM H$_2$O$_2$). A search for de novo C termini of AARS1 identified two internal cleavage sites with a mass shift of −0.984 Da at their C termini, consistent with amide modification (Supplementary Table 3). Amidated versions of both AARS1 C-terminal peptides induced potent reporter degradation in cells (Fig. 4i). This effect was only observed in the presence of a C-terminal amide and was entirely rescued by *FBXO31* knockout.

A recessive frameshift mutation in FBXO31 has been associated with neurodevelopmental disorders and intellectual disability, suggesting the potential for a neuron-specific phenotype[33]. We observed no impact of *FBXO31* knockout on cell fitness in rapidly proliferating cultured cells, and RNA-sequencing (RNA-seq) showed no transcriptional response to *FBXO31* knockdown in HEK293T cells (Extended Data Fig. 8a and Supplementary Table 6). However, RNA-seq analysis of in vitro derived neural progenitor cells (NPCs) engineered for CRISPR inhibition (CRISPRi) and isogenic differentiated neurons revealed a different picture (Extended Data Fig. 8b). *FBXO31* knockdown in NPCs induced a transcriptional response that was exacerbated in mature neurons (Extended Data Fig. 8c,d and Supplementary Table 6). Comparison with public RNA-seq datasets of stem-cell-derived neurons[34] identified FBXO31 responses as a subset of a signature induced by mutations causing amyotrophic lateral sclerosis (Extended Data Fig. 8e,f). While FBXO31 is ubiquitously expressed, its activity may be most required by long-lived, metabolically active cells, such as neurons, where damaged proteins can accumulate.

On the basis of our results, we conclude that FBXO31 acts as a quality-control factor that recognizes and removes endogenous CTAPs formed under oxidative stress. This is consistent with the ability of FBXO31 to broadly recognize amidated termini that could theoretically occur anywhere within a protein. As random damage across the proteome would evade detection by shotgun MS, the reported CTAPs may represent hotspots of more widespread damage by hydroxyl radicals.

## A pathogenic mutation redirects FBXO31

Loss of function mutations in FBXO31 cause recessive intellectual disability. Two studies have also reported a dominant de novo D334N mutation in FBXO31 among patients with diplegic spastic cerebral palsy[35,36]. This mutation eliminates the negative charge in the substrate-binding pocket of FBXO31 and was speculated to act by increased degradation of cyclin D1. We examined whether the D334N mutation alters FBXO31 substrate recognition and CTAP clearance.

In vitro, neither wild-type nor D334N-mutant FBXO31 showed any affinity for the proposed C-terminal degron of cyclin D1 (Extended Data Fig. 9a). However, the D334N mutation did abolish binding to C-terminal amide peptides (Extended Data Fig. 9a,b). Likewise, FBXO31(D334N,ΔF-box) expressed in *FBXO31*-knockout cells did not immunoprecipitate both a model CTAP (mCherry-CONH$_2$) and unmodified cyclin D1 (Extended Data Fig. 9c). However, simple loss of function does not account for the dominant inheritance of FBXO31(D334N). We therefore hypothesized that this mutation might also lead to recognition of neosubstrates.

To identify cellular targets of FBXO31(D334N), we performed co-IP–MS analysis of FBXO31(ΔF-box) using both the wild-type and D334N mutant cDNAs. Mutant FBXO31(D334N,ΔF-box) formed detectable interactions with 220 proteins, 195 of which were not detected for the wild-type (Fig. 5a and Supplementary Table 7). We tested whether these putative neosubstrates are also removed from cells in response to acute FBXO31(D334N) expression using a ligand-inducible shield-degron system (DD)[37]. Expression proteomics identified a marked reduction in the abundance of many of the bound candidates within 12 h of stabilizing DD–FBXO31(D334N), but their abundance was unaffected when expressing wild-type FBXO31 (Fig. 5b, Extended Data Fig. 10a and Supplementary Table 8). Among these newly bound and degraded substrates were core-essential proteins, including SUGT1, a central regulator of mitotic spindle assembly[38,39]. In vitro reconstitution of SCF–FBXO31 ubiquitylation confirmed that the D334N mutation abolished its CTAP-directed activity (Fig. 5c) and, instead, resulted in robust ubiquitylation of the neosubstrate SUGT1.

To understand how FBXO31(D334N) selects neosubstrates, we searched for common sequence features among proteins identified by MS. Alignment of their C termini showed a marked enrichment of basic residues at position −3 (Lys/Arg) and hydrophobic residues (Φ) at position −1 from the C terminus (Extended Data Fig. 10b). Using a genetically

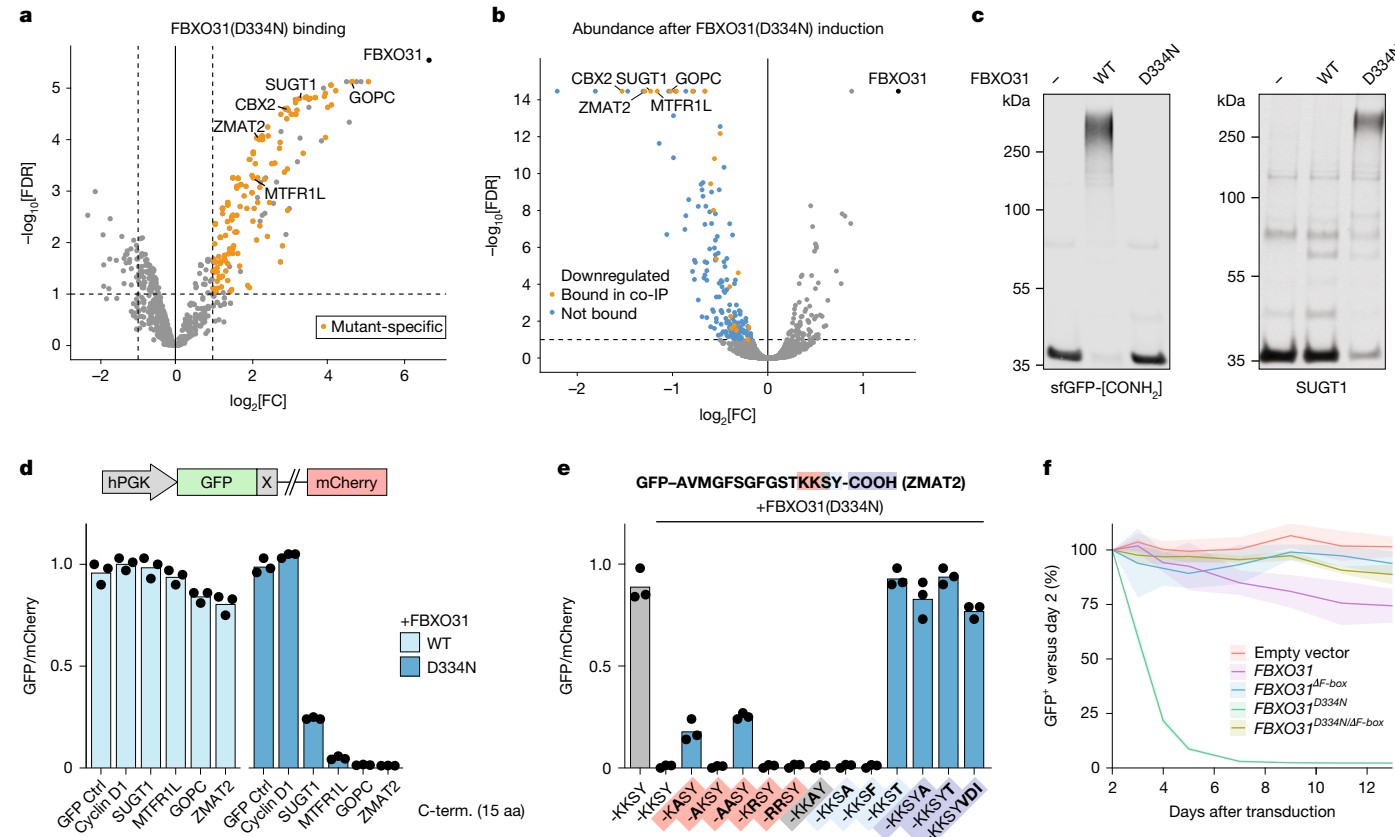

**Fig. 5 | The cerebral-palsy-associated mutation D334N shifts substrate selectivity of FBXO31. a**, Identification of FBXO31(D334N) neosubstrates by IP–MS analysis of HA-tagged FBXO31(ΔF-box,D334N) from *FBXO31*-knockout HEK293T cells. Proteins that do not co-IP with wild-type FBXO31 are highlighted in orange. **b**, Identification of proteome changes after acute induction of FBXO31(D334N) expression. *FBXO31*-knockout cells were stably transduced with ligand-inducible DD–FBXO31(D334N). The total protein abundance was measured by tandem-mass-tag MS after 12 h of protein induction (2 µM shield-1) and compared to shield-treated parental cells. Downregulated proteins (FDR < 0.1, log₂[fold change] < 0) are highlighted. **c**, In vitro ubiquitylation of model CTAP or SUGT1 by wild-type or mutant FBXO31. The blot is representative of two independent experiments. **d**, Protein stability assay for *FBXO31*-knockout

HEK293T cells expressing GFP with the indicated C termini (C-term.). Cells were transduced with vectors expressing wild-type FBXO31 or FBXO31(D334N) 2 days before measurement using flow cytometry. The *y* axis indicates the ratio of GFP fluorescence to the co-expressed control protein mCherry. aa, amino acids. **e**, Protein stability assay as in **d** for the ZMAT2 C terminus with the indicated mutations. **f**, Competitive proliferation assay measuring the impact of FBXO31 variant expression in *FBXO31*-KO HEK293T cells on cell fitness. The fraction of cells stably expressing GFP-linked cDNAs was measured over 12 days by flow cytometry. For **d** and **e**, data are the mean of three independent experiments (*n* = 3), each shown as black dots. For **f**, data are the mean ± s.d. of three independent experiments (*n* = 3). Gel source data are provided in Supplementary Fig. 1.

engineered dual-colour reporter of degron activity[40], we observed that the C-terminal sequences of each neosubstrate were ignored by wild-type FBXO31 but induced strong degradation by FBXO31(D334N) (Fig. 5d and Extended Data Fig. 10c,d). The C terminus of cyclin D1 was not a substrate of either wild-type or D334N FBXO31.

Using the reporter assay, we performed mutational analysis to test the role of the putative [Lys/Arg]-X-Φ-COOH motif. For the ZMAT2 C terminus, exchanging Lys−3 for Arg had no effect, while its mutation to Ala reduced degradation by FBXO31(D334N) (Fig. 5e and Extended Data Fig. 10e). Likewise, exchanging the terminal Tyr with hydrophobic amino acids (Phe or Ala) had no effect, yet exchange to hydrophilic Thr completely abolished D334N-mediated reporter degradation. Shifting the motif by addition of 1 or 3 amino acids to the C terminus also completely disrupted degron function. In the context of an intermediate FBXO31(D334N) degron, the SUGT1 C terminus, we observed a more stringent requirement for Lys and Tyr at the −3 and −1 positions (Extended Data Fig. 10f). Overall, we find that the [Lys/Arg]-X-Φ-COOH motif is necessary and sufficient for neosubstrate targeting by FBXO31(D334N). This well-defined recognition is in stark contrast to the broad targeting of amide-bearing C termini by wild-type FBXO31. We conclude that the D334N mutation does not merely modulate FBXO31 activity but also redirects it to an entirely new set of targets.

Degradation of SUGT1, ZMAT2 and other FBXO31(D334N) neosubstrates is expected to impair cell survival or proliferation. We therefore performed a competitive growth assay to quantify the impact of FBXO31(D334N) expression on cell fitness. While wild-type *FBXO31* cDNA expression was well tolerated in *FBXO31*-knockout HEK293 cells, the D334N mutant was rapidly depleted from co-culture (Fig. 5f). Deletion of the F-box motif required for SCF complex assembly fully abolished this effect, demonstrating that SCF–FBXO31(D334N) exerts a toxic ubiquitin ligase activity. On the basis of these results, we conclude that Asp334 is required for CTAP recognition, and the cerebral-palsy-associated mutation is dominant because it redirects ubiquitin ligase activity towards essential cellular proteins.

Overall, we identified C-terminal protein amides as a long overlooked chemical trigger of protein degradation. We identified their reader protein FBXO31 and describe how loss of amide selectivity causes severe collateral ubiquitylation by its cerebral-palsy-causing D334N variant.

## Discussion

To maintain protein homeostasis, cellular machinery must survey thousands of proteins for their chemical integrity. We propose that this can be achieved by modified amino acid degrons (MAADs) that

mark proteins for removal by reader proteins and downstream effectors. We identified a C-terminal amide as a bona fide MAAD found on oxidatively damaged proteins, marking them for degradation through SCF–FBXO31. Surveillance of oxidative protein damage by SCF–FBXO31 parallels the surveillance of oxidative genome damage by OGG1 and MUTYH[41]. Notably, FBXO31 is highly expressed in metabolically active tissues, including skeletal muscles and the brain[42]. By removing irreversibly damaged protein fragments, FBXO31 could prevent the accumulation of protein decay products in these tissues. Future studies on the impact of FBXO31 on diverse cell types throughout development and ageing in vivo could clarify how CTAP clearance shapes the human proteome.

α-Amidating fragmentation was introduced several decades ago as a potential mechanism of protein damage by ionizing radiation and oxidative stress[3]. The resulting C-terminal amide is an ideal degradation signal, as it is highly stable in solution and distinct from unmodified C termini in charge. Despite being detectable in public proteomics datasets, CTAPs remained largely overlooked, probably because such distributed fragmentation requires deep proteomic profiling and custom data analysis. We note that also the second cleavage product, an N-terminal keto-acid or aldehyde, could theoretically function as a MAAD. It remains possible that CTAPs are also formed by additional means, for example, by a to date unknown PAM-like lyase. Proteins that are amidated by PAM in the secretory pathway could also become FBXO31 clients if internalized or mislocalized to the cytoplasm.

An earlier study of stress-induced cyclin D1 downregulation suggested a role of FBXO31 in its destabilization[43]. Subsequent studies in other cell types and states could not confirm a universal role of FBXO31 in cyclin D1 regulation and, instead, implicated the ligase AMBRA1[44,45]. We also found no evidence for unmodified cyclin D1 degradation by either wild-type or D334N SCF–FBXO31. However, we speculate that the high doses of ionizing radiation and oxidizing agents used in earlier studies of FBXO31 might lead to fragmentation and CTAP formation.

The activity of sequence-based degrons can be modulated by PTMs that facilitate or block binding of ubiquitin ligases to defined amino acid motifs. Examples include the cell-cycle-regulated phospho degron of SIC1[46], or the hydroxyproline degron recognized by VHL[47]. These strictly sequence-dependent degrons typically have regulatory functions. By contrast, the C-terminal amide MAAD acts in a largely sequence-independent manner as an apparent quality-control signal. While substantial efforts have been made to find sequences targeted by the approximately 600 human ubiquitin ligases, it may be that some of these orphan ligases instead recognize MAADs. The human proteome is extensively modified, with at least 50% of peptides in MS scans carrying one of more than 700 catalogued modifications[9,10,48]. While we focused on effects of candidate MAADs on protein turnover, our semi-synthetic screening approach can be extended to chart the impact of chemical modifications on all steps in a protein's life cycle.

Few MAADs have been described to date, but they have considerable potential for therapeutic use. CRBN, a ubiquitin ligase often harnessed for targeted protein degradation, recognizes cyclized C-terminal imides[6]. Given that MAAD readers must recognize a well-defined chemical ligand while surveilling a broad range of putative clients, MAADs could provide entry points for developing degradation-inducing compounds. Harnessing native chemical recognition instead of repurposing sequence-based recognition could provide a promising alternative path to future targeted protein degradation drugs.

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

## Methods

### Peptide synthesis

Detailed synthetic methods and characterization of all peptides, pooled peptide libraries, custom peptide building blocks and protein–peptide conjugates generated for this study are provided in the Supplementary Information. Peptide sequences are provided in Supplementary Table 9.

### General method for the preparation of peptides for sortase reaction

Amino acids were loaded on chlorotrityl resin to access C-terminal carboxylic acids or Rink amide to access C-terminal amides (Supplementary Information). Automated peptide elongation was carried out on a Multisyntech Syro I parallel synthesizer according to the general peptide methods (Supplementary Information). The peptide was cleaved from the resin using TFA/DODT/$H_2O$ (95:2.5:2.5, v/v/v) for 1 h. The resin was removed by filtration and the filtrate concentrated under reduced pressure. The solution was triturated with diethyl ether and centrifuged to obtain crude peptide. The crude peptide was dissolved in $H_2O$/$CH_3N$ (1:1, v/v) + 0.1% (v/v) TFA and purified using preparative high-performance liquid chromatography. The peptide series, including pep_AARS1, pep_lib_VIN and pep_lib_YNR, was obtained from Craftide.

### Sortase-mediated protein conjugation

Peptides were dissolved in sortase reaction buffer (50 mM Tris, 150 mM NaCl, pH 7.4, at 4 °C). The pH was adjusted to 7–8. Peptide (final concentration 1 mM) was mixed with reporter protein (sfGFP, mCherry, BFP) (final concentration 75 μM). SortA 7 M was added (final concentration, 2 μM) and incubated at 4 °C for 18 h. Unreacted reporter protein and cleaved sortag were removed by Ni-NTA purification. The flow-through was collected and immediately buffer-exchanged to ion-exchange buffer (25 mM Tris, pH 8.5) using a desalting column (Cytiva). The sample was further purified by anion exchange using a MonoQ column (Cytiva) with a gradient of 0–25% high-salt buffer (25 mM Tris, 1 M NaCl, pH 8.5) in 25 CV. The fractions containing the product were pooled, buffer-exchanged to sortase reaction buffer supplemented with 0.5 mM TCEP and concentrated.

### General cell culture methods

HEK293 and HEK293T cells were cultured in DMEM containing GlutaMax (Thermo Fisher Scientific) and K562 cells were cultured in RPMI 1640 containing GlutaMax (Thermo Fisher Scientific) under standard conditions. Growth medium was further supplemented with 10% (v/v) fetal bovine serum (Thermo Fisher Scientific) and 100 U ml$^{-1}$ penicillin–streptomycin (Thermo Fisher Scientific). Stably transduced cells were selected using 2 μg ml$^{-1}$ puromycin (Thermo Fisher Scientific), 500 μg ml$^{-1}$ geneticin (Thermo Fisher Scientific) or 10 μg ml$^{-1}$ blasticidin (Thermo Fisher Scientific) starting 2 days after transduction.

Induction of doxycycline-dependent vectors was performed by addition of 500 ng ml$^{-1}$ doxycycline (Merck) every 48 h. Cells were treated with epoxomicin (Merck), folimycin A (concanamycin A, Merck), TAK243 (MedChem Express) or MLN4924 (MedChem Express), CSN5i-3 (MedChem Express), carfilzomib (MedChem Express), menadione (Merck) and auranofin (MedChem Express) or CB-5083 (MedChem Express) as indicated in the figure legends. $H_2O_2$ was freshly diluted in serum-free DMEM growth medium to 200 μM immediately before use and was kept protected from direct light. Cell lines were regularly tested for mycoplasma contamination using the MycoAlert Mycoplasma Detection Kit (Lonza).

All of the cell lines were obtained from certified vendors. Human cell lines were regularly tested for mycoplasma contamination using a commercially available kit (Lonza, MycoAlert). Throughout the duration of this study, no contamination was detected. CRISPR-competent K562 clones were reauthenticated by STR genotyping (Microsynth).

### Culture, editing and differentiation of NPCs

Human NPCs were grown from a previously established culture derived from the H9 human embryonic stem cell line[49]. Cells were expanded on plates coated with laminin (R&D Systems, 3446-005-01) in NPC base medium (50% Neurobasal Plus Medium (A3582901), 50% DMEM/F-12 (31331028), supplemented with 1× GlutaMax, 35050061, 1× N-2 supplement (17502048), 1× B-27 supplement without vitamin A (12587010) and 50 μM 2-mercaptoethanol (31350010); all Thermo Fisher Scientific). The medium was freshly supplemented with 10 ng ml$^{-1}$ humanKine bFGF (Merck, GF446-50UG) or 10 ng ml$^{-1}$ heat stable recombinant human FGF-basic (PeproTech, 100-18HS), 10 ng ml$^{-1}$ EGF (PeproTech, AF-100-15) and 20 ng ml$^{-1}$ BDNF (PeproTech, 450-02).

For neuronal differentiation, cells were seeded onto plates coated with laminin and polyornithine (Merck, P3655). Differentiation was induced by gradual withdrawal of growth factors (twofold dilution in growth-factor-free NPC medium on days 1, 3 and 5) followed by a serial twofold dilution of NPC base medium in neuronal maturation medium (Neurobasal plus medium, supplemented with GlutaMax, full B-27 supplement (Thermo Fisher Scientific, A3582801), MEM non-essential amino acids (Thermo Fisher Scientific, 11140050), antibiotic-antimycotic (Thermo Fisher Scientific, 15240062), dibutyryl cyclic AMP (StemCell Technologies, 100-0244), 20 ng ml$^{-1}$ BDNF and 20 ng ml$^{-1}$ GDNF (Peprotech, 450-10)) on days 7 and 11. From day 13 on, cells were grown in full neuronal maturation medium until collection.

CRISPRi-competent NPCs were generated by knock-in of a modified dCas9-KRAB-BFP cassette into the *CLYBL* locus according to a previously established strategy for stable CRISPRi in mature human neurons[50]. NPCs were electroporated using the Amaxa Nucleofector II and the Cell Line Nucleofector Kit V (Lonza) to deliver TALEN plasmids for *CLYBL* targeting[51] (pZT-C13-L1, pZT-C13-R1; Addgene, 62196 and 62197) and the homology donor pJC2528 (Supplementary Table 9). Cells were selected for correct integration using G418 (0.1 mg ml$^{-1}$, Thermo Fisher Scientific). Positive clones were pooled and expanded. To activate dCas9-KRAB expression, cells were electroporated with pCAG-Cre[52] (Addgene, 13775) and BFP-expressing cells were enriched by fluorescence-activated cell sorting at 72 h after electroporation.

### Virus production

Lentiviral vectors were packaged in HEK293T cells using standard methods. In brief, cells were incubated with plasmid DNA (transfer plasmid, pCMV-dR8.2 dvpr and pCMV-VSV-G at a weight ratio of 4:2:1) and polyethyleneimine (PEI, molecular mass of around 25,000 Da) at a ratio of 1:3 (w/w) of total plasmid DNA to PEI. The viral supernatant was collected at 48–72 h after nucleofection by ultrafiltration and supplemented with 4 μg ml$^{-1}$ polybrene. Gene knockdown was performed in a subclonal K562 cell line stably expressing dCas9-KRAB-BFP (Addgene, 102244), provided by E. J. Aird. For CRISPRi, sgRNAs targeting near transcription start sites were chosen based on previously optimized design rules[53] and cloned into the vector pCRISPRia. Puromycin was used to select for cells stably expressing sgRNAs. All custom vectors generated for this study will be made available through Addgene after publication of this Article. A complete list of plasmids used in this study is provided in Supplementary Table 9.

### Flow cytometry assays and cell sorting

Analytical flow cytometry was performed on the Attune NxT flow cytometer (Thermo Fisher Scientific). Single-cell isolation and sorting for CRISPR-screen analysis was performed on the Sony SH-800 cell sorter. Flow cytometry data were quantified and visualized using FlowJo 10.

For measuring the stability of recombinant or semi-synthetic proteins, 1.5–2.0 × 10$^5$ cells were electroporated with 200 pmol of protein in 20 μl cuvettes using a 4D nucleofector device (Lonza) according to

the manufacturer's recommendations. After protein delivery, cells were left to recover at 37 °C for 30–45 min and washed with PBS before measuring the initial mean fluorescence intensity values ($t_0$) using flow cytometry. Subsequent measurements were performed at the indicated timepoints. Baseline fluorescence values of mock-electroporated cells receiving no protein were measured in parallel and subtracted for each timepoint. Background-corrected fluorescence values were normalized to $t_0$.

Cell surface expression of CD55 was assessed by immunostaining and flow cytometry. In brief, cells were pelleted and resuspended in primary antibody diluted 1:200 in PBS containing 5% FBS. (APC-anti-human-CD55[JS11], BioLegend). After 20 min of incubation on ice, cells were washed three times with PBS followed by data acquisition. For competitive growth assays, cells were transduced with lentiviral vectors delivering FBXO31-IRES-GFP cDNA variants or an empty vector control (IRES-GFP) at a multiplicity of infection (MOI) of about 0.3. The fraction of transduced cells was measured every 2 days by flow cytometry and normalized to initial transduction levels (day 2) as a measure of relative fitness.

For mapping FBXO31(D334N) neosubstrate degrons, we cloned sequences encoding the C-terminal 15 amino acids of candidate substrates or mutants thereof into the C terminus of GFP in the previously established two-colour reporter vector Cilantro2[40] (Supplementary Table 9). After G418 selection and sorting of transduced cells, *FBXO31* cDNAs were delivered by transduction with lentiviral vectors encoding for FBXO31-IRES-mTagBFP or its D334N-mutant version (Supplementary Table 9). Fluorescence of the GFP reporter and mCherry control was measured by flow cytometry 2 days after transduction. Protein stability among mTagBFP-positive cells was measured as the ratio of median fluorescence values of GFP and mCherry (Supplementary Fig. 1d).

### Generation of clonal cell lines

For genome-wide CRISPR screening, K562 cells were made competent for inducible gene knockout (iCas9) by transduction with vectors SRPB (pHR-SFFV-rtTA3-PGK-Bsr) and 3GCasT (pHR-TRE3G-hSpCas 9-NLS-FLAG-2A-Thy1.1). Cas9-P2A-Thy1.1 expression was induced with doxycycline for 2 days and cells staining positive for Thy1.1 were isolated by single-cell sorting. Clonal lines were screened for cells that show no evidence of *CD55* knockout after viral delivery of sgCD55 (Supplementary Table 9) by antibody staining and flow cytometry, as well as efficient knockout after addition of doxycycline for 9 days.

*FBXO31*-knockout cells were generated by electroporation of cells with Cas9–sgRNA ribonucleoprotein particles as described previously[54] using the sgRNAs listed in Supplementary Table 9 based on previously optimized design rules[20,55]. In brief, in vitro transcription templates were generated by PCR using Q5 polymerase (New England Biolabs) with the primers listed in Supplementary Table 9 and used for in vitro transcription by T7 RNA polymerase (New England Biolabs). The resulting RNA was purified using a spin-column kit (RNeasy mini kit, QIAGEN) and 120 pmol of sgRNA was complexed with 100 pmol of recombinant SpCas9 protein at room temperature for 20 min. Cas9 protein was provided by the Genome Engineering and Measurement Lab (GEML) of the Functional Genomics Center Zurich (FGCZ of the University of Zurich and ETH). Assembled sgRNA–Cas9 complexes were delivered to cells using the 4D nucleofector kit (Lonza) according to the manufacturer's instructions. Clonal cell lines were isolated by single-cell sorting and characterized by genomic DNA extraction (Lucigen QuickExtract), PCR amplification of edited loci using Q5 polymerase (New England Biolabs) with genotyping and NGS primers (Supplementary Table 9). Pooled next-generation sequencing of edited loci was performed by GEML (FGCZ of the University of Zurich and ETH) on the MiSeq sequencer (Illumina) with 150 bp paired-end reads. Deep sequencing reads were analysed using CRISPResso2[56] (v.2.3.0) to identify compound heterozygous knockout clones.

### Western blotting and antibodies used in this study

Except for co-IP studies, immunoblotting was performed on whole-cell lysates in radioimmunoprecipitation buffer (RIPA, 50 mM Tris-HCl, 150 mM NaCl, 0.25% deoxycholic acid, 1% NP-40, 1 mM EDTA, pH 7.4) supplemented with protease inhibitors (Halt protease inhibitor cocktail, Thermo Fisher Scientific). Proteins were analysed by polyacrylamide gel electrophoresis (PAGE) and wet transfer onto nitrocellulose membranes (0.2 μm pore size) using standard methods. Membranes were blocked with TBS-T (150 mM NaCl, 20 mM Tris, 0.1% (w/v) Tween-20, pH 7.4) containing 5% skimmed milk powder and incubated with primary antibodies diluted in TBST containing 5% bovine serum albumin (BSA) and 0.05% (w/v) sodium azide. The primary western blotting antibodies used in this study were as follows: FBXO31 (Abcam, ab86137; Human Protein Atlas, HPA030150), GFP (Abcam, ab6556), mCherry (Abcam, ab183628), SKP1 (Cell Signaling Technology, 2156), CUL1 (Invitrogen, 71-8700), AARS1 (Fortis/Bethyl Life Science, A303-473A), GLUL (Fortis/Bethyl Life Science, A305-323A), cyclin D1 (Abcam, ab134175), HA (Cell Signaling Technology, 3724) and SUGT1 (Bethyl, A302-944A). Primary western blotting antibodies for mCherry and GFP were used at a dilution of 1:2,000. All of the other primary western blotting antibodies were used at a 1:1,000 dilution. Detection of primary antibodies was performed using far-red fluorescently labelled secondary antibodies (LI-COR Biosciences, 926-32213 and 926-68072) at a dilution of 1:15,000. The blots were scanned on the Odyssey CLx scanner (LI-COR Biosciences).

### CRISPR screening and analysis

iCas9 cells were transduced with the pooled lentiviral sgRNA library TKOv3[20] at an MOI of about 0.3 as measured by serial dilution, puromycin selection and viability assay (CellTiter-Glo 2.0, Promega). Two pools of $1.2 × 10^8$ cells each were transduced, yielding an approximately 500-fold library coverage, which was maintained throughout all of the cell culture steps. Cas9 expression was induced by addition of doxycycline for 5 days before delivery of reporter proteins. To isolate CTAP-clearance-deficient cells., ≥4 large-scale nucleofections were performed by combining $5 × 10^7$ cells, 2,000 pmol sfGFP-pep1-CONH$_2$ (sfGFP-GGGKDLEGKGGSAGSGSAGGSKYPYDVPDYAKS-[CONH$_2$]) and 2,000 pmol of a degron-tagged control protein (mTagBFP2–RXXG) in a 100 μl nucleofection reaction (4D nucleofector kit SE plus supplement SF1, Lonza)

Next, 14 h after nucleofection, cells were transferred on ice and sorted into a CTAP-clearance-deficient population (sfGFP$^+$ mTagBFP2$^-$) and a control population (sfGFP$^-$mTagBFP$^-$). Genomic DNA was extracted from snap-frozen sorted cells using the Gentra Pure kit (QIAGEN). sgRNA cassettes were isolated by two rounds of PCR using a previously published strategy[20] with NEBNext Ultra II Q5 Master Mix (New England Biolabs) and primer pairs listed in Supplementary Table 9.

Protospacers were quantified by deep sequencing using the NovaSeq SP 200 platform (Illumina) with 21 initial dark cycles by GEML (FGCZ of the University of Zurich and ETH) . sgRNA counts were retrieved using 'mageck count' (MAGeCK v.0.5.9.3) with the default parameters[57]. Raw sgRNA counts are provided in Supplementary Table 1. Enrichment of sgRNAs targeting the same gene in CTAP-clearance-deficient cells versus the control population was estimated with the 'mageck test' command using a paired design for screening duplicates with option --remove-zero both and otherwise the default parameters.

### Recombinant protein production in *Escherichia coli*

Fluorescent protein expression and purification: sortase-tagged variants of GFP, mCherry and mTagBFP2 were expressed from IPTG-inducible expression plasmids listed in Supplementary Table 9 in *E. coli* strain BL21(DE3). Expression was induced at an optical density at 600 nm of 0.6 using 1 mM IPTG followed by continued culture for 6 h at 37 °C and freezing of cell pellets. The samples were lysed by sonication

in buffer NBA (200 mM NaCl, 50 mM HEPES, 25 mM imidazole, 1 mM TCEP, pH 8.0) His-tagged protein was enriched from cleared lysates on the HisTrap FF column (Cytiva) by elution with 250 mM imidazole in sonication buffer. Crude HisTrap eluates were used as input for sortylation followed by further purification as described above.

Codon-optimized cDNAs of SKP1(A2P,Δ38–43,Δ71–82) and His$_6$-Smt3-StrepII-FBXO31.1(Δ1–65) were generated by gene synthesis (Integrated DNA Technologies) based on previously optimized expression constructs[58,59] and cloned into the bacterial expression vector pET28b (Supplementary Table 9). Expression was performed at 18 °C overnight in the presence of 0.5 mM IPTG in *E. coli* strain BL21(DE3) in Terrific Broth (Faust) in the presence of 50 μg ml⁻¹ kanamycin. After bacterial cell lysis by sonication in buffer NBA (200 mM NaCl, 50 mM HEPES, 25 mM imidazole, 1 mM TCEP, pH 8.0), recombinant protein was enriched on the HisTrap FF column (Cytiva) by elution with 250 mM imidazole in the same buffer. The His$_6$–Smt3 tag was removed by digestion with purified SUMO protease Ulp1 at 4 °C overnight. Binary SKP1–FBXO31 complexes were further purified by anion-exchange chromatography (HisTrap Q HP, Cytiva) in 50 mM HEPES (pH 7.5) and 5 mM DTT on a linear gradient of 100–1,000 mM NaCl followed by gel filtration (HiPrep Sephacryl S-100 HR, Cytiva) in 200 mM NaCl, 50 mM HEPES and 1 mM TCEP. Sortase (SortA-7M) and Ulp1 were prepared by recombinant expression *E. coli* as previously reported[60,61].

## Protein expression and purification in insect cells

Human cDNA sequences of tRNA-LC subunits were subcloned into modified pLIB vectors. HSPC117 was N-terminally 8×His-tagged and FAM98A(1–340) was modified with N-terminal 2×StrepII-tag. DDX1 and RTRAF (CGI-99) were subcloned without a tag and in full length. The resulting gene cassettes were combined into a single multigene construct using the bigBAC system[62]. The multigene construct was integrated into a baculovirus genome, transfected into SF9 cells for virus production and finally used for infection of large-scale protein production cultures as described previously[63]. SF9 cells were grown in SF-4 Baculo Express ICM ready to use medium (Bioconcept, 9-00F38-K). Initial transfection was performed using the TransIT-Insect Transfection Reagent (Mirus, Cat. MIR 6104). Then, 50 ml of infected preculture was used to infect 500–750 ml SF9 cultures at 1.5 million cells per ml or multiples thereof. Infected large-scale cultures were grown for 72 h at 27 °C and 90 rpm.

Large-scale cultures were pelleted by centrifugation, washed in tRNA-LC lysis buffer (500 mM KCl, 20 mM HEPES-KOH, 30 mM imidazole, pH 7.8). Flash-frozen cell pellets were stored at −80 °C. Cells were lysed in tRNA-LC lysis buffer by sonication (Bandelin Sonopuls HD 3200, VS 70T probe). Cleared lysate was incubated with equilibrated HIS-Select nickel affinity resin (Merck, P6611) at 4 °C. The resin was washed extensively with tRNA-LC lysis buffer on a gravity-flow column and captured protein was eluted in Ni-elution buffer (150 mM KCl, 20 mM HEPES-KOH, 250 mM Imidazole, pH 7.8). A second affinity purification was performed with equilibrated Streptactin Sepharose high performance resin (Cytiva, 28-9355-99) by binding for 1 h at 4 °C on a rolling table, washing on a gravity flow column with tRNA-LC wash buffer (150 mM KCl, 20 mM HEPES-KOH, pH 7.8) and eluted in Strep-elution buffer (150 mM KCl, 20 mM HEPES-KOH, 5 mM desthiobiotin, pH 7.8). The quality and efficiency of all protein purification steps was monitored using SDS–PAGE.

## Pooled library IP and TIMS-TOF analysis

For each IP, 1 pmol of peptide library was incubated with an equimolar amount of recombinant Strep-II-tagged FBXO31–SKP1 complex in 1 ml of binding buffer (200 mM NaCl, 25 mM HEPES pH 7.5, 1 mM TCEP) with pre-equilibrated magnetic Strep-Tactin XT beads (240 μl MagStrep type 3 XT bead slurry, IBA life sciences). After 1 h of binding at room temperature on a rotator, the beads were washed twice with binding buffer, twice with wash buffer (150 mM NaCl, 25 mM HEPES, 1 mM TCEP) and eluted in 150 mM NaCl, 50 mM HEPES, 100 mM biotin (pH 7.7). Eluted peptides

and an equal amount of input library (1 pmol) were diluted 1:1 with triethylammonium bicarbonate (50 mM, pH 8.0). Peptides were isobarically labelled with TMT 2-plex (Thermo Fisher Scientific) labels according to the manufacturer's instructions. The input library was modified with TMT-126 and eluted peptides with TMT-127. After labelling, peptides were combined 1:1. Combined peptides were desalted using 100 μl C18 ZipTips and dried (Savant SpeedVac). The dried peptides were resuspended in 5 vol% acetonitrile, 0.1 vol% formic acid.

Tandem MS experiments were performed using the ESI-TIMS-QTOF-MS system (TimsTOF Pro, Bruker Daltonics) with collision-induced dissociation and N$_2$ as the collision gas. Peptides were pressure loaded onto a reversed-phase 25 cm × 75 μm inner diameter C18 1.6 μm column (Ionoptics) with a reversed-phase 5 mm × 0.3 mm inner diameter C18 5 μm column (Thermo Fisher Scientific) as a guard column at 40 °C. The mobile phase consisted of water with 0.1% formic acid (A) and acetonitrile with 0.1% formic acid (B). The gradient started at 2% of B and was linearly increased to 35% B in 120 min at flow rate of 300 μl min⁻¹. A second gradient profile, started at 35% of B and was linearly increased to 95% B in 2 min at flow rate of 300 μl min⁻¹. This was followed by isocratic conditions of 95% B at flow rate of 300 μl min⁻¹ for 8 min. The total run time, including the conditioning of the column to the initial conditions, was 163 min. Further data processing was performed using Data Analysis 5.3 software (Bruker Daltonics) using a processing script to generate export files and reports.

Mascot Server v.2.8.1. (Matrix Science) was used to match spectra against a custom reference of common contaminants concatenated with all potential peptides generated by pooled SPPS (10 ppm peptide mass tolerance, 0.05 Da fragment mass tolerance and, for peptide–amide libraries, −0.98 Da at peptide termini as a variable modification). For each peptide, reporter ion intensities were summed across multiple detections and scaled to the sum of reporter ion intensities for all input library peptides yielding relative reporter ion intensities for cross-comparisons between runs. For each experiment, two parallel IPs were analysed. For wild-type FBXO31 co-IP, two MS measurements were run for each IP for improved library coverage.

## In vitro ubiquitylation assay

2 μM of recombinant FBXO31–SKP1 complexes was combined with 0.5 μM of substrate, 50 μM recombinant ubiquitin, 50 nM UBE1, 500 nM UBE2R1, 500 nM UBE2D3 and 500 nM CUL1–NEDD8–RBX1 complex in 50 mM Tris/HCl (pH 7.5), 100 mM NaCl, 10 mM MgCl$_2$, 10 mM ATP and 0.5 mM TCEP. The reactions were incubated at 30 °C for 1 h unless indicated otherwise and analysed by immunoblotting. All protein components except for FBXO31–SKP1 and model substrates were sourced commercially (R&D Systems).

## HA–FBXO31 co-IP

Co-IP of semi-synthetic model substrates (mCherry-GGGRRLEGKE EDEKGSRASDDFRDLR-[COOH/CONH$_2$]) was performed in *FBXO31*-knockout HEK293 cells stably transduced with pLenti-EF1A-HA-FBXO31-PGK-Neo or the indicated mutant derivatives (Supplementary Table 9). Cells were nucleofected 2 h before collection and treated with 2 μM MLN4924 (MedChemExpress) and 500 nM epoxomicin (Merck) or 1 μM carfilzomib (MedChemExpress) to block protein degradation and stabilize CRL complexes.

Co-IP of endogenous FBXO31 clients and model CTAPs was performed in *FBXO31*-knockout HEK293 or HEK293T cells stably transduced with the lentiviral vectors expressing cDNAs of HA-tagged FBXO31 or mutant derivatives thereof as noted in figure legends (Supplementary Table 9). Cells were cultured in serum-free DMEM supplemented with 500 nM epoxomicin with or without addition of H$_2$O$_2$ at a final concentration of 200 μM for 1 h before cell collection.

For both endogenous and synthetic client IP, cells were collected by centrifugation and washed in ice-cold PBS followed by lysis in SCF-IP buffer (150 mM NaCl, 50 mM Tris-HCl, 1 mM EDTA, 0.1% Igepal CA-630,

5% (v/v) glycerol, pH 7.5) supplemented with Halt protease inhibitor cocktail (Thermo Fisher Scientific) by rotation for 30 min at 4 °C. Debris was removed by centrifugation (4 °C, 30 min, 20,000*g*) and the total protein concentrations were measured using the Bradford protein assay (Thermo Fisher Scientific). Typically, 140 µg of protein was incubated with 12 µl of Pierce anti-HA magnetic bead resin (Thermo Fisher Scientific) on a rotator wheel at 4 °C overnight. The beads were washed three times for 5 min in SCF-IP buffer and bound proteins were eluted twice using 0.1 M glycine (pH 2), followed by addition of 0.25 volumes neutralization buffer (1.5 M NaCl, 0.5 M Tris-HCl, pH 8.0).

### Co-IP MS of HA-tagged FBXO31

IP–MS of HA-tagged FBXO31 was performed with *FBXO31*-knockout HEK293T cells stably expressing HA–FBXO31(ΔF-box) from vector pLenti-EF1A-HA-FBXO31(ΔF-box)-IRES-GFP or an indicated mutant cDNA. For each condition, three replicate samples were processed from separately passaged cultures. As a negative control, we used identical *FBXO31*-knockout HEK293T cells not expressing bait cDNAs. For IP analysis of oxidatively stressed cells, cultures were incubated in serum-free DMEM supplemented with 500 nM epoxomicin with or without addition of H$_2$O$_2$ at a final concentration of 200 µM for 1 h before to cell collection.

Cells were trypsinized and washed twice with ice-cold PBS containing 500 nM epoxomicin and resuspended in lysis buffer (300 mM NaCl, 50 mM Tris-HCl, 0.5% (v/v) Igepal CA-630) freshly supplemented with Halt Protease Inhibitor Cocktail (Thermo Fisher Scientific). Cell lysates were obtained by rotation for 30 min at 4 °C and were cleared by centrifugation (4 °C, 30 min, 20,000*g*). For IP, 1,000 µg of cellular protein extract was incubated with 80 µl of pre-equilibrated bead slurry (Pierce anti-HA magnetic beads, Thermo Fisher Scientific) in a total volume of 250 µl of lysis buffer on a rotator wheel at 4 °C for 4 h. Magnetic beads were washed four times with lysis buffer and twice with PBS followed by elution with 80 µl 0.1 M glycine (pH 2.0) at room temperature with gentle shaking for 10 min. The supernatant was neutralized with 0.25 volumes of 1.5 M NaCl, 0.5 M Tris-HCl pH 7.5.

Eluates were processed by the proteomics group of FGCZ (University of Zurich and ETH) by in-solution tryptic digest and LC–MS. In brief, LC–MS analysis was performed on the Orbitrap Exploris 480 Mass Spectrometer (Thermo Fisher Scientific) coupled by ESI to the ACQUITY UPLC M-Class System (Waters). Data-dependent acquisition was performed over a 65 min runtime with MS1 precursor mass scans at 120,000 resolution across 350–1,200 *m/z*. Precursors were isolated with a 1.2 *m/z* window followed by HCD fragmentation at 30% collision energy. MS2 scans were performed at 30,000 resolution across 350–1,200 *m/z*.

Spectra were searched against the human proteome (UniProt: UP000005640) using the MSFragger engine[64,65] (v.4.1) through Frag-Pipe (v.22.0) with the default parameters for closed search of fully tryptic peptides and sequential FDR filtering of peptide spectrum matches (PSMs), peptides and proteins by philosopher[66] with the default parameters (protein FDR ≤ 0.01). Common contaminants[67] were removed and enrichment of interactors was calculated using limma v.3.58.1, by fitting a linear model to normalized spectral counts for each sample plus one pseudocount followed by empirical bayes moderation and Benjamini–Hochberg correction of *P* values for multiple-hypothesis testing. Detailed search parameters and acquisition settings are deposited alongside the raw data at Pride (https://www.ebi.ac.uk/pride/).

### IP–MS of Flag-tagged AARS1

IP–MS analysis of 3×Flag-tagged AARS1 was performed on *FBXO31*-knockout HEK293T cells transiently transfected with pCDNA4TO-CMV-3xFLAG-Halo-AARS1 using 30 µg of PEI (molecular mass, ~25,000) and 10 µg of plasmid DNA. Two days after transfection, cells were challenged with 200 µM H$_2$O$_2$ in serum-free DMEM. Then, 5 min after H$_2$O$_2$ treatment, cells were additionally supplemented with 500 nM epoxomicin. Cells were washed three times in PBS supplemented with 500 nM epoxomicin and snap-frozen. Cell pellets were lysed by resuspension

in RIPA buffer (50 mM Tris-HCl, 150 mM NaCl, 0.25% deoxycholic acid, 1% NP-40, 1 mM EDTA, pH 7.4) supplemented with protease inhibitors (Halt protease inhibitor cocktail, Thermo Fisher Scientific). Then, 250 µg of protein was incubated with 25 µl of pre-equilibrated anti-Flag resin (M2 anti-FLAG beads, Merck) on a rotator at 4 °C overnight. The beads were washed once with RIPA buffer, four times with high-salt RIPA buffer (50 mM Tris-HCl, 500 mM NaCl, 0.25% deoxycholic acid, 1% NP-40, 1 mM EDTA, pH 7.4) and twice with TBS (150 mM NaCl, 50 mM Tris pH 7.5) followed by elution in 100 µl TBS + 150 ng µl$^{-1}$ competitor peptide (Pierce 3× DYKDDDDK Peptide, Thermo Fisher Scientific) at 37 °C for 30 min. The eluates were processed by the proteomics group of FGCZ (Uni Zürich/ETH) by in-solution digest with Glu-C protease and LC–MS/MS in data-dependent acquisition mode on the Orbitrap Exploris 480 Mass Spectrometer as described above for IP–MS analysis of FBXO31.

Spectra were searched against the human proteome (UniProt: UP000005640) using the MSFragger engine[64,65] (v.4.1) through Frag-Pipe (v.22.0) with the default parameters for semi-tryptic search unless stated otherwise. In brief, peptide identification was based on *b*- and *y*-series ions allowing for up to 1 missed cleavage and up to 1 non-enzymatic peptide terminus, variable peptide C-terminal amidation (−0.984016 Da), M oxidation (+15.99490 Da), protein C-terminal acetylation (+42.01060 Da) and fixed C carbamidomethylation (+57.021464 Da). Sequential filtering for high-confidence PSMs, peptides and proteins was performed using philosopher[66] with the default parameters (protein FDR ≤ 0.01). For identification of amide-bearing de novo C termini, internal AARS1-matching peptides were filtered for modified C termini not matching a Glu-C cleavage site (D or E). Detailed search parameters are deposited alongside the raw data at Pride (https://www.ebi.ac.uk/pride/).

### Expression proteomics by tandem mass tag MS

For inducible expression of DD-3×Flag-FBXO31, *FBXO31*-knockout HEK293 cells were transduced with pLenti-EF1A-DD-3×Flag-FBXO31-PGK-Neo or its D334N mutated variant (Supplementary Table 9). Cells were treated with 2 µM Shield-1 (MedChem Express) for 12 h before collection. All of the experiments were performed with three replicate samples treated and collected on separate days. Cell pellets were processed for whole-proteome quantification by FGCZ (Univesity of Zurich and ETH). In brief, cell pellets were lysed in 4% SDS in 100 mM Tris/HCl pH 8.2 by boiling and mechanical homogenization. Per sample, 50 µg of total protein was reduced (2 mM TCEP) and alkylated (15 mM 2-chloroacetamide) and further processed using a fully automated SP3 purification, digest and clean-up workflow[68]. The samples were isobarically labelled with TMT10plex (Thermo Fisher Scientific), pre-fractionated by RP-HPLC on an XBridge Peptide BEH C18 column (Waters) and pooled into eight fractions.

In brief, LC–MS analysis was performed on the Orbitrap Exploris 480 Mass Spectrometer (Thermo Fisher Scientific) coupled by ESI to the ACQUITY UPLC M-Class System (Waters). Data-dependent acquisition was performed over 110 min runtime with MS1 precursor mass scans at 120,000 resolution across 350–1,400 *m/z*. Precursors were isolated with a 1.2 *m/z* window followed by HCD fragmentation at 30% collision energy. MS2 scans were performed at 45,000 resolution across 350–1,400 *m/z*.

Raw MS data were processed using ProteomeDiscoverer 2.4 (Thermo Fisher Scientific). Spectra were searched against the UniProt human reference proteome (downloaded on 29 April 2022) and common contaminants with Sequest HT with a peptide-level FDR cut-off of 0.01. Detection of differentially abundant proteins was performed on reporter ion intensities for D334N or wild-type FBXO31-expressing cells compared with Shield-1-treated control cells lacking *FBXO31* cDNA.

### Re-analysis of public proteomes for CTAP detection

Label-free DDA-MS-based proteomics of tryptic digests of healthy human tissues from single donors were downloaded from PRIDE archive PXD010154, samples 1277 (brain), 1499 (adipose tissue), 1306 (thyroid)

and 1296 (heart)[69]. Peptide search was performed exactly as described above for Flag–AARS1 IP–MS. For downstream analysis, fully enzymatic peptides were defined as having termini that match trypsin cleavage sites or annotated N and C termini, accounting for clipping of N-terminal Met. Semi-enzymatic peptides were defined as having one cleavage that does not match an annotated protein terminus or trypsin cleavage site. Full search parameters and PSM-level search results used for quantification of CTAP formation are deposited at Pride (https://www.ebi.ac.uk/pride/).

## In vitro protein fragmentation monitoring

Purified human haemoglobin was obtained from Abcam (ab77858) and dissolved in DPBS (Thermo Fisher Scientific 14190144). In vitro fragmentation reactions were performed at a final concentration of $1 \text{ g l}^{-1}$ haemoglobin and indicated concentrations of $H_2O_2$ by incubation at 37 °C for 1 h. Purified human tRNA-LC was diluted in PBS to a final reaction volume of $0.25 \text{ g l}^{-1}$ and incubated with the indicated concentrations of $H_2O_2$ and $CuCl_2$ for 30 min at room temperature for in vitro fragmentation. All of the reactions were stopped by flash-freezing in liquid nitrogen. Thawed reactions were immediately precipitated and processed for tandem MS by the FGCZ (University of Zurich and ETH). In brief, proteins were precipitated by addition of TCA (Merck) to a final concentration of 5%, washed twice with ice-cold acetone, air-dried and resuspended in aqueous buffer containing 10 mM Tris (pH 8.2) and 20 mM $CaCl_2$. The samples were reduced and alkylated with 5 mM TCEP and 15 mM 2-chloroacetamide (30 min, 30 °C) followed by enzymatic digest with the indicated enzymes (trypsin, ArgC, trypsin-N or chymotrypsin). LC coupled to MS in data-dependent acquisition mode was performed as described above for FBXO31 IP–MS experiments.

Raw MS/MS files were searched for semi-enzymatic peptides using the MSFragger engine[64,65] (v.4.1) through FragPipe (v.22.0) with the default parameters for the respective proteases additionally allowing for peptide-C-terminal amidation (−0.9840 Da). Search was performed against the respective purified proteins and common contaminants with decoys. Sequential filtering was performed with Philosopher for protein FDR < 0.01 (v.5.1.1). In addition to initial filtering, only PSMs with a probability of >0.01 were used for plotting. Complete search parameters are deposited alongside raw data files and search results at the PRIDE archive (https://www.ebi.ac.uk/pride/).

As a control for technical noise, in vitro fragmented haemoglobin was reanalysed also allowing multiples of 0.984 as C-terminal mass shifts (from −2.952 to +2.952).

## Active CRL profiling

Treatments were performed in four separate cultures as individual replicates for each tested condition. For active CRL profiling, a subclonal line of K562 cells was pretreated with 1 µM carfilzomib (MedChem Express, HY-10455) and 5 µM p97/VCP inhibitor CB-5083 (MedChem Express, HY-12861) for 5 min. Cells were either left untreated or treated with 200 µM freshly diluted $H_2O_2$ for 2 h. To preserve the active CRL repertoire during cell collection, cells were treated with an N8-block protocol[70]. Specifically, cells were exposed to 1 µM MLN4924 (MedChemExpress, HY-70062) and 1 µM CSN5i-3 (MedChemExpress, HY-112134) in PBS for 3 min. After treatment, cells were washed with PBS (Gibco, 14190094) and snap-frozen in pellets of around $5 \times 10^6$ cells each. Thawed pellets were lysed in 380 µl of lysis buffer (25 mM HEPES pH 7.4, 5% glycerol, 150 mM NaCl, 0.5% NP-40, 1× HALT protease/phosphatase inhibitor (Thermo Fisher Scientific, 78440), 2 µM MLN4924, 2 µM CSN5i-3). Lysates were clarified by centrifugation at about 20,000g for 3 min at 4 °C and filtered through 0.22 µm spin filters (Corning, 8161). An active Cullin-binding Fab[31] was immobilized on high-capacity magnetic Streptavidin beads (Promega, V7820) according to the manufacturer's instructions. The equivalent of 10 µl bead slurry of the Fab-coated beads was then added to the cleared lysates and incubated for 45 min at 4 °C with gentle rotation. Beads were washed twice with lysis buffer, twice with wash buffer (lysis buffer without

IGEPAL CA-63) and twice with HBS (25 mM HEPES pH 7.4 and 150 mM NaCl). Elution was performed with 50 µl 0.1% TFA in water. The elution was then heated at 98 °C for 5 min. Subsequently, 2-chloroacetamide and TCEP were added to final concentrations of 40 mM and 10 mM, respectively, and the samples were incubated at 45 °C for 5 min. Digestion was carried out overnight at 37 °C with agitation (1,200 rpm) using a 1:100 w/w ratio of trypsin (Sigma-Aldrich, T6567) and LysC (FUJIFILM Wako, 125-05061). The samples were quenched by addition of TFA to a final concentration of 1% and 200 ng of peptides were loaded Evotips Pure (EvoSep, EV2011) according to the manufacturer's instructions.

The samples were measured using the Evosep One LC system (EvoSep, EV-1000) coupled to a TimsTOF Pro 2 mass spectrometer (Bruker Daltonics). The 30 samples per day method was used with a 15 cm × 150 µm column packed with 1.9 µm $C_{18}$-beads (Bruker Daltonics, 1893471), maintained at 50 °C and coupled to a 10 µM fused silica ID emitter (Bruker Daltonics, 1865691) within a CaptiveSpray ion source (Bruker Daltonics). Mobile phases were composed of 0.1% formic acid in water (buffer A) and 99.9% acetonitrile/0.1% formic acid (buffer B). For data acquisition, a dia-PASEF method with 20 dia-PASEF scans separated into two ion mobility windows per scan, covering an $m/z$ range from 350 to 1,200 was used. Variable window widths were determined using py_diaid[71], and the ion mobility range was set between 0.7 to 1.3 V s cm$^{-2}$. The accumulation and ramp times were both set to 100 ms. Collision energy was ramped linearly from 20 eV at $1/K_0$ = 0.6 V s cm$^{-2}$ to 59 eV at $1/K_0$ = 1.6 V s cm$^{-2}$.

DIA raw files were searched using library-free search in DIA-NN[72] (v.1.8.1, permitting one missed cleavage, one variable modification (N-terminal acetylation and methionine oxidation), cysteine carbamidomethylation as a fixed modification, with MBR and 'deep-learning-based spectra, RT and IM prediction' enabled). Data were analysed using the Perseus software package[73] (v.1.6.7.0). Protein intensities were log$_2$-transformed, and the datasets were filtered to contain no missing value in at least one experimental condition for all of the protein groups. Missing values were imputed using a normal distribution with a width of 0.3 and a downshift of 1.8.

## Visualization of protein structures and alignments

All protein structures were downloaded from the PDB under the accession numbers noted in respective figure legends. Structures and electrostatic potential maps were rendered using ChimeraX-1.5[74]. The composite structure of SCF–FBXO31 was generated from substructures 5VZU (FBXO31) and 6TTU (NEDD8–CUL1–RBX1–SKP1). Structures were aligned by their shared SKP1 substructure in using MatchMaker with the default settings. Alignment and conservation analysis of select FBXO31 orthologues and of FBXO31(D334N) neosubstrates was performed using JalView[75] (v.2.11).

## FP assays and analysis

Increasing concentrations of FBXO31–SKP1 complex (0–5 µM) were incubated with fluorescein-labelled peptides (2–10 nM) in 25 mM HEPES, 150 mM NaCl, pH 7.5 and incubated at room temperature for 30 min. FP was measured on a Victor Nivo Plate Reader (Perkin Elmer). After subtraction of the baseline signal, the polarization levels were normalized to the highest signal of each pair of amidated and non-amidated peptides for wild-type FBXO31. Binding curves were fitted by least-squares regression and half-maximal binding concentrations were extracted using Prism 9 (Graphpad) assuming one-site binding, maximal binding at 100% of the measured signal and no contribution from unspecific interactions. Dissociation constants for the highest affinity probes reported in this study are close to the probe concentrations used for FP assays and are therefore reported only as approximate values.

## RNA-seq and differential gene expression analysis

Total RNA was collected from CRISPRi competent HEK293T cells, NPCs and NPC-derived neurons by column purification (RNeasy Mini, QIAGEN). Poly-A-enriched full-length mRNA-sequencing libraries

were generated and sequenced by Novogene (RSPR00102) using the NovaSeq X Plus series platform for 150 bp paired-end sequencing. Raw sequencing reads were aligned to a custom human genome reference based on GrCH38 without alternative chromosomes and including commonly used cDNAs. Reads were aligned with STAR-aligner[76] (v.2.7.10a) and counting was performed using FeatureCounts[77] (v.2.0.6) against a custom transcriptome reference. Differential gene expression analysis was performed using DESeq2[78] (v.1.42.1) with the default parameters. Differentially expressed genes were searched against a database of published experimental signatures with rummagene[79]. Transcriptional changes in in vitro differentiated somatic motor neurons carrying amyotrophic-lateral-sclerosis-associated mutations were plotted based on previously published differential gene expression results[34].

### Reporting summary

Further information on research design is available in the Nature Portfolio Reporting Summary linked to this article.

## Data availability

All large-scale datasets displayed in this paper are provided in the Supplementary Information. Uncropped images of all gels and blots are provided in Supplementary Fig. 1. Unprocessed data for proteomics and RNA-seq experiments are accessible through the following public repositories. Raw proteomics data are deposited alongside primary search results and complete search parameters at Pride under accession numbers PXD055535 (FBXO31, IP–MS), PXD055814 (all CTAP identifications), PXD055518 (CRL profiling) and PXD055818 (TMT-based proteome quantification). Raw sequencing reads for RNA-seq samples have been deposited at the SRA under BioProject number PRJNA1173967. Deep sequencing read counts for CRISPR screen analysis are provided in Supplementary Table 9. Public proteomics data reanalysed in this study were obtained from Pride under accession number PXD010154, sample numbers 1277, 1499, 1306 and 1296. Public protein structures were downloaded from the PDB (5VZU, 6TTU, 6DO3, 6LEY and 7Y3A). RNA-seq reads were aligned against the human genome reference GRCh38. Proteomics data were searched against the UniProt human proteome reference UP000005640. Source data are provided with this paper.

## Code availability

There is no custom code central to this paper.

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

**Acknowledgements** We thank E. J. Aird, M. Banović, J. Fielden, M. E. Karasu and J. R. Liang and all of the other members of the Corn and Bode laboratories for discussions. We thank A. Gvozdenovic for kind support with proofreading. A. Dittmann, T. Kockmann, L. Kunz, C.-w. Lin, P. Nanni and S. Pfammatter of the Functional Genomics Center Zürich (FGCZ, ETH/University of Zürich) supported this work through proteomics sample processing, acquisition and data analysis. L. Bertschi, M. Meier and B. Rubi of the Molecular and Biomolecular Analysis Service (MoBiAS) in the Department of Chemistry and Applied Bioscience (ETHZ) provided HRMS and assisted with multiplex peptide binding screens. S. Kreutzer and Z. Kontarakis of the Genome Engineering and Measurement Lab (GEML, ETH/University of Zurich) provided recombinant SpCas9 and a viral sgRNA library and performed deep sequencing. We thank J. Greenwald for discussions and support regarding protein expression. This work was supported by an ETH Zurich Postdoctoral Fellowship program and an EU Horizon 2020, Marie Skłodowska-Curie grant 898175 (M.F.M.), by the Scholarship Fund of the Swiss Chemical Industry (R.H.), by the NOMIS Foundation, by the Lotte and Adolf Hotz-Sprenger Stiftung, by the European Research Council under the European Union's Horizon 2020 Research and Innovation Program (grant agreement no. 855741, DDREAMM) (J.E.C.) and by the Max Planck Society (M.M. and B.A.S.).

**Author contributions** The study was conceptualized by M.F.M., J.F., R.H., J.W.B. and J.E.C.; M.F.M. performed the CRISPR screen. All cellular characterization of FBXO31 and its mutants was performed by M.F.M. with M.C., M.P.A.-B. and N.D.S.; J.F., R.H. and F.K. prepared peptides and GFP reporter conjugates. M.F.M. and J.F. performed in vitro characterization of FBXO31 including FP binding assays and pooled-peptide library analysis. M.F.M. performed IP–MS profiling. M.F.M., J.F., L.T.H. and V.B. performed CRL profiling. tRNA ligase complex was prepared by M.M.P. and A.S.N. RNA-seq experiments were performed by M.F.M. and A.D.-L. Reanalysis of public proteomes was performed by M.F.M. The manuscript was written by M.F.M., J.F., J.W.B. and J.E.C. with review and editing by M.F.M., J.F., M.C., R.H., L.T.H., M.M.P., A.D.-L., F.K., M.P.A.-B., N.D.S., B.A.S., J.W.B. and J.E.C. The study was supervised by M.M., S.J., M.J., B.A.S., J.W.B. and J.E.C.

**Funding** Open access funding provided by Swiss Federal Institute of Technology Zurich.

**Competing interests** M.F.M., J.F., R.H., J.W.B. and J.E.C. have filed patent applications related to MAAD discovery and mechanisms of FBXO31 substrate recognition performed in this study (WO/2024/115740, WO/2024/115746; both pending). J.W.B. and J.E.C. are founders of and M.F.M. and J.F. provide consultancy to Serac Biosciences. J.E.C. serves on the scientific advisory board of Mission Therapeutics. B.A.S. serves on the scientific advisory boards of Biotheryx and Proxygen. M.M. is an indirect investor of the Evosep company. The other authors declare no competing interests.

**Additional information**
**Correspondence and requests for materials** should be addressed to Jeffrey W. Bode or Jacob E. Corn.

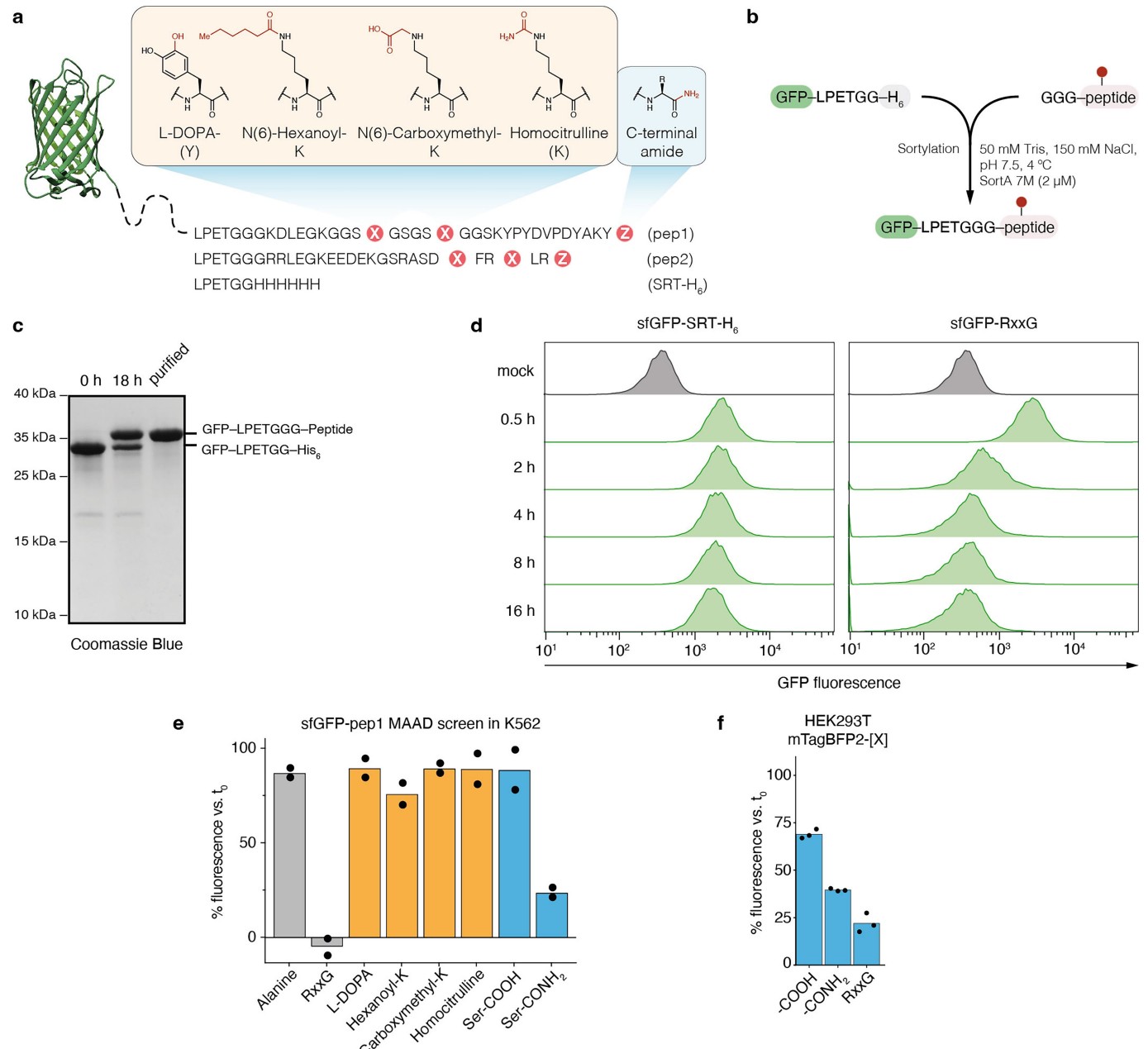

**Extended Data Fig. 1 | A screen for modified amino acid degrons (MAADs).**
**a**, Schematic showing sfGFP bearing various PTMs tested for MAAD activity.
Modified sequences are shown, and modified positions are highlighted. The
modification on each amino acid is shown in red. Corresponding unmodified
amino acids are indicated in brackets. **b**, Schematic showing modification of
sfGFP with a C-terminal sortylation tag. "SortA 7 M" conjugates sfGFP and
peptide containing an N-terminal GGG-motif. **c**, Exemplary SDS-PAGE analysis
of sortase reaction showing sfGFP, crude reaction mixture and purified sfGFP-
conjugate. Conjugate was purified by Ni-NTA to remove unreacted sfGFP,
cleaved sortag and SortA 7 M, followed by ion exchange chromatography. The
blot is representative of four parallel reactions. Similar results were obtained
for all sortylation reactions throughout this study. **d**, Exemplary time-course

experiment measuring sfGFP turnover in K562 cells. Cells received sfGFP
carrying a C-terminal sortase tag (sfGFP-SGGLPETGGHHHHHHV) or its
conjugated form carrying an RxxG degron motif (sfGFP-SGGLPETGGGRRL
EGKEEDEKGSRASDRFRGLR). **e**, Results of a screen for MAAD activity. Pep1
MAAD reporters shown in **a** were used in an in-cell reporter assay as in **d** in K562
cells. Bars represent means of two independent experiments ($n = 2$), each
shown as black dots. **f**, Validation of C-terminal amidation as a MAAD as in **e** in
HEK293T cells using mTagBFP conjugated to amidated (-$CONH_2$) or unmodified
(-COOH) pep2 shown in **a** or the positive control degron (RxxG) as in **d**. Bars
represent means of three independent experiments ($n = 3$), each shown as
black dots. For gel source data, see Supplementary Fig. 1.

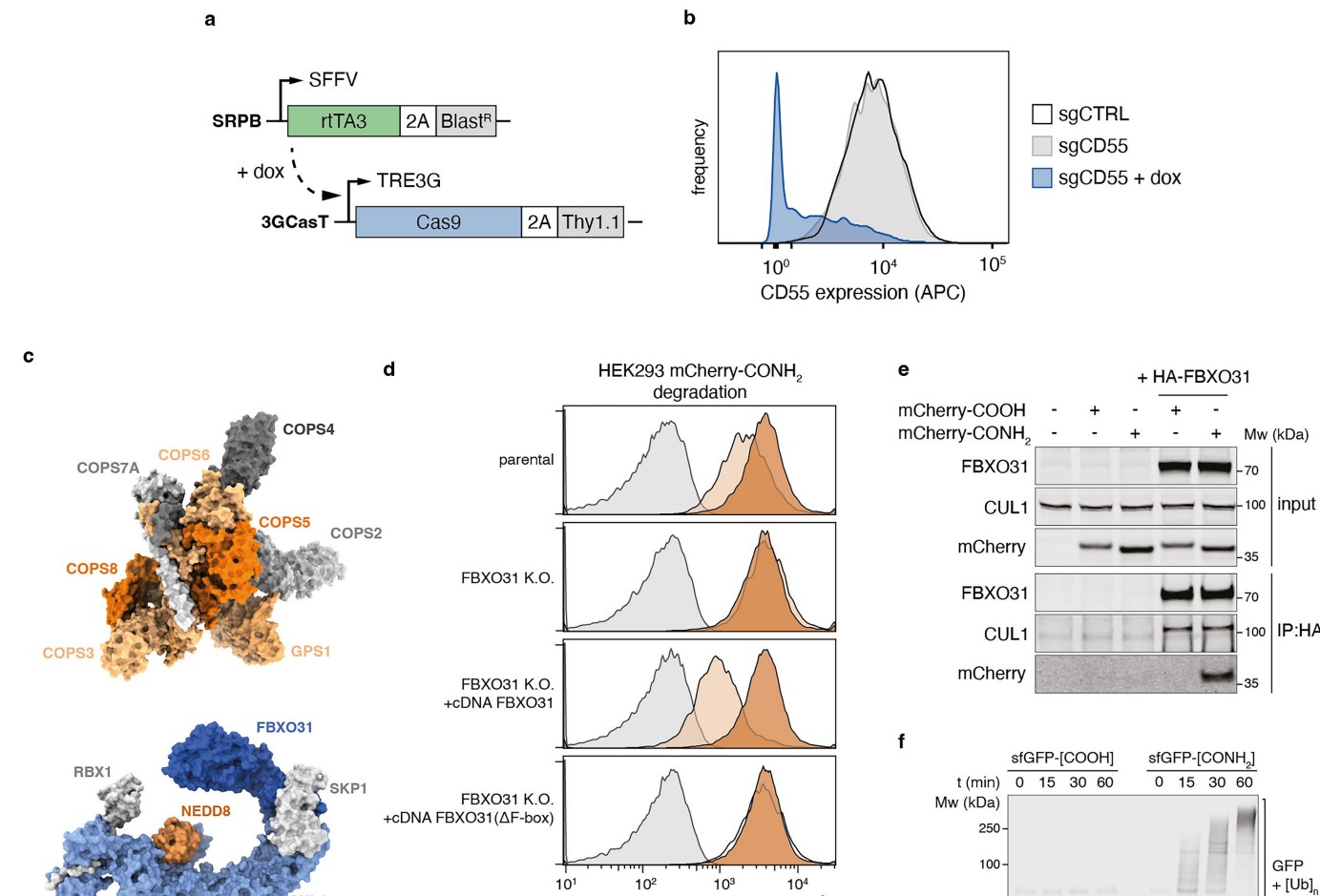

**Extended Data Fig. 2 | Inducible CRISPR screen identifies SCF/FBXO31 as a CTAP clearance factor. a**, Schematic of the vectors used to establish a doxycycline(dox)-inducible Cas9 expression system (iCas9) for CRISPR screening. **b**, Validation of inducible gene disruption in iCas9 cells using an sgRNA targeting cell surface marker CD55. Cells stably expressing sgCD55 were left untreated or on dox (500 ng/ml) for 9 days. Histogram shows CD55 surface expression measured by flow cytometry. **c**, Rendering of the COP9 signalosome complex from PDB:4D10 (top) and SCF/FBXO31 based on previously published structures (bottom, see Methods). Screening hits identified for CTAP clearance and NEDD8 are highlighted in blue or orange. **d**, CTAP degradation assay in *FBXO31* knockout cells with or without stable transduction of indicated cDNA

rescue constructs, as well as non-edited HEK293 cells (parental). **e**, Co-IP of HA-FBXO31 from *FBXO31* knockout HEK293 cells electroporated with the model substrate mCherry-pep2 (mCherry-GGGRRLEGKEEDEKGSRASD DFRDLR) with indicated C-termini. Cells were co-treated with the neddylation inhibitor MLN4924 (2 μM) and 500 nM epoxomicin and harvested after 2 h to stabilize CRL complexes and to prevent protein degradation. HA-FBXO31 was stably expressed as cDNA. **f**, In vitro ubiquitylation time course of sfGFP-pep1 (sfGFP-GGGKDLEGKGGSAGSGSAGGSKYPYDVPDYAKS) with indicated C-termini incubated with SCF/FBXO31, E1 and E2 enzymes. For gel source data, see Supplementary Fig. 1. Blots shown in **e** and **f** are representative of three and two independent experiments respectively.

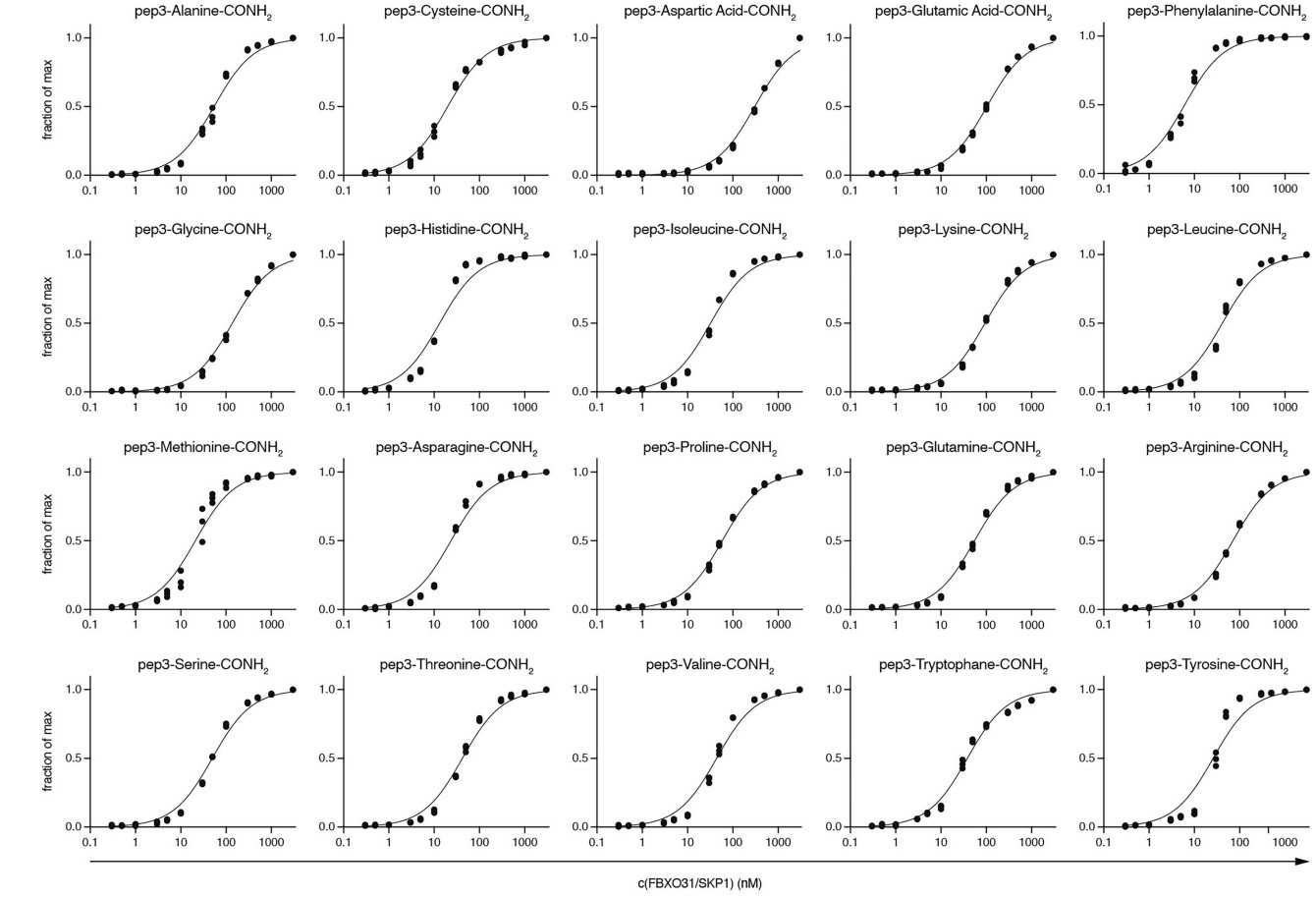

**Extended Data Fig. 3 | CTAP binding across terminal amino acid contexts by FBXO31.** Individual binding curves of pep3 (KKYRYDVPDYSA[X]-CONH$_2$) with each of the 20 canonical proteinogenic amino acids in the C-terminal positions. Corresponding dissociation constants are summarized in Fig. 3b. Black dots represent individual measurements for three parallel experiments ($n$ = 3).

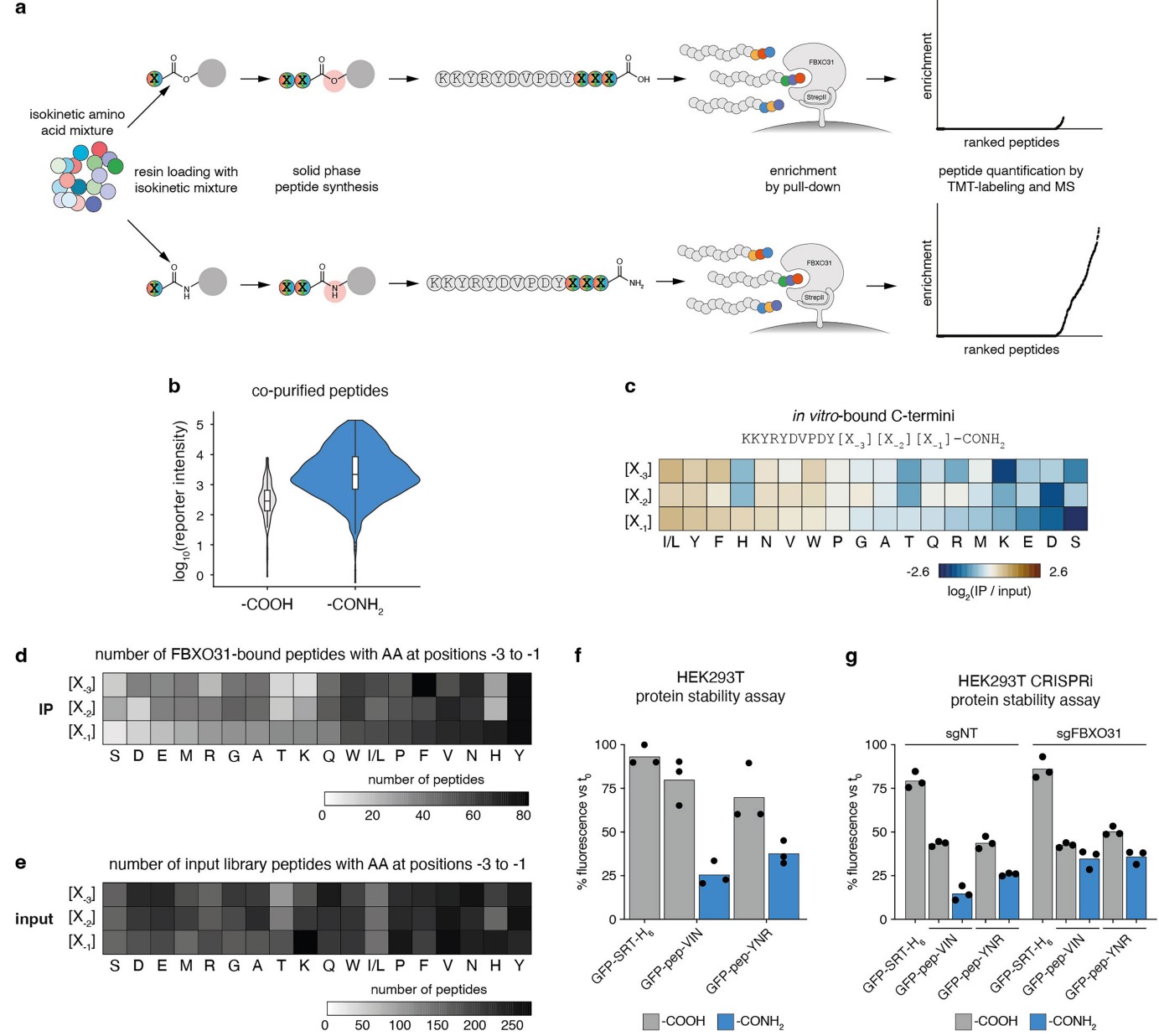

**Extended Data Fig. 4 | A pooled in vitro interaction screen for CTAP binding preferences of FBXO31. a**, Schematic showing analysis of peptide libraries with variable C-terminal amino acids by FBXO31 pull-down and subsequent MS/MS analysis. Peptide libraries are prepared using isokinetic amino acid mixtures. **b**, Violin plot showing number and relative recovery of peptides bound by FBXO31 in vitro as in **a** for unmodified (-COOH) and amide-bearing (-CONH$_2$) C-termini. Y-axis values depict total reporter ion intensities for identified peptides scaled to an isobarically labelled input library (see Methods). The median scaled intensity differs by 7.6-fold (p < 10-17, Wilcoxon rank-sum test). **c**, Heatmap of amino acid frequencies among FBXO31-bound peptides identified in **a** relative to input. **d**, Heatmaps showing absolute amino acid frequencies in the three terminal positions of 841 unique C-terminally amidated peptides co-precipitated with FBXO31. **e**, Heatmap as in **d** for all 3817 unique peptides identified in the input library. **f**, In-cell protein stability assay for sfGFP conjugated to top-scoring amide-bearing C-termini bound by FBXO31 in the pooled interaction screen. CRISPRi-competent HEK293T cells were transduced with non-targeting (NT) or FBXO31-targeting guides as indicated. **g**, In-cell protein stability assay as in **f** for a CRISPRi-competent clone of HEK293T expressing indicated sgRNAs. For **f** and **g**, bars represent the mean of three independent experiments (n = 3), each represented by black dots.

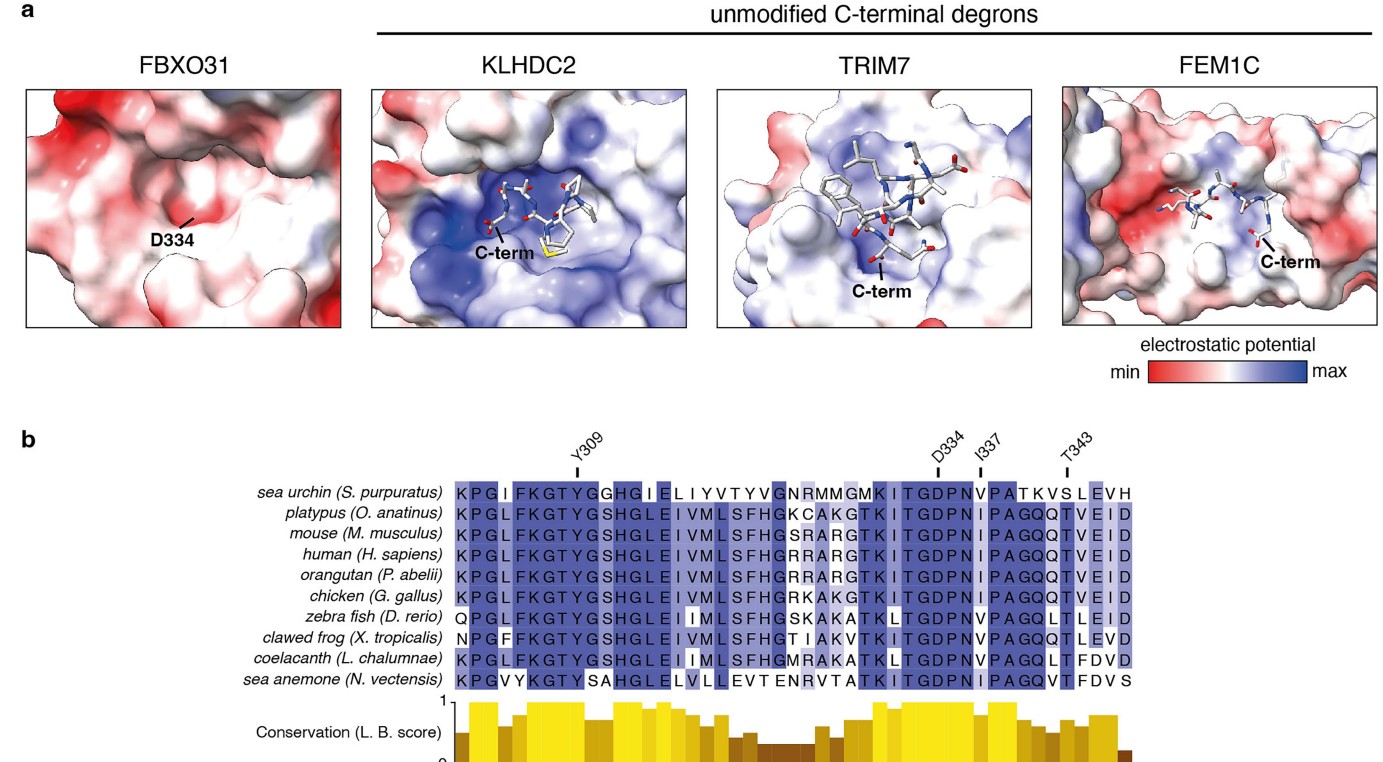

**Extended Data Fig. 5 | A conserved binding pocket enables amide-recognition by FBXO31. a,** Comparison of the negatively charged binding substrate pocket of FBXO31 with readers of unmodified C-terminal degrons. Electrostatic potential maps were rendered based on published structures with PDB accessions 5VZT (FBXO31), 6DO3 (KLHDC2), 7Y3A (TRIM7) and 6LEY (FEM1C). **b,** Protein sequence alignment of the FBXO31 substrate binding pocket from orthologues of indicated species. Conservation between the displayed sequences is expressed as Linvingston-Barton conservation score.

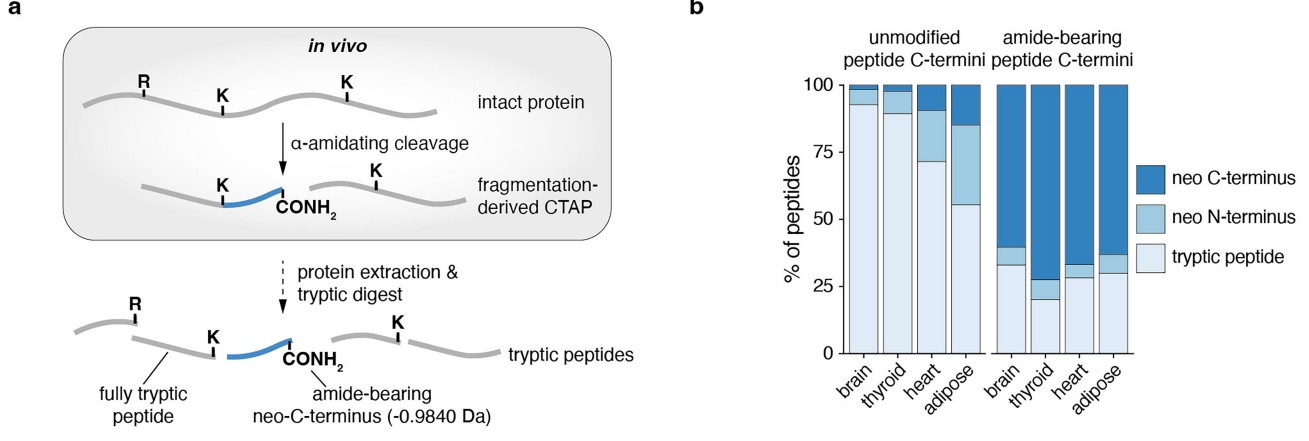

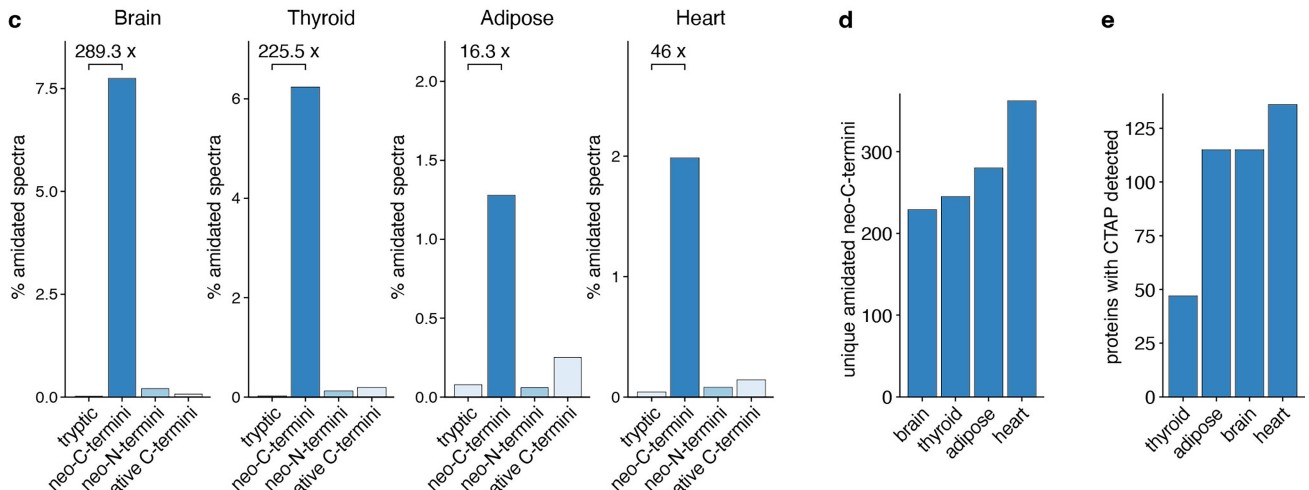

**Extended Data Fig. 6 | Evidence of CTAP formation in human tissue proteomes. a**, Schematic showing a CTAP stemming from alpha-amidating protein fragmentation and resulting peptides following tryptic digest for mass spectrometry. **b**, Re-analysis of public proteome data from indicated tissues showing the enrichment of C-terminal cleavage sites in amidated spectra. Peptides were divided into such with unchanged mass and C-terminally amidated ones (−0.984 Da) (see Methods). **c**, Complementary analysis to **b** quantifying degrees

of C-terminal amidation among identified spectra. Peptides were divided into such with fully enzymatic termini (tryptic), non-enzymatic C-termini (neo-C-termini), non-enzymatic N-termini (neo-N-termini) and native protein C-termini. Insets show the enrichment of amidation among neo-C-termini compared to tryptic peptides. **d**, Number of unique amide-bearing neo-C-termini (CTAPs) in indicated tissues from tissue proteomes shown in **b**. **e**, Number of individual proteins showing CTAP-formation in **b**.

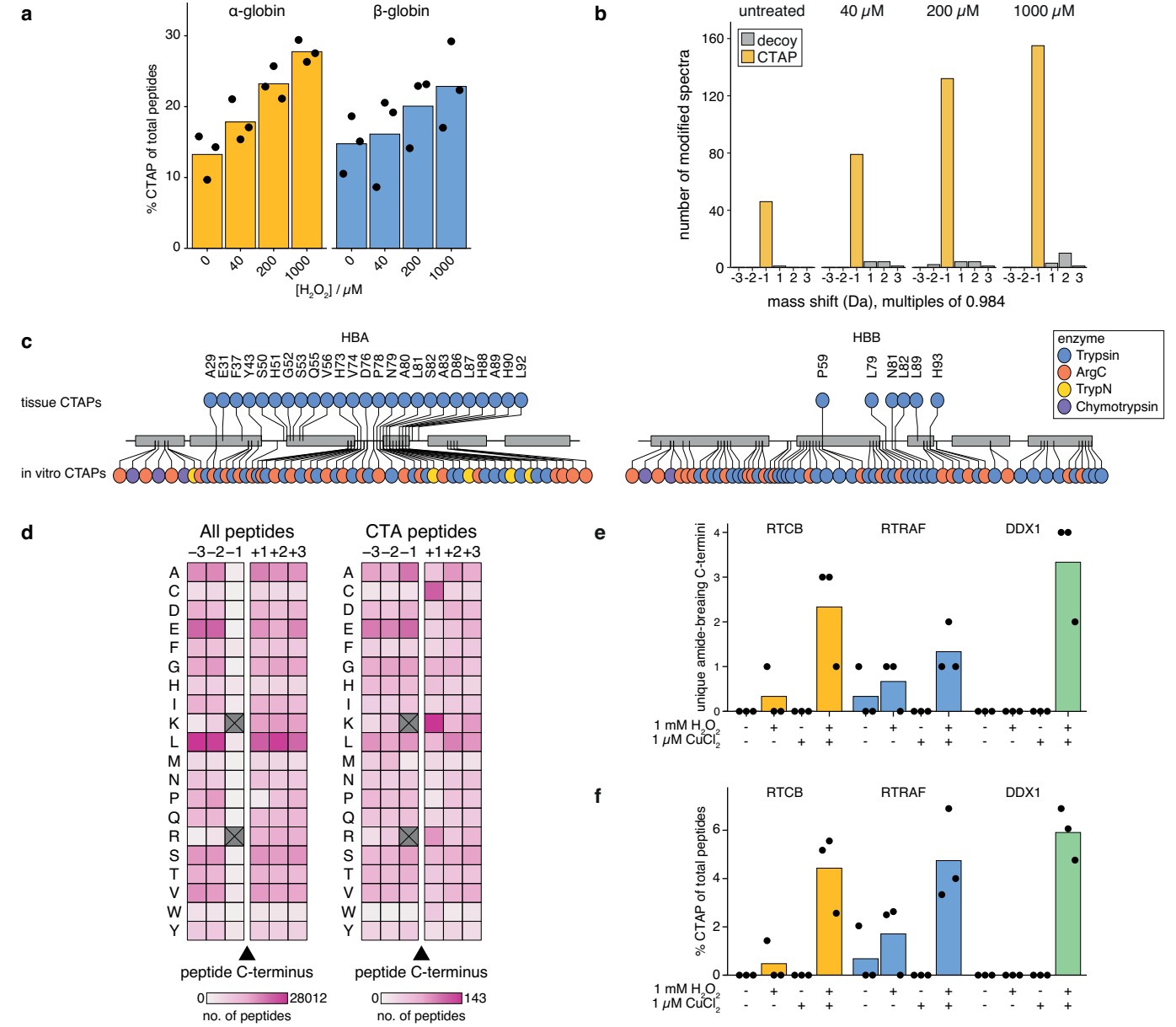

**Extended Data Fig. 7 | Oxidative protein damage is sufficient to trigger CTAP formation in vitro. a**, In vitro reconstitution of CTAP formation by oxidative protein cleavage. Purified human haemoglobin was exposed to indicated concentrations of hydrogen peroxide for 1 h at 37 °C. C-terminal amide-bearing neo-C-termini were detected by tryptic digest and MS/MS. **b**, MS/MS-based measurement of in vitro CTAP formation in haemoglobin as in **a**. To test specificity of the reported mass shifts in **a**, C-terminal mass shifts corresponding to different multiple of 0.984 were allowed. Mass shifts corresponding to CTAP-formation are highlighted in yellow. Bars represent total spectra identified in three independent experiments ($n = 3$). **c**, Comparison of CTAPs detected in public human proteomics data and following in vitro oxidative damage shown in **a**. Dots represent unique CTAP C-termini identified in each condition. Colours indicate proteases used during MS sample preparation. **d**, CTAP cleavage

patterns of in vivo CTAPs identified in public proteomics data as in Extended Data Fig. 6. Heatmaps indicate the number of amino acids detected among peptides at the three residues preceding or following CTAP cleavage sites, as well as at the C-termini for all quantified peptides for reference. K and R in position -1 from the peptide C-terminus were omitted as they are by definition absent in neo-C-termini and form the overwhelming majority in tryptic peptides. **e**, In vitro reconstitution of CTAP formation for recombinant human tRNA-ligase complex incubated for 30 min at room temperature in presence or absence of hydrogen peroxide and copper as indicated. C-terminal amide-bearing neo-C-termini were detected by MS/MS as in **a**. **f**, CTAP formation as in **e** measured as percentage of total detected peptides. For **a**, **e** and **f**, bars represent the mean of three independent experiments ($n = 3$), each represented by black dots.

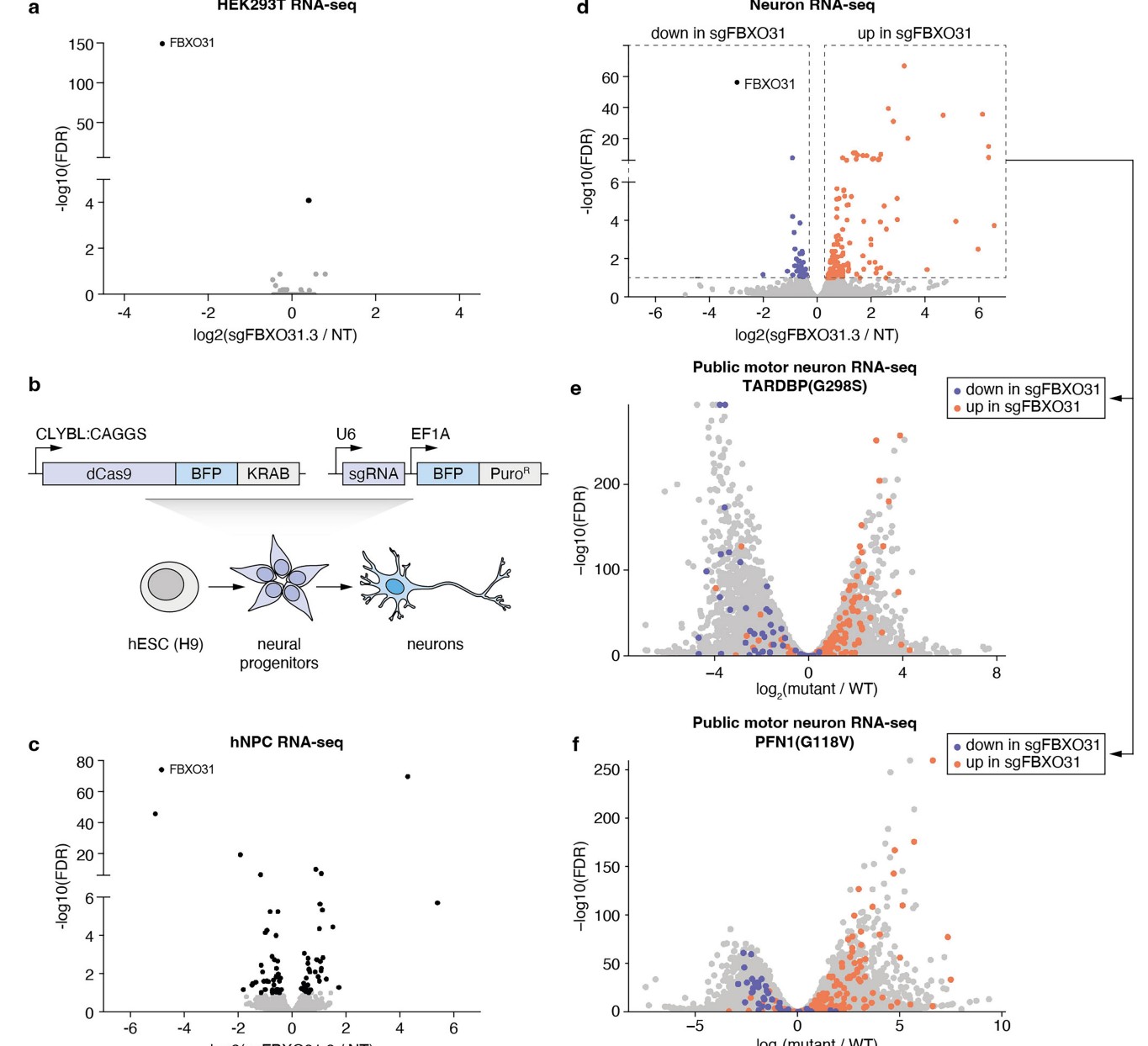

**Extended Data Fig. 8 | FBXO31-loss disproportionately affects mature neurons in vitro. a**, RNA-sequencing (RNA-seq) results showing lack of transcriptional responses to *FBXO31* knockdown by CRISPRi in rapidly proliferating HEK293T cells. **b**, Schematic of the generation of embryonic stem cell-derived neural progenitor cells (NPC). Cells were engineered to express the CRISPRi effector by knock-in into the CLYBL locus. Gene-specific sgRNAs are subsequently delivered by lentiviral integration. **c**, Transcriptional responses

to *FBXO31* knockdown in NPCs measured by RNA-seq. **d**, Transcriptional response to *FBXO31* knockdown in mature neurons after 24 days of differentiation. **e,f**, Published transcriptional signature of familial amyotrophic lateral sclerosis (ALS) driven by TARDBP(G298S) or PFN1(G118V) respectively as measured by RNA-seq of wild-type and mutant motor neurons. Genes regulated by *FBXO31* knockdown in neurons in **d** are highlighted as indicated.

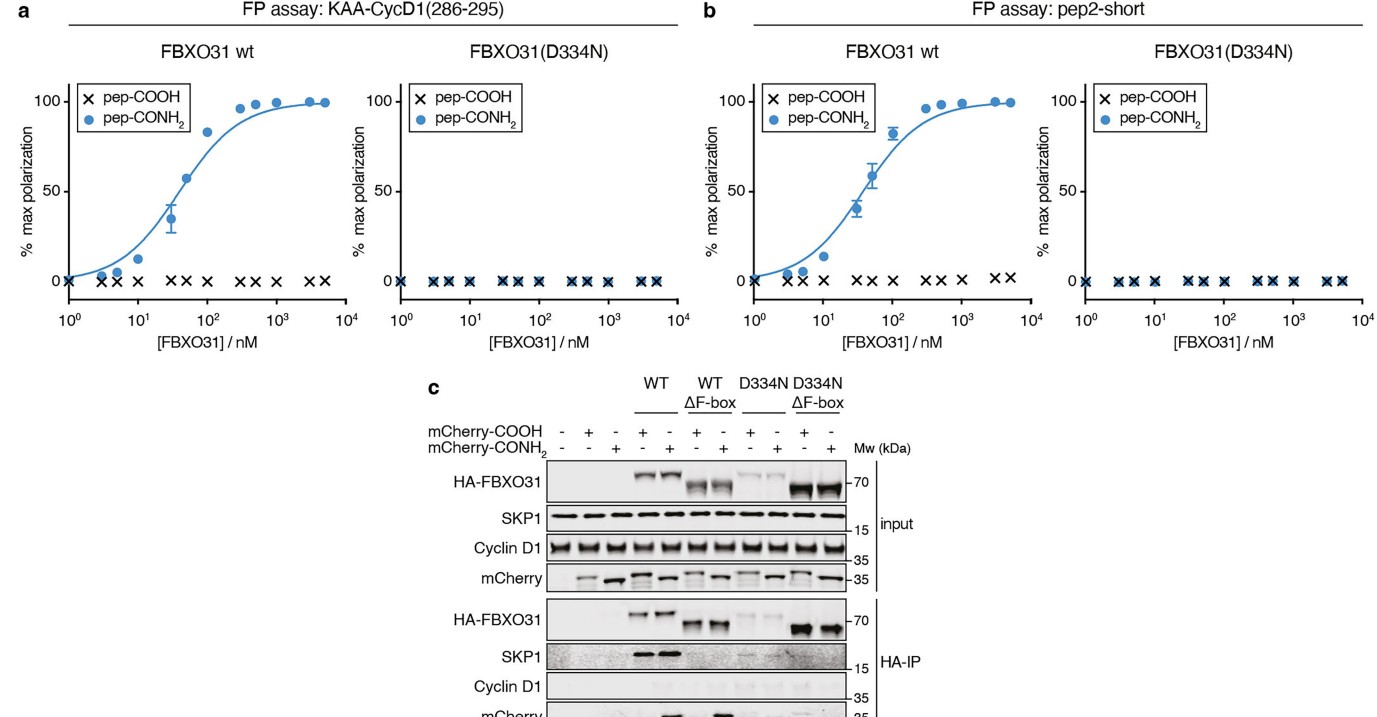

**Extended Data Fig. 9 | The D334N mutation disrupts CTAP recognition by FBXO31. a**, FP assay of for a peptide derived from the C-terminus of Cyclin D1 ([fluorescein]-KAATPTDVRDVDI) with (blue) or without C-terminal amide (black) in the presence of wild-type FBXO31/SKP1 (left) or its D334N mutant form. **b**, FP assays as **a** for a second peptide pair (pep2-short, KEEDEKGSRAS DDFRDLR). **c**, Co-IP of indicated HA-tagged FBXO31 cDNAs in *FBXO31* knockout HEK293 cells electroporated with a model substrate (mCherry-GGGRRLEGK EEDEKGSRASDDFRDLR). Cells were co-treated with 2 μM MLN4924 and 500 nM epoxomicin and harvested after 2 h. The blot is representative of two independent experiments. For **a** and **b**, data points represent mean ± SD (*n* = 3 parallel experiments). For gel source data, see Supplementary Fig. 1.

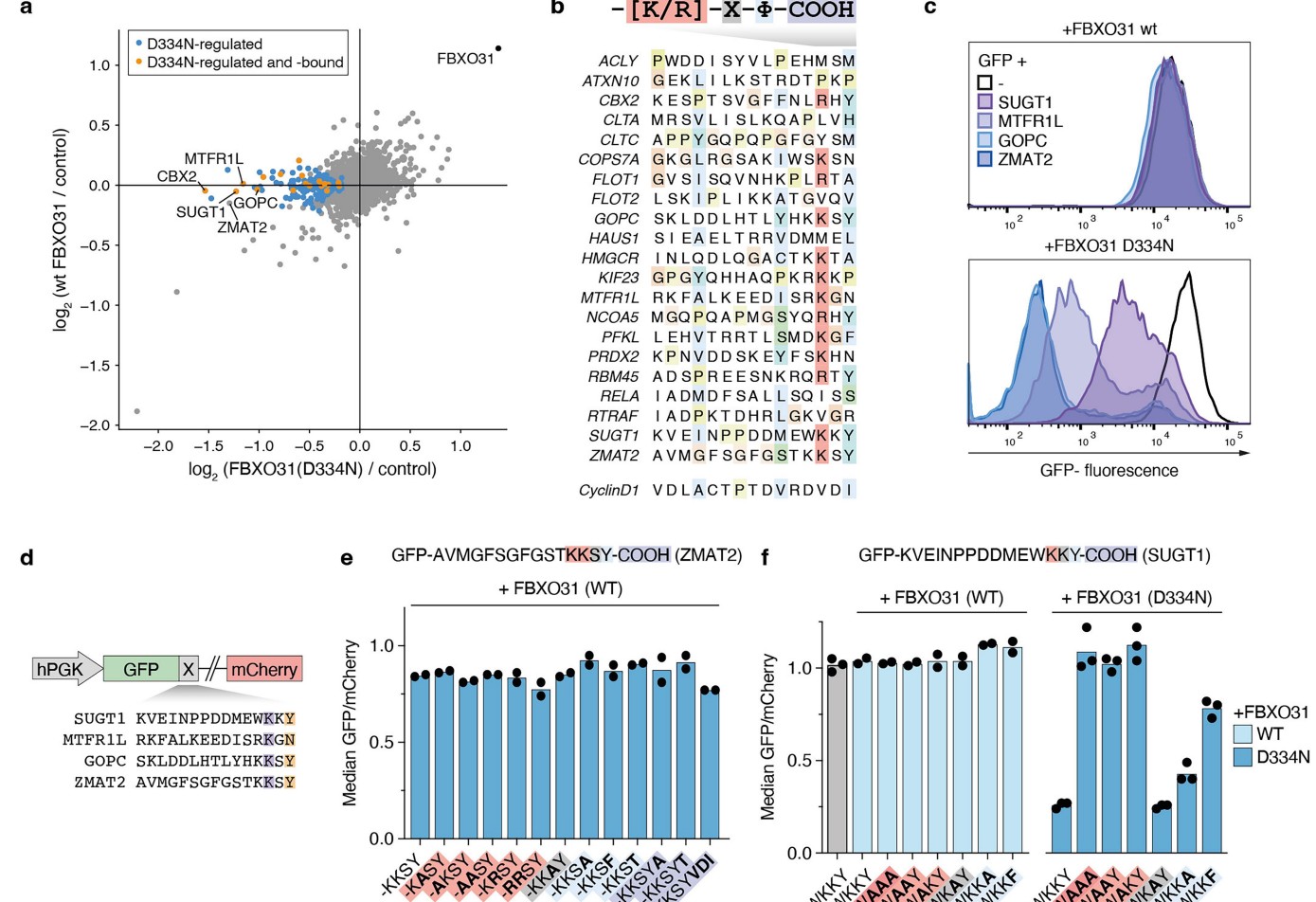

**Extended Data Fig. 10 | A C-terminal degron motif marks FBXO31(D334N) neosubstrates. a**, Comparison of proteome-wide responses to FBXO31 induction for wild-type or D334N mutant DD-3xFLAG-FBXO31 induced by 12 h of treatment with shield-1 measured by TMT-MS. Proteins are highlighted in blue if they are down-regulated only by mutant FBXO31 expression (FDR < 0.1) and in orange if they are also detected by FBXO31(D334N, ΔF-box) co-IP MS (FDR < 0.1). **b**, C-termini of bound and down-regulated proteins highlighted in **a**. Highlighted residues show a shared C-terminal motif. Inset on top shows a common C-terminal motif consisting of a basic residue (K/R) at position −3 and a hydrophobic residue (Φ) at position −1 from the C-terminus. **c**, Flow cytometry plot of the degron assay shown in Fig. 5d for *FBXO31* knockout HEK293T cells

expressing a GFP reporter with the indicated C-termini. Cells were additionally transduced with cDNAs encoding for FBXO31 wt (top) or FBXO31(D334N) (bottom) 2 days prior to acquisition. **d**, Schematic of the protein stability reporter vector used in **c**. **e**, Protein stability assay for the ZMAT2 C-terminal degron reporter and mutants thereof in *FBXO31* knockout HEK293T cells. Cells additionally received wildtype FBXO31 cDNA as a control for neosubstrate recognition in Fig. 5e. Y-axis values indicate the ratio of GFP-fluorescence to the co-expressed control protein mCherry. **f**, Mutational mapping of the C-terminal neo-substrate degron in SUGT1 as in **e**. For **e** and **f**, bars represent means of independent experiments each shown as black dots (*n* = 3 for FBXO31(D334N) and non-transduced cells, *n* = 2 for FBXO31(WT)).

# Reporting Summary

## Statistics

For all statistical analyses, confirm that the following items are present in the figure legend, table legend, main text, or Methods section.

| n/a | Confirmed | |
|---|---|---|
| ☐ | ☒ | The exact sample size (*n*) for each experimental group/condition, given as a discrete number and unit of measurement |
| ☐ | ☒ | A statement on whether measurements were taken from distinct samples or whether the same sample was measured repeatedly |
| ☐ | ☒ | The statistical test(s) used AND whether they are one- or two-sided *Only common tests should be described solely by name; describe more complex techniques in the Methods section.* |
| ☒ | ☐ | A description of all covariates tested |
| ☐ | ☒ | A description of any assumptions or corrections, such as tests of normality and adjustment for multiple comparisons |
| ☐ | ☒ | A full description of the statistical parameters including central tendency (e.g. means) or other basic estimates (e.g. regression coefficient) AND variation (e.g. standard deviation) or associated estimates of uncertainty (e.g. confidence intervals) |
| ☒ | ☐ | For null hypothesis testing, the test statistic (e.g. *F*, *t*, *r*) with confidence intervals, effect sizes, degrees of freedom and *P* value noted *Give P values as exact values whenever suitable.* |
| ☒ | ☐ | For Bayesian analysis, information on the choice of priors and Markov chain Monte Carlo settings |
| ☒ | ☐ | For hierarchical and complex designs, identification of the appropriate level for tests and full reporting of outcomes |
| ☒ | ☐ | Estimates of effect sizes (e.g. Cohen's *d*, Pearson's *r*), indicating how they were calculated |

*Our web collection on statistics for biologists contains articles on many of the points above.*

## Software and code

Policy information about availability of computer code

| Data collection | No specific software was used for data collection except the firmwares of measurement devices listed in Methods. |
|---|---|
| Data analysis | For FP-assays, curve fitting was performed using Prism 9 (GraphPad). CRISPR screen analysis was performed using MAGeCK (v0.5.9.3). Next generation sequencing validation of gene knockout was performed with CRISPResso2 (v2.3.0) TMT-based expression proteomics data was analyzed using Proteome Discoverer 2.4 (Thermo Fisher Scientific). For CRL-ABP DIA raw data was searched using DIA-NN (version 1.8.1) and the Perseus software package (v.1.6.7.0). IP-MS data was analyzed using FragPipe (v22.0) and the R package limma (3.58.1). Peptide library screens were analyzed using Mascot Server 2.8.1. (Matrix Science). Flow cytometry data was analyzed using FLowJo 10. Protein structures were visualized using ChimeraX-1.5. Paralogous protein sequences were aligned using JalView 2.11. RNA-seq data was analyzed with STAR-aligner (v 2.7.10a), FeatureCounts (v2.0.6) and DESeq2 (version 1.42.1). |

For manuscripts utilizing custom algorithms or software that are central to the research but not yet described in published literature, software must be made available to editors and reviewers. We strongly encourage code deposition in a community repository (e.g. GitHub). See the Nature Portfolio guidelines for submitting code & software for further information.

# Data

Policy information about availability of data

All manuscripts must include a data availability statement. This statement should provide the following information, where applicable:

- Accession codes, unique identifiers, or web links for publicly available datasets
- A description of any restrictions on data availability
- For clinical datasets or third party data, please ensure that the statement adheres to our policy

All large-scale data sets displayed in this paper are provided in extended data tables. All additional numerical data displayed in this study is provided as source data tables. Uncropped images of all gels and blots are provided in Supplementary Figure 1. Unprocessed data for proteomics and RNA-seq experiments is accessible through the following public repositories. Raw proteomics data is deposited alongside primary search results and complete search parameters at ebi.ac.uk/pride/ under accession numbers PXD055535 (FBXO31 IP-MS), PXD055814 (all CTAP identifications), PXD055518 (CRL profiling), and PXD055818 (TMT-based proteome quantification). Raw sequencing reads for RNA-seq samples are deposited to ncbi.nlm.nih.gov/sra under bioproject number PRJNA1173967. Deep sequencing read counts for CRISPR screen analysis are provided in Supplementary Table 9.

Public proteomics data re-analyzed in this study was obtained from ebi.ac.uk/pride/ under accession number PXD010154, sample numbers 1277, 1499, 1306 and 1296. Public protein structures were downloaded from www.rcsb.org (accession numbers 5VZU, 6TTU, 6DO3, 6LEY and 7Y3A). RNA-seq reads were aligned against the human genome reference GRCh38. Proteomics data was searched against the Uniprot human proteome reference UP000005640.

# Research involving human participants, their data, or biological material

Policy information about studies with human participants or human data. See also policy information about sex, gender (identity/presentation), and sexual orientation and race, ethnicity and racism.

| | |
|---|---|
| Reporting on sex and gender | n.a. |
| Reporting on race, ethnicity, or other socially relevant groupings | n.a. |
| Population characteristics | n.a. |
| Recruitment | n.a. |
| Ethics oversight | n.a. |

Note that full information on the approval of the study protocol must also be provided in the manuscript.

# Field-specific reporting

Please select the one below that is the best fit for your research. If you are not sure, read the appropriate sections before making your selection.

☒ Life sciences ☐ Behavioural & social sciences ☐ Ecological, evolutionary & environmental sciences

For a reference copy of the document with all sections, see nature.com/documents/nr-reporting-summary-flat.pdf

# Life sciences study design

All studies must disclose on these points even when the disclosure is negative.

| | |
|---|---|
| Sample size | For CRISPR screening, the chosen library representation, number of sgRNAs and replicates was based on prior experience with CRISPR screening and in accordance with mathematical models of screen performance (Nagy & Kampmann, BMC Bioinfrmatics 2017).<br><br>For IP-MS experiments, replicate numbers of initial experiments were based on prior experience. Based on the variation of initial experiments a sample size of n=3 was estimated to be required for detecting >2-fold enrichment with a power of >0.8 using a two-sided unpaired t-test comparing means of spectral counts. Based on these estimates, other IP-MS experiments with identical cell types and antibodies were also performed using three replicates.<br><br>All other experiments were performed on purified proteins and immortalized isogenic human cell lines that display minimal variance between independent experiments. We therefore chose a sample size of n=3 based on own experience and common practice in the field for work with immortalized cell lines. |
| Data exclusions | No data was excluded from analysis for any of the shown experiments. In-cell protein stability assays were aborted if electroporation failed. |
| Replication | Replicate numbers and designs of individual experiments are indicated in the respective figure legends. In addition, key experiments including in cell degradation assays, co-IP assays and in vitro binding assays were independently replicated by two or more experimenters. No attempt at replication failed. |

| Randomization | All experiments were performed on immortalized human cell lines or purified proteins. No assignment of specimens into experimental groups which would require randomization was performed. |
| Blinding | All measurements were performed automatically by devices (flow cytometers, photometers, gel scanners, mass spectrometers or NGS devices). No manual counting or measurements were performed. Therefore, no blinding was required. |

# Reporting for specific materials, systems and methods

We require information from authors about some types of materials, experimental systems and methods used in many studies. Here, indicate whether each material, system or method listed is relevant to your study. If you are not sure if a list item applies to your research, read the appropriate section before selecting a response.

## Materials & experimental systems

| n/a | Involved in the study |
| --- | --- |
| ☐ | ☒ Antibodies |
| ☐ | ☒ Eukaryotic cell lines |
| ☒ | ☐ Palaeontology and archaeology |
| ☒ | ☐ Animals and other organisms |
| ☒ | ☐ Clinical data |
| ☒ | ☐ Dual use research of concern |
| ☒ | ☐ Plants |

## Methods

| n/a | Involved in the study |
| --- | --- |
| ☒ | ☐ ChIP-seq |
| ☐ | ☒ Flow cytometry |
| ☒ | ☐ MRI-based neuroimaging |

## Antibodies

| Antibodies used | Primary western blotting antibodies: FBXO31 (Abcam, ab86137), FBXO31 (HPA HPA030150), GFP (Abcam, ab6556), mCherry (Abcam, ab183628), SKP1 (Cell Signaling Technology, 2156), CUL1 (Invitrogen, 71-8700), AARS1 (Fortis/Bethyl Life Science, A303-473A), GLUL (Fortis/Bethyl Life Science, A305-323A), Cyclin D1 (Abcam, ab134175), HA (Cell Signaling Technology, 3724) and SUGT1 (Bethyl, A302-944A). Primary western blotting antibodies for mCherry and GFP were used at 1:2000 dilution. All other primary antibodies were used at 1:1000 dilution. <br><br> Secondary western blotting antibodies: LI-COR Biosciences, cat. nr. 926-32213 and 926-68072. Dilution 1:15000 <br><br> Flow cytometry antibodies: APC-anti-human-CD55[JS11] (Biolegend). Dilution 1:200 for cell immunostaining. |
| Validation | Antibodies for FBXO31 and HA were validated by overexpression in knockout backgrounds. Antibodies against GFP, mCherry and SUGT1 were validated by immuno-blotting of purified proteins. The SKP1 antibody was chosen based on the vendors' validation and 34 previously published studies. The CUL1 antibody was chosen based on 60 previous publications and validated internally by treatment with neddylation inhibitor MLN4924 nducing the expected Mw shift. The Cyclin D1 antibody was chosen based on previous validation by overexpression and studies of its regulation (e.g. doi.org/10.1038/s41586-021-03445-y). The AARS1 antibody was validated in previous studies by cDNA overexpression (e.g. doi.org/10.1101/2022.05.25.493316). The GLUL antibody was validated by the vendor by immunoprecipitation and blotting with two independently derived antibodies. |

## Eukaryotic cell lines

Policy information about cell lines and Sex and Gender in Research

| Cell line source(s) | Parental cell lines (K562, HEK293, HEK293T) were obtained from ATCC or the Berkeley Cell Culture Facility (UC Berkeley). SF9 insect cells were obtained from Thermo Fisher Scientific. NPCs were derived by the group of Prof. Jessberger |
| Authentication | Parental cell lines were obtained authenticated from certified vendors. HEK293 and HEK293T were distinguished using custom primers targeting the large T antigen. K562 CRIPSRi cells were re-validated by STR-phenotyping (Microsynth). Neural stem cell identity and differentiation was confirmed by immunofluorescence imaging and transcriptomics. SF9 insect cells were obtained authenticated from Thermo Fisher. |
| Mycoplasma contamination | Human cell lines were regularly tested for mycoplasma contamination using a commercially available kit (Lonza, MycoAlert). Throughout the duration of this study, no contamination was detected. Sf9 cells used exclusively for protein production were not tested for presence of mycoplasma. |
| Commonly misidentified lines (See ICLAC register) | none |

# Plants

| | |
|---|---|
| Seed stocks | n.a. |
| Novel plant genotypes | n.a. |
| Authentication | n.a. |

# Flow Cytometry

## Plots

Confirm that:

☒ The axis labels state the marker and fluorochrome used (e.g. CD4-FITC).

☒ The axis scales are clearly visible. Include numbers along axes only for bottom left plot of group (a 'group' is an analysis of identical markers).

☐ All plots are contour plots with outliers or pseudocolor plots.

☒ A numerical value for number of cells or percentage (with statistics) is provided.

## Methodology

| | |
|---|---|
| Sample preparation | (See also Methods). For protein stability assays, cells were centrifuged and resuspended in PBS prior to acquisition. Cell-surface expression of CD55 was assessed by immunostaining (incubation with antibody APC-CD55 diluted 1:200 in PBS + 5% FBS, see Methods). |
| Instrument | Analytical flow cytometry was performed on an Attune NxT cytometer (Thermo Fisher Scientific). Cell sorting was performed using a Sony SH-800 cell sorter. |
| Software | FlowJo 10 |
| Cell population abundance | For CRISPR screening, 32 million cells of a background population of GFP-negative cells (bottom 50%) were obtained per replicate. For the CTAP-clearance deficient target population (BFP-GFP+; ca ), 1.3 and 2.2 million cells were obtained per replicate. |
| Gating strategy | See also Supplementary Figure 2.  Events were subsetted for live cells based on their FSC/SSC characteristics and further gated for singlets based on height and width of the forward scatter. For competitive proliferation assays, cells expressing FBXO31 cDNAs were identified by additional gating for the co-delivered GFP marker, using non-transduced cells as a reference sample.<br><br>For CRIPSR screening, the CTAP degradation-proficient background population was defined as the lower 50% of GFP expression, which also appears GFP-negative compared to a non-nucleofected control population. The CTAP degradation-proficient target population was defined as BFP-GFP+ based on a negative control population receiving no fluorescent protein. For all other experiments, no dichotomization into marker-positive and -negative populations was performed.<br><br>We confirmed in initial experiments that protein electroporation uniformly delivered fluorescent reporter proteins to all cells, therefore not necessitating division into electroporated and non-electroporated cells. We therefore extracted the median fluorescence intensity for all live-gated single cells to quantify model substrate abundance throughout protein stability assays.<br><br>For dual-color reporter assays of protein stability, cells were gated for successful transduction and selection for the reporter vector by subsetting live single cells for mCherry-positive cells.<br><br>Where a single fluorescence parameter (GFP) or its derivative (dual-color-reporter) was analyzed, fluorescence levels were plotted as histograms instead of pseudocolor or contour plots. |

☒ Tick this box to confirm that a figure exemplifying the gating strategy is provided in the Supplementary Information.

