## [Peer Review file · Nature]

C-terminal amides mark proteins for degradation via SCF/FBXO31

Corresponding Author: Professor Jacob E. Corn and Jeffrey W. Bode

File names: Yfgjcb\ Ug\ VYYb\ gYYb\ VmiA\ YfYZf\ YfYf\

Version 0:

Reviewer comments:

Referee #1

(Remarks to the Author)

Muhar, Farnung, et al use a GFP reporter assay to display a set of chemical modifications as candidate forms of protein damage. They found that proteins bearing C-terminal amides (CTAPs) were degraded in a ubiquitin- and proteasome-dependent manner and use a genome-wide CRISPR screen to discover FBXO31 as the E3 ligase recognizing this motif. The interaction is rigorously characterized against a peptide library to reveal the C-terminal amide as the minimal structural determinant with some preference for hydrophobic amino acids. C-terminal amides may form by oxidative cleavage. In support of this mechanism, the authors use IP-MS to reveal a set of proteins with increased binding to FBXO31 after hydrogen peroxide treatment and show that two of these potential substrates (GLUL and AARS1) are co-immunoprecipitated and the C-terminal amide sequence serves as substrates for FBXO31 when installed to the GFP reporter. Finally, the authors closely evaluated FBXO31(D334N), a mutant associated with cerebral-palsy, and found that the mutation abolishes CTAP-directed activity and gains the ability to target certain neosubstrates such as SUGT1 for ubiquitination. Expression of FBXO31(D334N) was less tolerated in co-culture compared to that of WT, which may relate to the toxicity of this mutant. Overall, this manuscript reports a novel contribution to our understanding of the role of non-enzymatic chemical modifications, with implications in protein homeostasis, E3 ligase biology, and potentially the eventual therapeutic utilization of these observations. The evidence in favor of FBXO31 recognition of C-terminal amides is well-grounded and rigorously demonstrated, which affords a very interesting explanation for a dominant negative mutation associated with cerebral-palsy. However, the crux of this study lies in the connection of FBXO31 in targeting endogenous CTAPs (Figure 4). While the authors provide initial evidence in support of FBXO31 regulation of endogenous CTAPs, additional studies that establish the cellular occurrence of CTAPs and that these modifications on endogenous substrates are recognized by FBXO31 will strengthen the key conclusions of this work.

Major points

1. Establishing that an endogenous protein substrate with a C-terminal amide modification is a bona fide substrate of the FBXO31 is critical to support the evidence from co-IP experiments and the GFP reporter assay already provided. At a minimum demonstration of FBXO31 substrate recognition beyond engineering GFP is necessary. The demonstration of a bona fide substrate may be achieved by extending data with AARS1. Is the truncated AARS1 that binds to FBXO31 a ubiquitylated substrate in cells or in vitro? Does mutation of the oxidative cleavage sites on AARS1 alter formation of this damage event, such that it is no longer generated and therefore rescued as a substrate for FBXO31? Is there a structural rationale and/or conservation of the AARS1 site conserved and potentially a signal site for oxidative damage?
2. The formation of CTAPs is proposed to occur by an oxidative cleavage mechanism with H₂O₂, which is a non-specific form of protein damage that would preferentially target many other oxidizable sites in cells. It is very possible that the extent of this form of protein damage has been overlooked to date, adding to the novelty of this work. Additional understanding of C-terminal amide formation on endogenous substrates and their response to FBXO31 knockdown would provide context for the extent of this form of protein damage, further establish these modifications as responsive to FBXO31, and show that the CTAP fragments accumulate in its absence. For instance, measurement of CTAP formation/FBXO31 recognition on specific proteins in vitro or comparison of modification events in FBXO31 KO cells (versus WT) after oxidative damage would establish that these modifications accumulate in response to loss of FBXO31.
3. CTAP formation is supported by performing proteomic analysis of public proteome datasets for neo-C-termini. Tables reporting these peptides need to be provided. Do the authors observe a consensus sequence in human tissue proteomics

that aligns with the preferential recognition sequence for FBXO31? Do the proteins in these data align with those that are observed by the authors following oxidative damage? Does analysis of a dataset following oxidative damage (be it publicly available or from the authors) possess an increase in C-terminal amide modifications?

4. The pooled peptide input library should be added as a supplementary table. For the D334N selection, it is clear the C-terminal amide pool has been lost, but why not gain the C-terminal carboxylic acid pool, given that many substrates are gained in the IP-MS and global proteomics?

5. For Figure 3f, please add rescue with FBXO31 KO or mutation.

6. Does HA-FBXO31(deltaFBox) still ligand to peptide substrates similarly to the full length FBXO31?

Minor points

1. The oxidative cleavage mechanism will generate alongside a potentially highly toxic neo-N-terminal aldehyde (in the case of glycine) or ketone (other amino acids). The neo-N-terminal fragment would be potentially more damaging to the cell as it is the oxidized fragment. Are these neo-N-termini observed in the datasets and is there a conserved cellular response known or observed? Relevant to the CTAP itself, how long lived is the C-terminal amide when formed, how quickly can it be hydrolyzed? Why would the cell conserve an approach to get rid of something that can hydrolyze? Please discuss.

2. In Line 389–390, the authors state that “C-terminal amidation could render cyclin D1 an FBXO31 client in other cell types or following stress...”. Have the authors tried to validate that oxidative stress can lead to Cyclin D1 interaction with WT but not D334N, similar to Figure 4d? This would link the previously reported biology of FBXO31 to its regulation of oxidation-induced CTAPs.

3. The LC-MS/MS acquisition settings for all proteomics experiments are missing and should be elaborated in the Methods section.

4. Please specify the protease used for digestion in the Methods section “Expression proteomics by tandem mass tag (TMT) MS”. The reference for automated SP3 workflow should also be included.

Referee #2

(Remarks to the Author)

Substrate specificity for E3 ubiquitin ligases is at the heart of ubiquitin signaling. While early work had suggested that E3 ligases mainly recognize linear degrons, which might in some cases be modified by phosphorylation or related signals, recent work has shown that E3 ligases can have a much broader substrate specificity. This included three-dimensional degrons as well as degrons modified at their C-termini. However, very few E3 ligases with specificity for C-terminal modifications have been identified (i.e. CRBN), and general platforms to find such enzymes have not been developed.

In this study, Corn et al. develop a system to screen for E3 ligases that identify substrates carrying specific posttranslational modifications. This work thus lays the groundwork for much larger studies interrogating PTM-driven protein degradation, which is important by itself. Using their new approach, the authors found that proteins carrying C-terminal amides (CTAPs), potentially generated by oxidative cleavage of the peptide backbone, are unstable and cleared by the ubiquitin-proteasome system. They used CRISPR screens to identify SCF-FBXO31 as the E3 ligase responsible for CTAP clearance. The data documenting the role of SCF-FBXO31 in CTAP clearance includes cellular approaches, such as rescue of knockout, and reconstitution of binding and ubiquitylation – it shows readers how substrate-enzyme interactions should always be documented for the ubiquitin field! They next identified the degron for FBXO31 – showing a requirement for a C-terminal amide, but not much more, as expected for a broadly acting quality control enzyme. They suggest that CTAPs are formed under oxidative stress conditions, although experiments were performed under harsh H₂O₂ treatment. A disease-linked mutation in FBXO31 leads to degradation of neo-substrates that are not normally turned over through FBXO31, suggesting that the striking specificity of FBXO31 is critical for human physiology.

Overall, this is a spectacular paper with respect to the characterization of an E3 ligase with an interesting substrate specificity for CTAPs. This paper sets the stage for the discovery of many E3 ligases with specificity for modified degrons. The study lacks, however, physiological significance, as it is neither clear whether specific conditions generate CTAPs, why CTAPs would need to be eliminated from cells (i.e. if they are so rare that they can only be discovered upon harsh oxidative stress), and whether this pathway provides cells with survival benefits. These issues need to be addressed by new experiments, but I strongly believe that the authors could accomplish this fairly simply (as outlined below) to turn this study into a landmark paper fully worth of publication in Nature.

Major issues:

1. Critical to the relevance of this study, CTAPs must be formed *in vivo* under conditions that are experienced by cells within an organism, and FBXO31 must recognize and clear such cleavage products. The experiments showing AARS1 cleavage upon treatment of cells with 200 μ M H₂O₂ are not sufficient to address this question. For a paper potentially defining a new

research field (which I believe this manuscript has the potential to accomplish), I would expect an IP/MS experiment showing binding to an endogenously cleaved protein, under conditions that can be experienced in an organism (i.e. not in response to H₂O₂). Maybe the authors could produce oxidative stress upon depletion of SOD1 or related ROS-detoxifying enzymes, as seen in ALS patients? Maybe the authors could deplete metallothionein, the protein that is most strongly correlating with FBXO31 in DepMap and that protects cells from oxidative stress? Or they could look in cells of the immune system after pathogen invasion, i.e. conditions that produce ROS through NOX enzymes? Seeing that an endogenous CTAP, as identified through the MS profile, would be bound to FBXO31 under such conditions would massively increase the impact of this study. Knowing more about such substrates might also inform the authors about why CTAP proteins might be dangerous from cells and need to be removed immediately (in other words, why could cells not tolerate such proteins or remove them through complementary pathways, such as those detecting unfolding)?

2. The physiological relevance of FBXO31 is unclear. Do FBXO31-deletion cells die earlier than wildtype cells when exposed to oxidative stress or any other stress condition that might result in the cleavage of peptide backbones? This should be addressed through genetic interactions, not by treating cells with chemicals such as H₂O₂. As the principle of C-terminal modifications as targeting signals for E3 ligases has been established through Christina Woo's work on CRBN, I believe it is critical that evidence for physiological relevance is provided for this paper on FBXO31.

Referee #3

(Remarks to the Author)

In this manuscript by Muhar et al., the authors describe the discovery that the SCF substrate receptor FBXO31 recognizes c-terminal amide bearing peptides (which they call CTAP) and that this leads to degradation of a protein reporter of GFP/RFP conjugated to such a modified peptide. The authors further report that such c-terminal modification are the result of oxidative protein damage and cleared by FBXO31 to maintain cellular homeostasis.

The initial identification of CTAP followed a clever screen using a semi-synthetic protein library introduced into cells, which identified CTAP as the only modification leading to protein degradation. They next went on to identify the ligase machinery necessary for reporter degradation and identified and validated FBXO31. Using recombinant protein, peptide synthesis and biochemical and biophysical methods they establish that FBXO31 exhibits high affinity for amide bearing c-termini but not unmodified, with little specificity for the preceding amino acids. They also identify a mutation in the putative binding pocket of FBXO31 that prevents binding of CTAP peptides. To answer the most critical question, whether these CTAP modifications exist in vivo and whether FBXO31 plays a role in clearance, the authors first look at published global mass spectrometry datasets and identify the modification on truncated proteins albeit little detail is provided. They further turn to affinity mass spectrometry (IP-MS) to identify substrates of FBXO31 in response to H₂O₂ treatment, which leads them to identify AARS1 and GLUL amongst others. They lastly look at a patient mutation in FBXO31, which surprisingly instead of leading to impaired clearance of damaged proteins leads to degradation of a large set of specific substrates. They explain this by the D334N mutation being a gain of function mutation.

While the data characterizing the interaction of FBXO31 with CTAP peptides as well as the CTAP peptide serving as a degron on reporter proteins is well supported by data, significant open questions remain about the endogenous role of FBXO31. The endogenous role needs to be either toned down or clear evidence for the existence and physiological relevance of CTAP should be presented, which would make this work relevant to a broad audience.

Major points:

- The most important question for this work is whether CTAP bearing peptides exist in vivo. The authors address this predominantly through re-evaluation of public proteomics datasets (EDFig 6), however, the data presented is not convincing. Amidation is common (+0.9840 mass shift) and the presented single spectrum is not sufficient to convince me these are c-terminal modifications. The gold standard would be to use reference peptides to calibrate retention times and look at both retention time and MS fragmentation. Where do these peptides fall in the confidence space? These experiments are critical and should be done properly and presented as main figure.
- In addition to the above mentioned re-analysis of human tissue data, the authors also look for these modifications in FBXO31 knockout cells using FBXO31 as a bait under basal and H₂O₂ conditions. This leads them to identify additional putative substrates. There are some critical controls missing here that again weaken the data:
 - o The authors should repeat the same experiments in FBXO31 wt cells as these should clear all the peptides and therefore serve as the better positive control.
 - o The authors should present the peptide spectra supporting the presence of modifications and retention times / MS trace.
 - o How do the authors rule out secondary effects due to H₂O₂ treatment leading to some of the changes?
- Fig 4e. Proper analytical data needs to be shown in suppl/ED to support the statement. What fraction of AARS1/GLUL are cleaved at these sites?
- The findings on the D334N mutation are interesting, but could be interpreted in multiple different ways and raises more questions than answers. How are these neo-substrates recognized? It has to be the same pocket, does it change peptide specificity? What do these neo-substrates have in common? Could CTAP be an inhibitor for the ligase rather than being a substrate itself?
- Following on this, an obvious key question is what do the other mutations do in IP-MS/Global proteomics? Do they lead to similar gain of function?

Additional points:

- The authors should provide FP/binding data on all the mutants used (Y309A, D334N, I337D, T343V) to CTAP and control peptides
- It is interesting that none of the other modifications leads to degradation, does that argue against PTM being broadly responsible for regulating protein stability?

Version 1:

Reviewer comments:

Referee #1

(Remarks to the Author)

I enjoyed reading the updates to this manuscript. The authors have clearly extended their findings to thoroughly demonstrate the overlooked formation of CTAPs as a form of oxidative damage in vitro and in multiple cell types that is then recognized and removed by FBXO31. The gain of function FBXO31 D334N mutation is also remarkable. I only noted a few typos in the new figures that can be readily addressed by careful review. I look forward to sharing this work with my lab.

Referee #2

(Remarks to the Author)

The authors have addressed all my initial concerns. Congratulations to the authors on an important and thorough manuscript that will open up new areas of biology for investigation.

Referee #3

(Remarks to the Author)

As previously noted, this study by Muhar et al., reports a novel and potentially very important mechanism how a ubiquitin ligase would recognize damaged proteins for clearance. The experiments are well executed and the work clearly presented. The revised manuscript has addressed many of the points raised and substantially enhanced the overall manuscript, but unfortunately fails to address the key question raised by all three reviewers: Do these modifications exist under physiological conditions and is recognition by FBXO31 the mechanism for clearance. This is unfortunate given that many straight forward experiments were suggested by all reviewers. As reviewer #2 has pointed out, the ability of ubiquitin ligases to recognize c-terminal modifications on peptides has been established by work from Christina Woo's lab and this reviewer agrees that to justify publication in Nature, the existence of such modifications under physiological conditions needs to be established. The in vitro experiments added during revisions confirm that damage to proteins by e.g. H₂O₂ can lead to CTAP, but this is not the key question. If the authors can confirm the existence of those modifications and the necessity of the modifications for recognition by FBXO31 under physiological conditions, the paper would be an outstanding candidate for publication.

Major points

- The authors need to demonstrate the existence of CTAP modifications under physiological conditions using reference peptides or other stringent analytical methods. For example in the elegant neuron differentiation system they use.
- The authors should demonstrate that CTAP modifications on substrates are necessary for recognition by FBXO31 under physiological conditions.
- The simplest experiment to establish the link appears to be at their fingertips. In figure 4d-f they use IP-MS or ABP to identify FBXO31 substrates that are dependent on CTAP. From the MS data, they should be able to identify the CTAP bearing peptides present under these conditions. Using isotope labeled reference peptides, they should establish that H₂O₂ treatment in fact increases the abundance of the CTAP bearing peptides and that this correlates with recognition. The use of reference peptides would establish the presence of CTAP peptides under baseline conditions (e.g. in differentiating neurons) and would allow to establish the quantitative link to recognition.
- The fact that what appears to be the key finding of this paper, a previously overlooked modification renders protein unstable, is not accompanied by a main figure establishing the existence of these modifications under physiological conditions.

Minor points:

- Figure 2d would benefit from a specificity control by titrating unlabeled peptide
- Making protein can't be too hard given X-Ray structure

We thank the reviewers for the very encouraging feedback and for the thorough review of our manuscript. We are also grateful for the detailed suggestions on further improvements.

The major lines of commentary from all referees were the following:

- (1) Validate the proposed mechanism of CTAP formation by oxidative cleavage.
- (2) Establish FBXO31 activity under endogenous forms of oxidative stress.
- (3) Explore FBXO31's role in cell physiology.
- (4) Dissect neosubstrate recognition by FBXO31(D334N).
- (5) Rigorously validate and document proteomics experiments.

Our extensively revised manuscript now addresses these and other points with new experiments and analyses. They include:

- (1) Oxidative damage is sufficient to trigger CTAP formation by protein fragmentation. This is true for CTAPs previously identified in human tissues (hemoglobin) and for FBXO31 clients identified by IP-MS (tRNA ligase complex).
- (2) FBXO31 binds endogenous clients when the cellular redox balance is shifted to oxidative conditions. This is true when blocking the removal of endogenously produced reactive oxygen species or when stimulating their production.
- (3) Oxidative stress triggers activation of FBXO31, as determined by cullin ring ligase activity profiling method. Loss of FBXO31 leads to a neuron-specific response, consistent with neuronal phenotypes of human pathogenic FBXO31 mutations.
- (4) FBXO31(D334N) recognizes substrates by a new C-terminal degron sequence motif, which is necessary and sufficient to trigger mutant-induced protein degradation.
- (5) Initial proteomics results are validated by orthogonal methods and analyses. The revised manuscript also includes a thorough documentation of methods and search results and links to a raw data repository.

These and additional changes are highlighted in blue in the main text. Detailed point-by-point responses can be found below. We hope that the reviewers share our enthusiasm about these findings and thank them for their ideas to further strengthen this study.

Muhar, Farnung, et al use a GFP reporter assay to display a set of chemical modifications as candidate forms of protein damage. They found that proteins bearing C-terminal amides (CTAPs) were degraded in a ubiquitin- and proteasome-dependent manner and use a genome-wide CRISPR screen to discover FBXO31 as the E3 ligase recognizing this motif. The interaction is rigorously characterized against a peptide library to reveal the C-terminal amide as the minimal structural determinant with some preference for hydrophobic amino acids. C-terminal amides may form by oxidative cleavage. In support of this mechanism, the authors use IP-MS to reveal a set of proteins with increased binding to FBXO31 after hydrogen peroxide treatment and show that two of these potential substrates (GLUL and AARS1) are co-immunoprecipitated and the C-terminal amide sequence serves as substrates for FBXO31 when installed to the GFP reporter. Finally, the authors closely evaluated FBXO31(D334N), a mutant associated with cerebral-palsy, and found that the mutation abolishes CTAP-directed activity and gains the ability to target certain neosubstrates such as SUGT1 for ubiquitination. Expression of FBXO31(D334N) was less tolerated in co-culture compared to that of WT, which may relate to the toxicity of this mutant.

Overall, this manuscript reports a novel contribution to our understanding of the role of non-enzymatic chemical modifications, with implications in protein homeostasis, E3 ligase biology, and potentially the eventual therapeutic utilization of these observations. The evidence in favor of FBXO31 recognition of C-terminal amides is well-grounded and rigorously demonstrated, which affords a very interesting explanation for a dominant negative mutation associated with cerebral-palsy. However, the crux of this study lies in the connection of FBXO31 in targeting endogenous CTAPs (Figure 4). While the authors provide initial evidence in support of FBXO31 regulation of endogenous CTAPs, additional studies that establish the cellular occurrence of CTAPs and that these modifications on endogenous substrates are recognized by FBXO31 will strengthen the key conclusions of this work.

We thank Reviewer #1 for the thorough and very encouraging review. As described in more detail below, we have addressed the cellular significance in several ways, including finding FBXO31 as one of the two top ligases activated in response to oxidative stress, selective binding of FBXO31 to endogenous targets in response to elevated endogenous ROS, and establishing that previously identified FBXO31 clients form CTAPs upon oxidative damage.

Major points

1. Establishing that an endogenous protein substrate with a C-terminal amide modification is a bona fide substrate of the FBXO31 is critical to support the evidence from co-IP experiments and the GFP reporter assay already provided. At a minimum demonstration of FBXO31 substrate recognition beyond engineering GFP is necessary. The demonstration of a bona fide substrate may be achieved by extending data with AARS1. Is the truncated AARS1 that binds to FBXO31 a ubiquitylated substrate in cells or in vitro? Does mutation of the oxidative cleavage sites on AARS1 alter formation of this damage event, such that it is no longer generated and therefore rescued as a substrate for

FBXO31? Is there are structural rationale and/or conservation of the AARS1 site conserved and potentially a signal site for oxidative damage?

We initially attempted to perform reductionist *in vitro* experiments with AARS1 as a model protein. However, AARS1 was unstable when overexpressed from cDNA, precluding analysis of stress-induced cleavage. We therefore proceeded to validate CTAP formation for another FBXO31 client, the tRNA-ligase complex, which is highly prone to oxidation due to its affinity for the Fenton-like catalyst Cu(II) (Asanović *et al.* 2021, *Mol Cell*). We recombinantly expressed and purified the tRNA ligase complex and tested whether oxidizing conditions are sufficient to induce CTAP formation. Mass spectrometry revealed that C-terminally amidated cleavage fragments form for all three core subunits of the complex (RTCB, RTRAF, DDX1) specifically in oxidizing conditions and in the presence of Cu(II) (new **Extended Data Fig. 7e, f**).

To test the involvement of CTAP formation in FBXO31's recognition of additional endogenous cellular clients, we asked whether changes in cellular redox homeostasis that lead to increased ROS induce targeting. Indeed, inhibition of the ROS detoxifying enzyme thioredoxin reductase with auranofin triggered FBXO31 binding to both AARS1 and GLUL (new **Fig. 4h**). The same was true for stimulating the production of endogenous ROS with menadione. Hence, multiple orthogonal oxidizing conditions (H₂O₂, auranofin, and menadione) all induce FBXO31 to bind the endogenous substrates we identified by mass spectrometry.

To test whether there are favored sites for CTAP formation, we analyzed sequences surrounding all cleavage sites identified in human tissue proteomes. Matching the proposed model for stochastic backbone fragmentation, CTAPs formed at virtually any residue, while also revealing some more labile sites, such as residues followed by a cysteine (new **Extended Data Fig. 7d**). We include a discussion of these findings in the revised manuscript.

*2. The formation of CTAPs is proposed to occur by an oxidative cleavage mechanism with H₂O₂, which is a non-specific form of protein damage that would preferentially target many other oxidizable sites in cells. It is very possible that the extent of this form of protein damage has been overlooked to date, adding to the novelty of this work. Additional understanding of C-terminal amide formation on endogenous substrates and their response to FBXO31 knockdown would provide context for the extent of this form of protein damage, further establish these modifications as responsive to FBXO31, and show that the CTAP fragments accumulate in its absence. For instance, measurement of CTAP formation/FBXO recognition on specific proteins *in vitro* or comparison of modification events in FBXO31 KO cells (versus WT) after oxidative damage would establish that these modifications accumulate in response to loss of FBXO31.*

As the reviewer points out, our analysis of tissue proteomes suggests that CTAP formation is a common event that remained overlooked to date. To validate that the reported spectra are *bona fide* CTAPs, we successfully reconstituted *in vitro* fragmentation for hemoglobin, a protein found to be C-terminally amidated across all analyzed tissues. In addition, hemoglobin's heme groups contain Fe(II) ions which can catalyze the formation of hydroxyl radicals and drive CTAP formation.

Mass spectrometry of the purified hemoglobin heterotetramer found extensive amidating cleavage for both hemoglobin subunits, which increased upon H₂O₂ treatment in a dose-dependent manner (new **Fig. 4b**, new **Extended Data Fig. 7a,b**). High-coverage mapping of amidated fragments revealed that cleavage occurs across both proteins at 75 different sites C-terminally to 14 different amino acids (new **Extended Data Fig. 7c**). These included most of the CTAP sites identified in human tissues. As described above, we similarly found that oxidizing conditions are sufficient to induce CTAP formation for the FBXO31 clients RTCB, RTRAF and DDX1 (new **Extended Data Fig. 7d, e**).

As an orthogonal approach, we profiled responses to oxidative stress in cells using a recently developed method for profiling active culling ring ligase complexes (*Henneberg et al. 2023, Nature Chemical Biology*). This method identified SCF/FBXO31 as the second most highly induced culling ring ligase in an erythroid cell line (new **Fig. 4c**). Among the highest scoring oxidation dependent CRL interactors were all hemoglobin subunits expressed in these cells (HBA, HBE, HBQ and HBZ) and several FBXO31 clients we previously identified by mass spectrometry, including AARS1 and GLUL (new **Fig. 4d**).

Together, these results demonstrate that oxidative protein damage is sufficient to induce FBXO31 activation and CTAP formation among its clients. They also confirm that the CTAPs found in human tissue proteomes can result from oxidative protein damage. This distributed form of damage is particularly hard to detect by mass spectrometry which could explain how it remained overlooked to date, as the reviewer pointed out.

3. CTAP formation is supported by performing proteomic analysis of public proteome datasets for neo-C-termini. Tables reporting these peptides need to be provided. Do the authors observe a consensus sequence in human tissue proteomics that aligns with the preferential recognition sequence for FBXO31? Do the proteins in these data align with those that are observed by the authors following oxidative damage? Does analysis of a dataset following oxidative damage (be it publicly available or from the authors) possess an increase in C-terminal amide modifications?

We have now provided an extensive list of all CTAP Peptides identified across all previously published and newly generated MS datasets for easy re-analysis by others (**Supplementary Data Table 3**). In addition, we submitted all raw data to the PRIDE database to enable re-analysis by others.

The revised manuscript now includes the referee's suggested analysis of cleavage sites for CTAPs detected in tissue proteomes. We observe no consensus sequence surrounding cleavage sites, in line with our model of unselective backbone oxidation (new **Extended Data Fig. 7d**). As described above, we also identified an increase in C-terminal amide modifications following oxidative damage of hemoglobin *in vitro* (new **Fig. 4b**, new **Extended Data Fig. 7a,b**). The majority of CTAP fragments identified in human tissues aligned with sites of CTAP formation found *in vitro* (new **Extended Data Fig. 7c**).

4. The pooled peptide input library should be added as a supplementary table. For the D334N selection, it is clear the C-terminal amide pool has been lost, but why not gain the

1C-terminal carboxylic acid pool, given that many substrates are gained in the IP-MS and global proteomics?

As suggested, we have now added a table of all input peptides detected for each IP to the table of *in vitro*-bound C-termini (new **Supplementary Table 2**). Given that peptides are identical throughout most positions our MS approach could not quantify same peptides in the two independent MS runs for many cases. We therefore refrain from statements on individual sequences and limit our interpretation to the aggregate analysis of amino acid preferences (**Extended Data Figure 4b-e**).

For the D334N mutant, we reported 21 proteins as significantly bound by IP-MS and degraded by whole-cell proteomics. Based on these proteins' putative consensus sequence, we carried out an extensive set of mutagenesis experiments to define the neosubstrate preferences of the D334N mutant (new **Fig. 5e**, new **Extended Data Fig. 10b-f**). We found that a C-terminal $-[K/R]-X-\Phi-COOH$ motif is sufficient to confer degradation by FBXO31(D334N). This relative selectivity contrasts with the broad targeting of C-terminal amides by the wildtype enzyme.

These results could explain why the peptide interaction screen did not identify mutant-specific clients. While the screen for FBXO31(D334N) correctly reported loss of CTAP binding, we therefore chose to omit this data to avoid misinterpretation.

5. For Figure 3f, please add rescue with FBXO31 KO or mutation.

We performed the rescue experiment in the background of FBXO31 knockdown by CRISPR interference. The background showed somewhat higher degradation of all reporters relative to the parental line, but degradation of a C-terminal amide was rescued by knockdown of FBXO31 (new **Extended Data Fig. 4g**).

6. Does HA-FBXO31(deltaFBox) still ligand to peptide substrates similarly to the full length FBXO31?

Yes. We directly compared binding to a model CTAP in cells (**Extended Data Fig. 9c**), finding that FBox deletion does not affect client binding by FBXO31. Also, for the D334N mutant we validated that clients identified by IP of the Δ FBox form were degraded in a C-terminal degenon reporter assay when the full-length mutant was expressed (new **Fig. 5a, b, and d**).

Minor points

1. The oxidative cleavage mechanism will generate alongside a potentially highly toxic neo-N-terminal aldehyde (in the case of glycine) or ketone (other amino acids). The neo-N-terminal fragment would be potentially more damaging to the cell as it is the oxidized fragment. Are these neo-N-termini observed in the datasets and is there a conserved cellular response known or observed? Relevant to the CTAP itself, how long lived is the C-terminal amide when formed, how quickly can it be hydrolyzed? Why would the cell conserve an approach to get rid of something that can hydrolyze? Please discuss.

We agree that it is possible that the complementing N-terminal ketone- or aldehyde-bearing fragments may also be degraded by an independent mechanism. Preliminary analysis found spectra matching carbonylated N-termini following *in vitro* oxidation of hemoglobin (data not shown). These could potentially be read by other ubiquitin ligases, such as the three additional cullin ring ligases we report as activated by oxidative stress (new Fig. 4d). However, further method development will be necessary to model these fragments. We therefore chose to avoid over-claiming additional putative N-terminal chemical degrons and would prefer to explore these potential new biologies in follow-up work.

C-terminal amides are frequently found on secreted peptides and are introduced in synthetic peptides to increase their stability in human plasma. Using recombinantly produced amide-bearing reporters, we confirmed that C-terminal amides on proteins and peptides were stable in solution at 37°C for up to 24h (Reviewer Fig. 1). This makes terminal amides particularly suitable as an intracellular signal. We now highlighted this feature in the updated discussion section.

Reviewer Fig. 1: C-terminal amides are highly stable *in vitro*. Peptides were incubated for 0 or 24h at 37°C and peptide masses were measured by ESI-MS followed by MaxEnt deconvolution of spectra to obtain the average molecular mass. Observed masses were within 0.5 Da of the theoretical average molecular masses and remained unchanged over 24h. Theoretical Mw of the Cys-dimerization products are 4715.5 Da and 4573.3 Da respectively.

2. In Line 389–390, the authors state that “C-terminal amidation could render cyclin D1 an FBXO31 client in other cell types or following stress...”. Have the authors tried to validate that oxidative stress can lead to Cyclin D1 interaction with WT but not D334N, similar to

Figure 4d? This would link the previously reported biology of FBXO31 to its regulation of oxidation-induced CTAPs.

Despite extensive testing in both the original manuscript and the revision, we found no evidence of FBXO31 activity towards CyclinD1. This includes lack of binding by FBXO31 under basal or oxidatively stressed conditions by IP-MS (**Supplementary Tables 5 and 6**) and CRL-activity-based profiling detects no enhanced binding of CyclinD1 to CRLs following H₂O₂ treatment (new **Supplementary Table 4**). However, we did not test whether Cyclin D1 is amidated under these conditions. To be prudent, we avoid speculations about CyclinD1 as a context-dependent substrate in the updated discussion.

CyclinD1 has also been suggested as a target of the D334N mutant. However, we find no evidence for non-amidated CyclinD1 as a substrate of the D334N mutant by co-IP (**Extended Data Fig. 9c**), IP-MS (**Supplementary Table 8**), fluorescence polarization assays with the proposed degron (**Extended Data Fig. 9a**), global proteomics by TMT-MS (**Supplementary Table 9**) and new *in-cell* degron profiling (new **Fig. 5d**). Instead, our data strongly indicates that the pathology of the D334N mutation is linked to its neosubstrate activity. We highlight this conclusion in the updated manuscript.

3. The LC-MS/MS acquisition settings for all proteomics experiments are missing and should be elaborated in the Methods section.

We apologize for the oversight and have updated the methods section to provide more details on MS/MS-acquisition. Given the number of different MS datasets, the new manuscript also refers to the PRIDE database upload which contains more detailed information on each individual experiment.

4. Please specify the protease used for digestion in the Methods section “Expression proteomics by tandem mass tag (TMT) MS”. The reference for automated SP3 workflow should also be included.

We apologize for the oversight. The methods section has been updated accordingly.

Substrate specificity for E3 ubiquitin ligases is at the heart of ubiquitin signaling. While early work had suggested that E3 ligases mainly recognize linear degrons, which might in some cases be modified by phosphorylation or related signals, recent work has shown that E3 ligases can have a much broader substrate specificity. This included three-dimensional degrons as well as degrons modified at their C-termini. However, very few E3 ligases with specificity for C-terminal modifications have been identified (i.e. CRBN), and general platforms to find such enzymes have not been developed.

In this study, Corn et al. develop a system to screen for E3 ligases that identify substrates carrying specific posttranslational modifications. This work thus lays the groundwork for much larger studies interrogating PTM-driven protein degradation, which is important by itself. Using their new approach, the authors found that proteins carrying C-terminal amides (CTAPs), potentially generated by oxidative cleavage of the peptide backbone, are unstable and cleared by the ubiquitin-proteasome system. They used CRISPR screens to identify SCF-FBXO31 as the E3 ligase responsible for CTAP clearance. The data documenting the role of SCF-FBXO31 in CTAP clearance includes cellular approaches, such as rescue of knockout, and reconstitution of binding and ubiquitylation – it shows readers how substrate-enzyme interactions should always be documented for the ubiquitin field! They next identified the degron for FBXO31 – showing a requirement for a C-terminal amide, but not much more, as expected for a broadly acting quality control enzyme. They suggest that CTAPs are formed under oxidative stress conditions, although experiments were performed under harsh H₂O₂ treatment. A disease-linked mutation in FBXO31 leads to degradation of neo-substrates that are not normally turned over through FBXO31, suggesting that the striking specificity of FBXO31 is critical for human physiology.

Overall, this is a spectacular paper with respect to the characterization of an E3 ligase with an interesting substrate specificity for CTAPs. This paper sets the stage for the discovery of many E3 ligases with specificity for modified degrons. The study lacks, however, physiological significance, as it is neither clear whether specific conditions generate CTAPs, why CTAPs would need to be eliminated from cells (i.e. if they are so rare that they can only be discovered upon harsh oxidative stress), and whether this pathway provides cells with survival benefits. These issues need to be addressed by new experiments, but I strongly believe that the authors could accomplish this fairly simply (as outlined below) to turn this study into a landmark paper fully worth of publication in Nature.

We thank the reviewer for the very encouraging feedback, which prompted us to validate and extend our previous findings in the context of endogenous ROS-formation, endogenous clients and endogenously expressed FBXO31.

Major issues:

- 1. Critical to the relevance of this study, CTAPs must be formed in vivo under conditions that are experienced by cells within an organism, and FBXO31 must recognize and clear such cleavage products. The experiments showing AARS1 cleavage upon treatment of*

cells with 200 μ M H₂O₂ are not sufficient to address this question. For a paper potentially defining a new research field (which I believe this manuscript has the potential to accomplish), I would expect an IP/MS experiment showing binding to an endogenously cleaved protein, under conditions that can be experienced in an organism (i.e. not in response to H₂O₂). Maybe the authors could produce oxidative stress upon depletion of SOD1 or related ROS-detoxifying enzymes, as seen in ALS patients? Maybe the authors could deplete metallothionein, the protein that is most strongly correlating with FBXO31 in DepMap and that protects cells from oxidative stress? Or they could look in cells of the immune system after pathogen invasion, i.e. conditions that produce ROS through NOX enzymes? Seeing that an endogenous CTAP, as identified through the MS profile, would be bound to FBXO31 under such conditions would massively increase the impact of this study. Knowing more about such substrates might also inform the authors about why CTAP proteins might be dangerous from cells and need to be removed immediately (in other words, why could cells not tolerate such proteins or remove them through complementary pathways, such as those detecting unfolding)?

As suggested, we tested whether changes in cellular redox homeostasis induce targeting of endogenous clients by FBXO31. As a model, we studied recognition of AARS1 and GLUL, which are recognized by FBXO31 under oxidative stress via its substrate binding pocket (**Fig. 4g**). Inhibition of the ROS detoxifying enzyme thioredoxin reductase with auranofin triggered FBXO31 binding to both endogenous clients within 2h in HEK293T cells (new **Fig. 4h**). The same was true for stimulating the production of endogenous ROS with menadione. In sum, cellular FBXO31 binding to endogenous targets is triggered by three orthogonal conditions (H₂O₂, auranofin, and menadione) that are either overtly oxidative or increase cellular ROS.

As an additional approach, we performed activity-based profiling of active culling ring ligase (CRL) complexes in K562 cells with and without H₂O₂ treatment. This experiment identified endogenously expressed SCF/FBXO31 as the second most highly induced CRL (new **Fig. 4c**) and confirmed AARS1 and GLUL as oxidation-induced CRL clients (new **Fig. 4d**).

In the initial manuscript, we also reported that CTAPs form *in vivo* in healthy human tissues based on re-analysis of public proteomics data. To validate that oxidative damage can directly induce CTAPs, we measured their formation *in vitro*. Hemoglobin was identified among CTAPs in all analyzed tissues, and new mass spectrometry experiments revealed *in vitro* cleavage and amidation of both hemoglobin subunits purified from human blood that was exacerbated by H₂O₂ treatment in a dose-dependent manner (new **Fig. 4b**, new **Extended Data Fig. 7a,b**). Notably, the four hemoglobin subunits expressed in K562 cells were also among oxidation induced CRL clients in the affinity-based profiling experiments (new **Fig. 4d**).

We note that CTAP formation by fragmentation is unselective, affecting at least 50 out of 147 amino acids in HBB (new **Extended Data Fig. 7c**, new **Supplementary Table 3**). MS-based detection of such a distributed mark therefore requires deep profiling of purified proteins (as shown for HBA, HBB, DDX1, RTCB and RTRAF; new **Extended Data Fig. 7**) or strong pre-enrichment of FBXO31 clients as shown previously for AARS1 (**Fig. 4i**).

2. The physiological relevance of FBXO31 is unclear. Do FBXO31-deletion cells die earlier than wildtype cells when exposed to oxidative stress or any other stress condition that might result in the cleavage of peptide backbones? This should be addressed through genetic interactions, not by treating cells with chemicals such as H₂O₂. As the principle of C-terminal modifications as targeting signals for E3 ligases has been established through Christina Woo's work on CRBN, I believe it is critical that evidence for physiological relevance is provided for this paper on FBXO31.

We profiled cellular responses to FBXO31 knockdown and found no proliferative disadvantage upon co-depletion of metallothioneins (MT1A, MT1E, MT1X) or Catalase (data not shown). However, short-term cell survival is a crude readout, especially under oxidative stress where damage to DNA and RNA prevail irrespective of FBXO31 function. We therefore tested more molecularly precise responses to FBXO31 knockdown by RNA-seq.

We were inspired by the observation that FBXO31 is most highly expressed in neurons and FBXO31 mutations in humans are associated with neuronal phenotypes. We therefore developed a new CRISPRi neuronal progenitor cell (NPC) line and tested the effect of FBXO31 knockdown in NPCs, differentiated isogenic neurons, and unrelated HEK293T cells (new **Extended Data Fig. 8**) Strikingly we found that FBXO31 loss selectively affects the neuronal transcriptome. Even though highly proliferative HEK293T cells express lower levels of FBXO31, we found that there was no detectable short-term phenotype of knockdown under basal conditions (new **Extended Data Fig. 8a**). In NPCs, we observed deregulation of a small number of transcripts (new **Extended Data Fig. 8c**). In differentiated mature neurons, we found a much stronger transcriptional phenotype (new **Extended Data Fig. 8d**), deregulation of neuronal and oligodendrocyte pathways and key transcripts such as NUROG2, MAPT and TUBB4A (new **Supplementary Table 7**). A database search found this signature to most closely match changes induced by engineering ALS mutations in *in vitro*-derived somatic motor neurons (new **Extended Data Fig. 8e, f**).

These results and clinical human phenotypes of FBXO31 loss suggest that, despite FBXO31's ubiquitous activity, the high metabolic activity and non-proliferative nature of neurons might render them particularly sensitive to its loss. This would be in line with CTAP accumulation in highly metabolically active yet non-proliferative tissues. This does not rule out longer-term phenotypes of chronic FBXO31 disruption in other cell types. Further understanding the role of FBXO31 in acute and long-term cell homeostasis in various tissues will be the subject of dedicated future studies that may include mouse models.

In this manuscript by Muhar et al., the authors describe the discovery that the SCF substrate receptor FBXO31 recognizes c-terminal amide bearing peptides (which they call CTAP) and that this leads to degradation of a protein reporter of GFP/RFP conjugated to such a modified peptide. The authors further report that such c-terminal modifications are the result of oxidative protein damage and cleared by FBXO31 to maintain cellular homeostasis.

The initial identification of CTAP followed a clever screen using a semi-synthetic protein library introduced into cells, which identified CTAP as the only modification leading to protein degradation. They next went on to identify the ligase machinery necessary for reporter degradation and identified and validated FBXO31. Using recombinant protein, peptide synthesis and biochemical and biophysical methods they establish that FBXO31 exhibits high affinity for amide bearing c-termini but not unmodified, with little specificity for the preceding amino acids. They also identify a mutation in the putative binding pocket of FBXO31 that prevents binding of CTAP peptides. To answer the most critical question, whether these CTAP modifications exist in vivo and whether FBXO31 plays a role in clearance, the authors first look at published global mass spectrometry datasets and identify the modification on truncated proteins albeit little detail is provided. They further turn to affinity mass spectrometry (IP-MS) to identify substrates of FBXO31 in response to H₂O₂ treatment, which leads them to identify AARS1 and GLUL amongst others. They lastly look at a patient mutation in FBXO31, which surprisingly instead of leading to impaired clearance of damaged proteins leads to degradation of a large set of specific substrates. They explain this by the D334N mutation being a gain of function mutation. While the data characterizing the interaction of FBXO31 with CTAP peptides as well as the CTAP peptide serving as a degron on reporter proteins is well supported by data, significant open questions remain about the endogenous role of FBXO31. The endogenous role needs to be either toned down or clear evidence for the existence and physiological relevance of CTAP should be presented, which would make this work relevant to a broad audience.

We thank Reviewer 3 for the detailed review and suggestions. In response, we extended our study by profiling the activity of endogenously expressed FBXO31, identifying endogenous CTAPs and FBXO31 substrates, and cellular responses to FBXO31 loss. We furthermore performed the suggested validations of D334N client recognition. These experiments helped strengthen the manuscript, as detailed below.

Major points:

The most important question for this work is whether CTAP bearing peptides exist in vivo. The authors address this pre-dominantly through re-evaluation of public proteomics datasets (EDFig 6), however, the data presented is not convincing. Amidation is common (+0.9840 mass shift) and the presented single spectrum is not sufficient to convince me these are c-terminal modifications. The gold standard would be to use reference peptides to calibrate retention times and look at both retention time and MS fragmentation. Where do these peptides fall in the confidence space? These experiments are critical and should be done properly and presented as main figure.

A more thorough validation of the reported *in vivo* CTAPs is important and is now addressed in the revised manuscript. As an alternative to the suggested targeted MS approach, we performed several alternative analyses and experiments that we believe equally benchmark our identification of CTAPs.

(1) To minimize signal from low-quality spectra, we applied stringent quality filters, only quantifying peptides with $FDR \leq 1\%$ for whole proteome searches and $FDR \leq 0.1\%$ for purified proteins. For reference, we provide all primary search results for CTAPs in **Supplementary Table 3**, and complete search results raw data via the PRIDE database.

(2) For two top-scoring *in vivo* CTAPs (HBA and HBB), we now experimentally validate CTAP formation by α -amidating cleavage of purified proteins under *in vitro* oxidizing conditions. The majority of the reported *in vivo* fragments were also formed *in vitro* (new **Extended Data Fig. 7c**).

(3) In public datasets, C-terminal amidation (-0.984 Da) is an extremely rare event among tryptic spectra (0.03-0.2 %) but is up to 289-fold enriched among spectra with non-tryptic C-termini (new **Extended Data Fig. 6c**). And vice versa, while non-enzymatic C-termini are generally rare, they represent the majority of peptides among spectra with a C-terminal -0.984 Da shift (**Extended Data Fig. 6b**). These data support that C-terminal amidation is primarily a product of protein fragmentation.

(4) No other mass shift of this magnitude could be observed for peptide C-termini upon *in vitro* oxidative CTAP formation of purified hemoglobin (new **Extended Data Figure 7b**), ruling out technical error as source of the reported mass shifts.

(5) Amidation has not been reported to occur on protein side chains. The only other known modification that could lead to a similar mass shift is oxidation of lysine to amino adipic semialdehyde. However, the majority of CTAP peptides we identified *in vivo* do not contain Lys (61%).

Taken together, these new experiments and analyses make us confident that our proteomic definition of CTAPs recovers true amidated neo-C-termini.

In addition to the above mentioned re-analysis of human tissue data, the authors also look for these modifications in FBXO31 knockout cells using FBXO31 as a bait under basal and H2O2 conditions. This leads them to identify additional putative substrates. There are some critical controls missing here that again weaken the data: o The authors should repeat the same experiments in FBXO31 wt cells as these should clear all the peptides and therefore serve as the better positive control.

As suggested by the reviewer, we performed the IP-MS analysis of full-length FBXO31 under basal conditions as a first test. As expected, we identified the expected SCF complex members (CUL1, SKP1, RBX1). And as anticipated by the referee, we also found a loss of clients such as the tRNA-ligase complex, consistent with full-length FBXO31 now ubiquitinating and degrading CTAP modified clients (see **Reviewer Figure 3.1** below).

Reviewer Fig. 3.1 – Full-length FBXO31 IP validates activity towards clients. Left: Comparison of co-IP-MS signal for enzymatically inactive FBXO31 (dFBox) and full length FBXO31. Axis values represent the mean of three replicates expressed as log₂-transformed spectral counts. One pseudocount was added to display proteins absent from either condition. Blue dots highlight the interactors reported in this manuscript in unstressed cells. Right: Ratio of signal for FBXO31 clients by IP of FBXO31(dFBox) versus full-length FBXO31.

Since receiving the reviewer's comments, a novel method for activity-based profiling of cullin ring ligases (CRL-ABP) was published (Henneberg et al. Nature Biotechnology, 2023). This CRL-ABP method provides a direct measure of CRL activity and their targets using full-length, endogenous cullin components. We therefore used CRL-ABP to monitor ubiquitin ligase responses to oxidative stress. We found that endogenous FBXO31 is one of the most highly activated CRL components upon oxidative stress (new Fig. 4c). CRL-ABP also independently validated the IP-MS identified FBXO31 clients AARS1 and GLUL as oxidation-dependent CRL targets (Fig. 4d). CRL-ABP in an erythroid cell background also found oxidation-induced targeting of hemoglobins, which matches their identification as CTAPs in human tissues. We furthermore validated that hemoglobins undergo α -amidating cleavage *in vitro* to form CTAPs (new Fig. 4b, new Extended Data Fig. 7a-c). Altogether, our data strongly support an endogenous role for FBXO31 in recognizing and clearing CTAPs that form in response to oxidative stress.

o The authors should present the peptide spectra supporting the presence of modifications and retention times / MS trace.

We have now added the primary results of peptide identification for all reported CTAPs in the supplementary materials to the paper including retention time and modifications (new **Supplementary Table 3**). In addition, raw data will be accessible through the PRIDE database.

o How do the authors rule out secondary effects due to H₂O₂ treatment leading to some of the changes?

At the cellular level, we can now rule out the the reported client interaction is specific to H₂O₂ treatment. Preventing the removal of endogenously produced reactive oxygen species,

or stimulating their production triggered rapid client recognition by FBXO31, phenocopying H₂O₂ treatment (new **Fig. 4h**).

As described above, we have now performed *in vitro* experiments on purified proteins, demonstrating that oxidative conditions alone are sufficient to trigger CTAP formation by α -amidating cleavage. We first identified widespread CTAP formation after *in vitro* oxidative treatment of hemoglobin, matching identification of hemoglobin CTAPs in tissue proteome searches (new **Fig. 4b**, new **Extended Data Fig. 7a-c and d-e**). Hemoglobin is known to be prone to oxidative damage with four Fe(II) ions in its heme group serving as Fenton catalysts which produce hydroxyl radicals that also trigger alpha amidating cleavage.

We also verified *in vitro* CTAP formation in the presence of Cu(II) and H₂O₂ across recombinantly expressed and purified components of the tRNA ligase complex (RTCB, DDX1 and RTRAF) (new **Extended Data Fig. 7e, f**). The tRNA ligase complex is known to be exceptionally sensitive to oxidative stress given its affinity for Cu(II) which also forms hydroxyl radicals in Fenton-like reactions (Asanović et al. 2021, Mol. Cell). Altogether, these *in vitro* results make us confident that alpha-amidating backbone cleavage is a direct effect of oxidative protein damage.

- Fig 4e. Proper analytical data needs to be shown in suppl/ED to support the statement. What fraction of AARS1/GLUL are cleaved at these sites?

While the CTAP formation sites we report may be more labile than others, proteins can in principle fragment at any position. This makes it impossible to capture all cleavage events, especially from complex samples like tissues or cells, since the signal is spread out across the entire proteome. We therefore instead performed extensive CTAP mapping on purified hemoglobin which provided deep coverage across two proteins (HBA and HBB), showing that up to 27.8% of identified peptides are CTAP fragments (new **Extended data fig. 7a**). We acknowledge that it is difficult to estimate the extent of CTAP formation across the proteome in a cellular context, and refrain from making exact statements on the percentage of damaged proteins in the manuscript.

- The findings on the D334N mutation are interesting, but could be interpreted in multiple different ways and raises more questions than answers. How are these neo-substrates recognized? It has to be the same pocket, does it change peptide specificity? What do these neo-substrates have in common?

We reported 21 proteins as D334N substrates due to significant binding by IP-MS and degradation by whole-cell proteomics. Alignment of the C-termini of these putative neosubstrates shows a common [K/R]-X- Φ -COOH motif (new **Extended Data Fig. 10b**). The C-terminal sequences of several neosubstrates was sufficient to induce degradation of a reporter protein by FBXO31(D334N) but not for wild type FBXO31 (new **Fig. 5d**, new **Extended Data Figure 10c, d**). Using systematic mutational analysis, we found that the the [K/R]-X- Φ -COOH motif is necessary for FBXO31(D334N) degradation in two C-terminal sequence contexts (new **Fig. 5e**, new **Extended Data Fig. 10e, f**). This relative selectivity contrasts with the broad targeting of C-terminal amides by the wildtype enzyme.

Could CTAP be an inhibitor for the ligase rather than being a substrate itself?

This is an excellent question. To address it, we performed a mixed *in vitro* ubiquitylation experiment, in which we provided FBXO31 with two model substrates, one of which carried a terminal amide and one with an unmodified C-terminus (**Fig. 2e**). This experiment showed that FBXO31 is completely specific for ubiquitinating the terminally amidated substrate. When we swapped the C-terminal amide to the second substrate, FBXO31 now ubiquitinated only this substrate. We therefore conclude that terminal amides directly mark FBXO31 substrates and can rule out any trans activation or inhibition by CTAPs.

Following on this, an obvious key question is what do the other mutations do in IP-MS/Global proteomics? Do they lead to similar gain of function?

For the initial manuscript, we designed several binding pocket mutations, which we now confirmed to be defective for CTAP binding and degradation (new **Fig. 3d-f**). We found that they do not show similar phenotypes as D334N. Expressing all of the loss-of-function mutations is well tolerated by cells, while continuous D334N expression is not. In a rapidly inducible FBXO31(D334N) expression system, we found full-length FBXO31(D334N) is acutely toxic.

As the reviewer suggested, we now also performed the suggested IP-MS experiment for one of the CTAP binding-deficient mutants, FBXO31(T343V). In a pilot experiment with single replicates per condition, the mutation conferred no neosubstrate binding under basal conditions. Following H₂O₂ treatment T343V showed reduced binding to wt clients including AARS1 and GLUL (**Reviewer Fig. 3.2**). Since the designed mutations are not reported for human patients and showed the expected loss-of-function, we decided not to study them in more detail. If the referee feels like this finding is essential to the paper, we could perform the additional replicate IP-MS runs

Reviewer Fig. 3: No pervasive binding of binding pocket mutant FBXO321(T343V). Pilot IP-MS experiments in FBXO31 knockout HEK293T cells comparing binding to the wildtype substrate pocket (HA-FBXO31(ΔFbox)) to the loss-of-function mutant (HA-FBXO31(ΔFbox, T343V)). Parallel experiments were performed in untreated cells and cells challenged with a pulse of H₂O₂ (20 min, 200 μM). Axes show log₂ of total spectral counts (TSC).

Additional points:

The authors should provide FP/binding data on all the mutants used (Y309A, D334N, I337D, T343V) to CTAP and control peptides

FBXO31 is exceptionally hard to express recombinantly, as confirmed by multiple collaborating groups, and it took a great deal of time to obtain enough FBXO31 for the experiments described in the first submission. To be able to scale to multiple mutations, we therefore performed the requested interaction study in cells by co-IP. Indeed, all of the loss-of-function mutations abrogate CTAP binding by FBXO31 in cells (**Fig. 3e**). Importantly, the mutations do not affect assembly of the SCF/FBXO31 ligase complex, since co-IP with CUL1 remained unchanged.

Because we had studied the disease mutant D334N in detail, we invested a great deal of effort in purifying the D334N mutant. As suggested by the referee, fluorescence polarization confirmed that this mutant no longer binds CTAPs (**Extended data fig. 9a, b**). D334N also does not bind CTAPs by cellular co-IP (**Extended data fig. 9c**).

It is interesting that none of the other modifications leads to degradation, does that argue against PTM being broadly responsible for regulating protein stability?

We were also intrigued to find that only C-terminal amidation triggered degradation. However, we have so far only tested five PTMs out of >700 known PTMs, and so would prefer not to speculate whether or not this activity is widespread. However, as suggested by all referees, we hope that our work stimulates widespread efforts to map the effect of the many “unusual” PTMs on protein stability.

Response to the Reviewers:

We thank all three reviewers once again for their time, effort and helpful feedback.

Referee #1 (Remarks to the Author):

I enjoyed reading the updates to this manuscript. The authors have clearly extended their findings to thoroughly demonstrate the overlooked formation of CTAPs as a form of oxidative damage in vitro and in multiple cell types that is then recognized and removed by FBXO31. The gain of function FBXO31 D334N mutation is also remarkable. I only noted a few typos in the new figures that can be readily addressed by careful review. I look forward to sharing this work with my lab.

We thank Referee #1 for the encouraging words and detailed feedback. We are also looking forward to sharing this new data with a wider public.

Referee #2 (Remarks to the Author):

The authors have addressed all my initial concerns. Congratulations to the authors on an important and thorough manuscript that will open up new areas of biology for investigation.

We thank Referee #2 for the time and thought invested in this manuscript. We share the excitement for the future research building on this work.

Referee #3 (Remarks to the Author):

As previously noted, this study by Muhar et al., reports a novel and potentially very important mechanism how a ubiquitin ligase would recognize damaged proteins for clearance. The experiments are well executed and the work clearly presented. The revised manuscript has addressed many of the points raised and substantially enhanced the overall manuscript, but unfortunately fails to address the key question raised by all three reviewers: Do these modifications exist under physiological conditions and is recognition by FBXO31 the mechanism for clearance. This is unfortunate given that many straight forward experiments were suggested by all reviewers. As reviewer #2 has pointed out, the ability of ubiquitin ligases to recognize c-terminal modifications on peptides has been established by work from Christina Woo's lab and this reviewer agrees that to justify publication in Nature, the existence of such modifications under physiological conditions needs to be established. The in vitro experiments added during revisions confirm that damage to proteins by e.g. H₂O₂ can lead to CTAP, but this is not the key question. If the authors can confirm the existence of those modifications and the necessity of the modifications for recognition by FBXO31 under physiological conditions, the paper would be an outstanding candidate for publication.

Major points

- The authors need to demonstrate the existence of CTAP modifications under physiological conditions using reference peptides or other stringent analytical methods. For example in the elegant neuron differentiation system they use.
- The authors should demonstrate that CTAP modifications on substrates are necessary for recognition by FBXO31 under physiological conditions.
- The simplest experiment to establish the link appears to be at their fingertips. In figure 4d-f they use IP-MS or ABP to identify FBXO31 substrates that are dependent on CTAP. From the MS data, they should be able to identify the CTAP bearing peptides present under these conditions. Using isotope labeled reference peptides, they should establish that H₂O₂ treatment in fact increases the abundance of the CTAP bearing peptides and that this correlates with recognition. The use of reference peptides would establish the presence of CTAP peptides under baseline conditions (e.g. in differentiating neurons) and would allow to establish the quantitative link to recognition.
- The fact that what appears to be the key finding of this paper, a previously overlooked modification renders protein unstable, is not accompanied by a main figure establishing the existence of these modifications under physiological conditions.

We thank Referee #3 for the careful review of our study. We appreciate that our effort *“has addressed many of the points raised and substantially enhanced the overall manuscript”*. The referee's main comment is to emphasize a desire to establish the existence of CTAPs under physiological conditions and that FBXO31 binds these modifications under physiological conditions.

We apologize if this was unclear and if our revised text gave the false impression that we were solely investigating H₂O₂. We performed several experiments exactly to establish the existence and engagement of CTAPs in multiple ways. We validated that CTAPs formed under physiological conditions in human tissues (Extended Data Fig. 6 a-e; Extended Data Fig. 7 c,d), including by deep MS profiling of hemoglobins obtained from healthy human donors (Fig. 4b, Extended Data Fig. 7 a-c). We also showed that endogenously produced ROS stress induces the FBXO31/client interaction by (Fig. 4h). We have edited the text to further explain and highlight these important data.

Future research on CTAP formation and FBXO31 function, specifically in neurons, is indeed an exciting future direction. We have made additional edits to the manuscript to highlight the potential for this avenue of research in the discussion (new page no. 13).

Minor points:

- Figure 2d would benefit from a specificity control by titrating unlabeled peptide

The most stringent way to control for specificity in this assay is by using point-mutants of the interaction face. In Extended Data Fig. 9b we use exactly the same peptide as in Figure 2d. The assay shows that either introducing the binding pocket mutation (D334N) on the protein's side or exchanging the C-terminal -OH for -NH₂ on the peptide abolish the interaction.

- Making protein can't be too hard given X-Ray structure

This was also our initial hope. But we note that a crystal structure emphasizes large amounts of a single protein and not scalability to multiple mutants. After extensive initial expression trials, we concluded that we could obtain sufficient quantities of a small number of SCF/FBXO31 constructs. Therefore, assaying CTAP recognition, CRL assembly and degradation of different CTAPs for all mutants directly in human cells allowed us to test more hypotheses and was also more informative for human cell biology.